# Interpolating Item and User Fairness in Multi-Sided Recommendations

**Qinyi Chen**[1]    **Jason Cheuk Nam Liang**[1]    **Negin Golrezaei**[1]    **Djallel Bouneffouf**[2]
[1]Massachusetts Institute of Technology    [2]IBM Research
{qinyic,jcnliang,golrezae}@mit.edu   djallel.bouneffouf@ibm.com

## Abstract

Today's online platforms heavily lean on algorithmic recommendations for bolstering user engagement and driving revenue. However, these recommendations can impact multiple stakeholders simultaneously—the platform, items (sellers), and users (customers)—each with their unique objectives, making it difficult to find the right middle ground that accommodates all stakeholders. To address this, we introduce a novel fair recommendation framework, Problem (FAIR), that flexibly balances multi-stakeholder interests via a constrained optimization formulation. We next explore Problem (FAIR) in a dynamic online setting where data uncertainty further adds complexity, and propose a low-regret algorithm FORM that concurrently performs real-time learning and fair recommendations, two tasks that are often at odds. Via both theoretical analysis and a numerical case study on real-world data, we demonstrate the efficacy of our framework and method in maintaining platform revenue while ensuring desired levels of fairness for both items and users.

## 1 Introduction

Online recommendation systems have become essential components of various digital platforms and marketplaces, playing a critical role in user experience and revenue generation. These platforms usually operate as multi-sided entities, recommending items (services, products, contents) to users (consumers). Examples include e-commerce (Amazon, eBay, Etsy), service platforms (Airbnb, Google Local Services), job portals (LinkedIn, Indeed), streaming services (Netflix, Spotify), and social media (Facebook, TikTok). On the platform, there are typically two main groups of stakeholders: **items** (products, services, contents) and **users** (those engaging with items), with the **platform** itself acting as an independent stakeholder due to its unique objectives.

However, as these platforms play a more vital role in societal and economic realms, fairness concerns have prompted increasing regulatory action. Notably, the European Union proposed the Digital Markets Act [54], which emphasizes the need for contestable and fair markets in the digital sector, designating recommendation systems as a key area of focus; the U.S. Algorithmic Accountability Act [47] requires companies to assess the impacts of their automated decision systems to ensure they are not biased or discriminatory. It has also become increasingly evident, to the platforms, that upholding fairness across their stakeholder groups not only strengthens relationships with these stakeholders, but also enhances long-term platform sustainability [67].

Fairness concerns within digital platforms significantly impact both users and items, manifesting in various forms of disparities. For users, issues like racial discrimination in Airbnb's host-guest matching [25] and gender disparity in career ads [43] highlight the critical need for enforcing user fairness. Similarly, items on the platforms also experience unfair allocation of opportunities induced by algorithmic decisions, such as e-commerce platforms favoring their own private-label products [19] and social media prioritizing influencers' contents over others [63], indicating a disparity that affects items (sellers, content creators) alike. These underscore the necessity to address biases and create a more inclusive environment for all stakeholders involved.

38th Conference on Neural Information Processing Systems (NeurIPS 2024).

Despite the many efforts by companies to mitigate fairness issues, most of the existing measures solely focus on one stakeholder group, sometimes even at the expense of another. For example, Airbnb has implemented strategies to mitigate racial discrimination among its users [4], yet their listings of comparable relevance might still not receive similar levels of visibility. Etsy implements measures to promote market diversity and supporting new entrants, which contributes to equality of opportunities for its items (sellers) [27, 13]; however, such a strategy may inadvertently disadvantage users by potentially overwhelming them with choices and hindering their ability to quickly find their preferred products. Similar to industrial practices, the majority of ongoing research on fairness in recommender systems (e.g., [13, 55, 71]) also focus solely on either users or items, but seldom both.

Addressing multi-sided fairness is inherently complex due to the competing interests and objectives of different stakeholders [39, 12]. If algorithms solely prioritize the platform's profits, this can lead to inequitable exposure for those more niche items and users who prefer them. In addition, what users perceive as fair may be viewed as unfair by items, and vice versa [16].

In face of these challenges, our work aims to answer the following two questions: (1) *What constitutes a fair recommendation within a multi-sided platform?* and (2) *How would a platform implement a fair recommendation in a practical online setting?* Our contributions are summarized as follows:

1. **A novel fair recommendation framework, Problem (FAIR).** Our fair recommendation problem, framed as a constrained optimization problem, adopts a novel multi-sided perspective that first achieves *within-group fairness* among items/users, and then enforces *cross-group* fairness that addresses the trade-offs *across* groups. Notably, Problem (FAIR) enables the platform to (i) flexibly define fair solutions for items/users rather than relying on a single predefined notion (see Section 2.2); (ii) adjust trade-offs between its own business goals and fairness for stakeholders (see, e.g., our case study in Section 4); (iii) flexibly accommodate additional operational considerations.

2. **A Fair Online Recommendation algorithm for Multi-sided platforms (FORM).** In Section 3, we study an online setting where the platform must ensure fairness for a sequence of arriving users amidst data uncertainty. We present FORM, a low-regret algorithm tailored for fair online recommendation, whose efficacy is further validated via a real-world case study (Section 4). The design of FORM contributes both methodologically and technically. (i) Methodologically, contrary to prior works (e.g., [55, 71]) that treats learning and enforcing fairness as two separate tasks, we recognize that these two tasks, when conducted simultaneously, are often at odds. FORM well balances learning and fairness via properly relaxing fairness constraints and introducing randomized exploration. (ii) Technically, FORM overcomes a non-trivial challenge of managing *uncertain fairness constraints* when solving a constrained optimization problem in an online setting with bandit feedback. While existing works on online constrained optimization all would require certain access to constraint feedback (see discussion in Section 1.1), our setup disallows even verifying the satisfaction of item/user fairness constraints, demanding novel design in FORM.

## 1.1 Related Works

Our work primarily contributes to the emerging area of algorithmic fairness in recommender systems. Additionally, from a technical aspect, our algorithm contributes to the field of constrained optimization with bandit feedback. We highlight the literature most relevant to our work in both areas.

**Algorithmic fairness in recommender systems.** Algorithmic fairness is an emerging topic [8, 2, 42] explored in various contexts such as supervised learning [14, 24], resource allocation [9, 37], opportunity allocation [36], scheduling [50], online matching [46], bandits [6], facility location [34], refugee assignment [29], assortment planning [17], online advertising [21], search [5], and online combinatorial optimization [32]. Our research is primarily aligned with works investigating fairness in recommender systems; see [67, 20] for comprehensive surveys.

The prior works on fairness in recommender systems can be categorized into three streams based on their subjects: *item fairness*, *user fairness* and *joint fairness*. Item fairness focuses on whether the decision treats items fairly, including fair ranking and display of search results [73, 11, 48, 10, 60, 66, 17, 72], similar prediction errors for items' ratings [57], and long-term fair exposure [31]. User fairness examines whether the recommendation is fair to different users, encompassing aspects like similar recommendation quality [26, 45, 68] and comparable explainability [30] across users.

The third stream, also the stream that our work fits in, focuses on joint fairness that concerns whether both items and users are treated fairly. However, the number of works in this stream remain scarce, as

suggested in [1]. Existing works mainly focuses on achieving item and user fairness based on certain pre-specified fairness notions: [13] seeks to balance item and user neighborhoods; [55] proposes a method that guarantees maxmin fair exposure for items and envy-free fairness for users; [71, 52] promote fairness with respect to item exposures and user normalized discounted cumulative gains. Additionally, [70] proposes and compares a family of joint multi-sided fairness metrics to tackle systematic biases in content exposure, and [69] introduces a multi-objective optimization framework that jointly optimizes accuracy and fairness for consumers and producers.

Our work distinguishes from the above works on multi-sided fairness in several aspects. (i) We not only impose fairness for multi-stakeholders (items/users), but also take the platform's revenue, which is often neglected, into consideration. (ii) Our framework is not confined to a single, pre-specified fairness/outcome notion. (iii) Most importantly, unlike works that solely focused on multi-sided fairness, we recognize the challenge of jointly handling fairness and learning, and propose an algorithm with theoretical guarantees (see Section 3).

**Constrained optimization with bandit feedback.** Our work formulates the fair recommendation problem in an online setting as a constrained optimization with bandit feedback (see Section 3.1). Our proposed algorithm (Algorithm 1) makes a notable contribution to the literature of constrained optimization with bandit feedback by addressing the challenge of having *uncertain constraints*.

Previous research has explored constrained optimization with bandit feedback in various contexts. Notably, works on *online learning with knapsack* [38, 15, 61] involves maximizing total reward while adhering to a resource consumption constraint, using feedback on rewards and resource usage. Other relevant studies include online bidding [28, 22], online allocation [7], and safe sequential decision-making [62], which develop algorithms to monitor and maintain "constraint balances" or address adversarial objectives and constraints. Our setting crucially differs from prior works by not assuming availability of constraint feedback or even the ability to monitor constraint satisfaction, making our problem more challenging (see, also, Section 3.2 for more discussions).

## 2 Preliminaries

Throughout this paper, boldface symbols denote vectors or matrices, while regular symbols represent scalars. For matrix $\boldsymbol{A} \in \mathbb{R}^{n \times m}$, $A_{i,j}$ is the element at the $i$-th row and $j$-th column; $\boldsymbol{A}_{i,:}$ and $\boldsymbol{A}_{:,j}$ are the $i$-th row and $j$-th column vectors, respectively. We set $[n] := \{1, 2, \ldots, n\}$, $\Delta_n$ as the probability distribution space over $[n]$, and $\Delta_n^m = \Delta_n \times \cdots \times \Delta_n$ (m times). For any $v \in \mathbb{R}$, $(v)^+ := \max(v, 0)$.

### 2.1 Platform's Recommendation Problem

Consider a platform performing recommendations for $T$ rounds, each indexed by $t \in [T]$, where one user visits the platform at each round. There are $N$ items on the platform, each indexed by $i \in [N]$, and the users are classified into $M$ types, each indexed by $j \in [M]$. At each round $t$, a type-$J_t$ user arrives and the platform observes the user's type.[1] Here, we consider a stochastic setting where the probability of having a type-$j$ user arrival is $\mathbb{P}[J_t = j] = p_j$, where $\boldsymbol{p} = (p_j)_{j \in [M]} \in \Delta_M$ denotes the *arrival probabilities*[2]. The platform needs to select an item to display to the user, denoted by $I_t$. If a type-$j$ user is presented with item $i$, he/she would choose/purchase the item with probability $y_{i,j} \in (0, 1)$, where $\boldsymbol{y} = (y_{i,j})_{\substack{i \in [N] \\ j \in [M]}}$ denotes the *purchase probabilities*. The platform then observes the user's purchase decision $z_t \in \{0, 1\}$. If item $i$ gets chosen/purchased, it generates revenue $r_i > 0$, where $\boldsymbol{r} = (r_i)_{i \in [N]}$ denotes the *revenues*.

Let $\boldsymbol{\theta} = (\boldsymbol{p}, \boldsymbol{y}, \boldsymbol{r}) \in \Theta$, where $\Theta = \Delta_M \times (0, 1)^{N \times M} \times \mathbb{R}^N$, define an instance of the platform's recommendation problem. Given the problem instance $\boldsymbol{\theta}$, the platform needs to determine its recommendation probabilities $\boldsymbol{x} \in \Delta_N^M$, where $x_{i,j}$ is the probability of offering item $i$ upon observing a type-$j$ user's arrival.[3] In Section 2, we first fix the problem instance $\boldsymbol{\theta}$ and omit variables'

---

[1]Real-world recommendation systems can often categorize users based on available features, even when sensitive attributes (e.g., gender, race) are restricted. These systems use data such as device types [64], zip codes [44] and purchase histories, ensuring that users with similar attributes tend to have aligned preferences.

[2]We later extend our model/method to handling periodic arrivals; see our discussion in Sections 3.6 and E.1.

[3]In our setting, a platform can make personalized recommendations based on type of the arriving user. However, even if personalization is limited due to legal considerations (e.g., Digital Market Act [54]) or user privacy settings, our framework, method and theoretical results all remain valid.

dependency on $\boldsymbol{\theta}$ whenever the context allows. Later, we will transition to an online setting where the problem instance $\boldsymbol{\theta}$ becomes unknown (see Section 3) reestablish the dependency in our discussion.

Here, we focus on a single-item recommendation setting, where the platform highlights one item to each arriving user. This approach applies to various real-world scenarios, such as Spotify's "Song of the Day", Amazon's "Best Seller", Medium's daily "Must-Read" article, etc. Later in Section 3.6 and our case study on Amazon review data in Section 4, we will show that the core concepts of our model and approach can naturally extend to recommending an assortment of items.

## 2.2 Single-Sided Fair Solutions: Within-Group Fairness for Items and Users

To properly address multi-sided fairness, we begin by first considering *within-group fairness*. That is, we seek to answer: *what constitutes a fair solution within our item/user group*? To address this question, it is important to understand the different *outcomes* that items and users respectively care about, and the fairness notions they would like to consider.

**Single-Sided Item-Fair Solutions.** Let $O_i^{\mathtt{I}}(\boldsymbol{x})$ be the expected outcome received by item $i$ at a round with recommendation probabilities $\boldsymbol{x}$. We let $O_i^{\mathtt{I}}(\boldsymbol{x})$ take the following general form: $O_i^{\mathtt{I}}(\boldsymbol{x}) = \boldsymbol{L}_{i,:}^{\top} \boldsymbol{x}_{i,:}$, where $\boldsymbol{L} = (L_{i,j})_{\substack{i \in [N] \\ j \in [M]}}$ and $L_{i,j}$ can be any proxy for the expected outcome received by item $i$ from type-$j$ user if it gets offered. One can consider any of the following metrics or their weighted combinations as the item's outcome: (1) *visibility*: $L_{i,j} = p_j$ and $O_i^{\mathtt{I}}(\boldsymbol{x}) = \sum_j p_j x_{i,j}$; (2) *marketshare*: $L_{i,j} = p_j y_{i,j}$ and $O_i^{\mathtt{I}}(\boldsymbol{x}) = \sum_j p_j y_{i,j} x_{i,j}$; (3) *expected revenue*: $L_{i,j} = r_i p_j y_{i,j}$ and $O_i^{\mathtt{I}}(\boldsymbol{x}) = \sum_j r_i p_j y_{i,j} x_{i,j}$.

For items, a common way to achieve fairness is via the optimization of a *social welfare function* (SWF), denoted as $W$, which merges fairness and efficiency metrics into a singular objective (see [18]). The maximization of SWF can incorporate a broad spectrum of fairness notions commonly adopted in practice, including maxmin fairness [58], Kalai-Smorodinsky (K-S) bargaining solution [41], Hooker-Williams fairness [37], Nash bargaining solution [53], demographic parity, etc. (See Section A.2 for an expanded discussion of these fairness notions and their social welfare functions.)

To preserve the generality of our framework, we let the platform freely determine the outcome function and fairness notion for items, by broadly defining the *item-fair solution* as follows.

**Definition 2.1 (Item-Fair Solution)** *Given items' outcome matrix $\boldsymbol{L}(\boldsymbol{\theta})$, and a SWF $W : \Delta_N^M \to \mathbb{R}$, an item-fair solution is given by: $\boldsymbol{f}^{I} \in \arg\max_{\boldsymbol{x} \in \Delta_N^M} W(\boldsymbol{O}^{I}(\boldsymbol{x}))$.*

One popular fairness notion that can be captured by Definition 2.1 is maxmin fairness, which ensures that the most disadvantaged item is allocated a fair portion of the outcome. A few prior works [55, 72] on fair recommendation solely focus on achieving maxmin fairness for the visibilities (exposure) received by the items. Our model, however, can flexibly accommodate any outcome functions as listed above and any fairness notions supported via SWF maximization (see Section A.2).

**Single-Sided User-Fair Solutions.** Let $O_j^{\mathtt{U}}(\boldsymbol{x})$ be the expected outcome received by a type-$j$ user at a round with recommendation probabilities $\boldsymbol{x}$. We again let $O_j^{\mathtt{U}}(\boldsymbol{x})$ take a general form: $O_j^{\mathtt{U}}(\boldsymbol{x}) = \boldsymbol{U}_{:,j}^{\top} \boldsymbol{x}_{:,j}$, where $\boldsymbol{U} = (U_{i,j})_{\substack{i \in [N] \\ j \in [M]}}$ and $U_{i,j}$ is the expected outcome received by a type-$j$ user if offered item $i$. Here, the specific form of the user's outcome matrix $\boldsymbol{U}(\boldsymbol{\theta})$ is determined by user's utility model, which can vary depending on contexts. See Section A.1 for some example forms of $\boldsymbol{U}(\boldsymbol{\theta})$ based on discrete choice models (multinomial logit (MNL), probit) used in demand modeling [65] and valuation-based models used in online auction design [51]. For generality, we do not impose restrictions on the form of $\boldsymbol{U}(\boldsymbol{\theta})$, but merely assume knowledge of it.

For users, achieving fairness is more straightforward. Since the platform can personalize recommendations based on user types, given the users' outcome matrix $\boldsymbol{U}(\boldsymbol{\theta})$, it is best for type-$j$ users to consistently receive the item that offers the highest utility. We thus define the *user-fair solution* as:

**Definition 2.2 (User-Fair Solution)** *Given users' outcome matrix $\boldsymbol{U}(\boldsymbol{\theta})$, the user-fair solution $\boldsymbol{f}^{U} \in \Delta_N^M$ is given by $\boldsymbol{f}_{i,j}^{U} = 1$ if $i = \arg\max_i U_{i,j}$ and $\boldsymbol{f}_{i,j}^{U} = 0$ otherwise, for all $i \in [N], j \in [M]$.*

## 2.3 Drawbacks of a Single-Sided Solution

While a single-sided (fair) solution for one stakeholder group can be straightforward to identify, a recommendation policy that solely benefits a single stakeholder group (platform, items, users) could result in extensive costs for the rest of the stakeholders, as suggested by the following proposition.

**Proposition 2.1** *Let $\epsilon_1 = \min \boldsymbol{y} / \max \boldsymbol{y}, \epsilon_2 = \min \boldsymbol{U} / \max \boldsymbol{U}$, and $\epsilon = \max\{\epsilon_1, \epsilon_2, 1/N\}$. There exists a problem instance such that: (1) The platform's revenue-maximizing solution results in zero outcomes for some items and all users attaining only $\epsilon$ of their maximum attainable outcome. (2) The item-fair solution leads to the platform receiving at most $2\epsilon$ of its maximum attainable revenue and any user achieving at most $2\epsilon$ of their maximum attainable outcome. (3) The user-fair solution results in zero outcomes for some items and the platform receiving $\epsilon$ of its maximum attainable revenue.*

Proposition 2.1 shows that single-sided solutions that can lead to extremely unfair outcomes for some stakeholders under certain scenarios. This motivates us to investigate the concept of *cross-group fairness* in the following section, where we aim to balance the needs of all stakeholders.

## 2.4 Multi-Sided Solutions: From Within-Group Fairness to Cross-Group Fairness

In this section, we proceed to investigate the concept of *cross-group fairness* by proposing a fair recommendation framework, Problem (FAIR). Specifically, we seek to answer the following: *what constitutes a fair recommendation policy across all stakeholder groups on a multi-sided platform?*

The platform, our main stakeholder, primarily wishes to optimize its expected revenue, denoted by $\text{REV}(\boldsymbol{x}) = \sum_{i,j} r_i p_j y_{i,j} x_{i,j}$. Whenever a type-$j$ user arrives, a revenue-maximizing platform with full knowledge of the instance would show the most profitable item, i.e., $i_j^\star = \arg\max_{i \in [N]} r_i y_{i,j}$, with probability one. We let $\text{OPT-REV} = \max_{\boldsymbol{x}} \text{REV}(\boldsymbol{x})$ be platform's maximum attainable expected revenue in the absence of fairness. Such a deterministic revenue-maximizing approach, as shown in Section 2.3, is evidently undesirable for the less profitable items and any user who prefer them.

To create a fair ecosystem, the platform wishes to impose certain levels of fairness for all items/users. In an offline setting with full knowledge of the instance $\boldsymbol{\theta}$, the platform can first compute item-fair and user-fair solutions $\boldsymbol{f}^{\text{I}}, \boldsymbol{f}^{\text{U}}$ and solve the following fair recommendation problem, Problem (FAIR), to prioritize its revenue maximization goal, while achieving cross-group fairness via fairness constraints:

$$\text{FAIR-REV} = \max_{\boldsymbol{x} \in \Delta_N^M} \text{REV}(\boldsymbol{x}) \quad \text{s.t.} \quad O_i^{\text{I}}(\boldsymbol{x}) \geq \delta^{\text{I}} \cdot O_i^{\text{I}}(\boldsymbol{f}^{\text{I}}) \ \forall \ i \in [N]$$

$$O_j^{\text{U}}(\boldsymbol{x}) \geq \delta^{\text{U}} \cdot O_j^{\text{U}}(\boldsymbol{f}^{\text{U}}) \ \forall \ j \in [M], \tag{FAIR}$$

where $\delta^{\text{I}}, \delta^{\text{U}} \in [0, 1]$ are the fairness parameters for items and users respectively. Here, the item/user fairness constraints ensure that the outcome received by any item/user is at least of a certain proportion of the outcome that they would otherwise receive under their within-group fair solution. One can think of $\delta^{\text{I}}$ and $\delta^{\text{U}}$ as tunable handles that the platform can adjust to regulate the tradeoff between fairness and its own revenue (see how a platform can tune these parameters in practice in Sections 4 and C). If a platform wishes to incorporate other operational considerations such as diversity constraints ($x_{\min} \leq x_{i,j} \leq x_{\max}$) or fairness for more stakeholder groups beyond items/users (e.g., DoorDash drivers, Airbnb hosts), additional constraints can be incorporated in a similar fashion.

We let $\boldsymbol{x}^\star$ denote the optimal solution to Problem (FAIR), which serves as the benchmark that we compare our algorithm against when we next work with the online setting (see Section 3).

**Remark 2.1 (Constrained optimization versus fairness regularizers)** *An alternative method of integrating fairness is to use fairness regularizers (i.e., Lagrangian multipliers) and optimize a weighted sum of platform's revenue and items'/users' outcomes. Nonetheless, our constrained optimization formulation offers several clear benefits: (i) while it is possible to translate our constrained optimization formulation into a weighted sum with fairness regularizers via dualization, the reverse process is not straightforward; (ii) our fairness parameters $\delta^I, \delta^U$ are directly interpretable; (iii) our framework accommodates additional operational considerations, with results remaining valid.*

**Remark 2.2 (Selecting fairness parameters via "price of fairness")** *To choose the right fairness parameters $\delta^I, \delta^U$, it is important to understand both (i) the extent of fairness needed for items/users,*

*and (ii) the cost of implementing fairness constraints to balance its revenue and stakeholder interests, often known as the "price of fairness" (PoF). In Section C, we investigate the concept of PoF under our framework (Problem (FAIR)), and show a piecewise linear dependency of the PoF on both (i) the amount of misalignment in the platform's and its stakeholders' objectives, (ii) the fairness parameters $\delta^I, \delta^U$ (see Theorem C.1). We further provide guidelines on how a platform can effectively tune its fairness parameters based on its desired fairness level and acceptable PoF via online A/B experimentation (see Section C for an extended discussion).*

## 3 Fair Online Recommendation Algorithm for Multi-Sided Platforms

In practice, user data are often missing or inaccurate, so solving Problem (FAIR) with flawed data could inadvertently result in unfair outcomes. Our online setting (Section 3.1) allows the platform to improve its estimates of user data via a data collection process over time. However, this simultaneous process of learning to be fair and understanding user preferences also introduces new challenges. In this section, we address the more intricate online setting and introduce an algorithm for this scenario.

### 3.1 Online Setting and Goals

In an online setting, the platform no longer has prior knowledge of user preference $\boldsymbol{y}$ and arrival rates $\boldsymbol{p}$. At each round $t$, some non-anticipating algorithm $\mathcal{A}$ generates the recommendation probabilities $\boldsymbol{x}_t \in \Delta_N^M$, only based on past recommendation $\{\boldsymbol{x}_{t'}\}_{t'<t}$ and purchase decisions $\{z_{t'}\}_{t'<t}$. After observing the type of the arriving user $J_t$, the platform offers item $i$ with probability $x_{t,i,J_t}$. Following this recommendation, the platform only observes the user's binary purchase decision $z_t \in \{0, 1\}$ for the offered item. We will measure the performance of our algorithm using the following metrics:

**Definition 3.1 (Revenue and fairness regrets)** *Consider Problem (FAIR) under a problem instance $\boldsymbol{\theta} \in \Theta$, item outcome function $\boldsymbol{L}(\boldsymbol{\theta}) : \Theta \to \mathbb{R}^N$, user outcome function $\boldsymbol{U}(\boldsymbol{\theta}) : \Theta \to \mathbb{R}^M$, item social welfare function $W : \mathbb{R}^N \to \mathbb{R}$, and fairness parameters $\delta^I, \delta^U \in [0, 1]$. Let $\mathcal{A}$ be a non-anticipating algorithm that generates recommendation probabilities $\boldsymbol{x}_t$ at each round $t \in [T]$. We define the **revenue regret**, denoted by $\mathcal{R}(T)$, and the **fairness regret**, denoted by $\mathcal{R}_F(T)$, of $\mathcal{A}$ respectively as*

$$\mathcal{R}(T) = \frac{1}{T} \sum_{t=1}^{T} \left( \text{REV}(\boldsymbol{x}^\star, \boldsymbol{\theta}) - \text{REV}(\boldsymbol{x}_t, \boldsymbol{\theta}) \right) \quad \text{and} \quad \mathcal{R}_F(T) = \max\{\mathcal{R}_F^I(T), \mathcal{R}_F^U(T)\},$$

*where $\mathcal{R}_F^I(T) = \max_i \frac{1}{T} \sum_{t=1}^{T} (\delta^I \cdot O_i^I(\boldsymbol{f}^I(\boldsymbol{\theta}), \boldsymbol{\theta}) - O_i^I(\boldsymbol{x}_t, \boldsymbol{\theta}))^+$ and $\mathcal{R}_F^U(T) = \max_j \frac{1}{T} \sum_{t=1}^{T} (\delta^U \cdot O_j^U(\boldsymbol{f}^U(\boldsymbol{\theta}), \boldsymbol{\theta}) - O_j^U(\boldsymbol{x}_t, \boldsymbol{\theta}))^+$ are respectively the maximum time-averaged violation of item/user-fair constraints. Here, $\text{REV}(\boldsymbol{x}, \boldsymbol{\theta}) = \sum_{i,j} r_i p_j y_{i,j} x_{i,j}$ is platform's expected revenue under instance $\boldsymbol{\theta}$; $O_i^I(\boldsymbol{x}, \boldsymbol{\theta}) = \boldsymbol{L}_{i,:}(\boldsymbol{\theta})^\top \boldsymbol{x}_{i,:}$ and $O_j^U(\boldsymbol{x}, \boldsymbol{\theta}) = \boldsymbol{U}_{:,j}(\boldsymbol{\theta})^\top \boldsymbol{x}_{:,j}$ are the item/user outcomes under $\boldsymbol{\theta}$; $\boldsymbol{f}^I(\boldsymbol{\theta})$ and $\boldsymbol{f}^U(\boldsymbol{\theta})$ are the item/user-fair solutions w.r.t. $\boldsymbol{L}(\boldsymbol{\theta}), \boldsymbol{U}(\boldsymbol{\theta})$ and SWF $W$, per Definition 2.1 and 2.2.*

Observe that in Definition 3.1, we reintroduce the dependency of our variables on the instance $\boldsymbol{\theta}$. This is because in the online setting, we typically work with an estimated instance, which in turn impacts our estimates for the platform's revenue, item/user outcomes as well as item/user-fair solutions.

### 3.2 Challenges of the Online Setting

Before proceeding, we highlight the two main challenges unique to the online setting.

**(1) Data uncertainty and partial feedback can interfere with evaluating fairness.** Due to lack of knowledge of the instance $\boldsymbol{\theta}$ and limited feedback (as we only observe the purchase decision for the offered items), it is difficult to assess the quality of our recommendations $\boldsymbol{x}_t$ at each round. In particular, verifying whether our fairness constraints are satisfied and/or measuring the amount of constraint violations require evaluating the item/user-fair solutions $\boldsymbol{f}^I(\boldsymbol{\theta}), \boldsymbol{f}^U(\boldsymbol{\theta})$ and item/user outcomes $O_i^I(\boldsymbol{f}^I(\boldsymbol{\theta}), \boldsymbol{\theta}), O_i^I(\boldsymbol{x}_t, \boldsymbol{\theta})$ and $O_j^U(\boldsymbol{f}^U(\boldsymbol{\theta}), \boldsymbol{\theta}), O_j^U(\boldsymbol{x}_t, \boldsymbol{\theta})$, all of which heavily depend on $\boldsymbol{\theta}$.

This sets our work apart from prior works on fairness in recommender systems, which assume full knowledge of the problem instance (e.g., [55, 71]), and from works on constrained optimization with bandit feedback (e.g., [38, 15, 61]), which rely on the availability of constraint feedback. As we

discussed in Section 1.1, the latter works need to access the amount of constraint violation or verify constraint satisfaction after each decision to update their policies. For example, in online learning with knapsack, the constraint is a resource budget, so constraint violations can be directly evaluated. Our work contributes to the literature on constrained optimization with bandit feedback by directly handling *uncertain constraints* and providing a sublinear regret bound (see Theorem 3.1).

**(2) Fairness can interfere with the quality of learning.** To ensure fairness for all stakeholder groups, some items (e.g., low-revenue or low-utility items) would necessarily receive lower recommendation probabilities than the others. Nonetheless, if we barely offer these items to the users, the lack of exploration could also lead to poor estimation for their purchase probability.

### 3.3 Algorithm Description

We now present our algorithm, called FORM (**F**air **O**nline **R**ecommendation algorithm for **M**ulti-sided platforms), which handles the aforementioned challenges by adopting a relaxation-then-exploration technique, allowing it to achieve both low revenue regret and fairness regret. The design of FORM is outlined in Algorithm 1, consisting of the following.

---

**Algorithm 1** Fair Online Recommendation Algorithm for Multi-Sided Platforms (FORM)

---

**Input:** (i) $N$ items with revenues $\boldsymbol{r} \in \mathbb{R}^N$, item outcome $\boldsymbol{L}(\boldsymbol{\theta}) : \Theta \to \mathbb{R}^{N \times M}$, item SWF $W : \Delta_N^M \to \mathbb{R}$; (ii) $M$ types of users with user outcome $\boldsymbol{U}(\boldsymbol{\theta}) : \Theta \to \mathbb{R}^{N \times M}$; (iii) fairness parameters $\delta^{\mathrm{I}}, \delta^{\mathrm{U}} \in [0, 1]$.

1. **Initialization.** Set $\hat{y}_{1,i,j} = 1/2$ and $\hat{p}_j = 1/M$ for $i \in [N], j \in [M]$. Let the magnitude of exploration be $\epsilon_t = \min\left\{N^{-1}, N^{-\frac{2}{3}} t^{-\frac{1}{3}}\right\}$, and define the magnitude of relaxation as

$$\eta_t = M \log(T) \max\{\Gamma_{y,t}, \Gamma_{p,t}\}, \tag{1}$$

   where $\Gamma_{y,t} = 2\log(T)/\sqrt{\epsilon_t \cdot \max\{1, t/\log(T) - \sqrt{t\log(T)/2}\}}$ and $\Gamma_{p,t} = 5\sqrt{\log(3T)/t}$.

2. For $t = 1, \dots, T$
   (a) **Solve a relaxed version of Problem** (FAIR) **under data uncertainty.** Given estimated instance $\hat{\boldsymbol{\theta}}_t = (\hat{\boldsymbol{p}}_t, \hat{\boldsymbol{y}}_t, \boldsymbol{r})$ and relaxation $\eta_t$, let $\hat{\boldsymbol{x}}_t$ be the optimal solution to Problem (FAIR-RELAX($\hat{\boldsymbol{\theta}}_t, \eta_t$)).
   (b) **Recommend with randomized exploration.**
      • Let the recommendation probability be $\boldsymbol{x}_{t,i,j} = (1 - N\epsilon_t)\hat{\boldsymbol{x}}_{t,i,j} + \epsilon_t$ for all $i \in [N], j \in [M]$.
      • Observe the type of the arriving user $J_t$ and offer item $I_t$ based on probabilities $I_t \sim \boldsymbol{x}_{t,:,J_t}$.
      • Observe purchase decision $z_t \in \{0, 1\}$ and update the number of user arrivals: $n_{J_t,t} = n_{J_t,t-1} + 1$.
   (c) **Update estimates for purchase and arrival probabilities.** Let

$$\hat{y}_{t+1,i,j} = \frac{1}{n_{j,t}} \sum_{k=1}^{n_{j,t}} \mathbb{I}\{I_{\tau_{j,k}} = i, z_{\tau_{j,k}} = 1\}/x_{\tau_{j,k},i,j}, \quad \hat{p}_{t+1,j} = \frac{1}{t}n_{j,t}, \quad \hat{\boldsymbol{\theta}}_{t+1} = (\hat{\boldsymbol{p}}_{t+1}, \hat{\boldsymbol{y}}_{t+1}, \boldsymbol{r}), \tag{2}$$

   where $\tau_{j,k} \in [T]$ denote the round at which the $k$th type-$j$ user arrives.

---

**Solve a relaxed version of Problem** (FAIR) **under the estimated instance.** Recall, from our first challenge, that we cannot directly verify if the fairness constraints have been satisfied due to having data uncertainty and partial feedback. If the platform solves Problem (FAIR) using the estimated instance, the flaw in estimation could easily lead to failure of maintaining fairness for some stakeholders. In face of this, FORM solves a *relaxed* version of Problem (FAIR), defined as follows:

$$\max_{\boldsymbol{x} \in \Delta_N^M} \quad \mathrm{REV}(\boldsymbol{x}, \hat{\boldsymbol{\theta}}_t) \quad \text{s.t.} \quad O_i^{\mathrm{I}}(\boldsymbol{x}, \hat{\boldsymbol{\theta}}_t) \geq \delta^{\mathrm{I}} \cdot O_i^{\mathrm{I}}(\boldsymbol{f}^{\mathrm{I}}(\hat{\boldsymbol{\theta}}_t), \hat{\boldsymbol{\theta}}_t) - \eta_t \quad \forall i \in [N]$$

$$O_j^{\mathrm{U}}(\boldsymbol{x}, \hat{\boldsymbol{\theta}}_t) \geq \delta^{\mathrm{U}} \cdot O_j^{\mathrm{U}}(\boldsymbol{f}^{\mathrm{U}}(\hat{\boldsymbol{\theta}}_t), \hat{\boldsymbol{\theta}}_t) - \eta_t \quad \forall j \in [M]. \quad \text{(FAIR-RELAX}(\hat{\boldsymbol{\theta}}_t, \eta_t))$$

where $\hat{\boldsymbol{\theta}}_t$ is the estimated instance at round $t$ and $\eta_t > 0$ is a parameter that regulates the magnitude of relaxation on our fairness constraints (Eq. (1)). Here, Problem (FAIR-RELAX($\hat{\boldsymbol{\theta}}_t, \eta_t$)) differs from Problem (FAIR) in that (i) it uses the estimated instance $\hat{\boldsymbol{\theta}}_t$ rather than the ground-truth instance $\boldsymbol{\theta}$, and (ii) it relaxes all fairness constraints by the amount of $\eta_t$. At a high level, the relaxation here ensures that the solution fair to all stakeholders (i.e., $\boldsymbol{x}^\star$) would be captured even under data uncertainty. The magnitude of relaxation $\eta_t$ depends on $\Gamma_{y,t}$ and $\Gamma_{p,t}$, which are respectively confidence bounds associated with estimated purchase/arrival probabilities (see Definition D.1). As our estimates become more accurate, both confidence bounds shrink, hence decreasing the magnitude of fairness relaxation.

**Recommend with randomized exploration.** In order to handle the second challenge of some items being inadequately explored, we incorporate randomized exploration by sampling from a distribution that perturbs the estimated solution to Problem (FAIR), $\hat{x}_t$, by a carefully tailored amount $\epsilon_t$. This allows ongoing exploration of all items, with the magnitude of exploration $\epsilon_t$ also decreasing over time as our parameter estimates improve, shifting from exploration towards greater exploitation.

**Unbiased estimators for our problem instance.** In Algorithm 1, for simplicity, we estimate purchase probabilities $y$ using an inverse probability weighted estimator and estimate arrival probabilities $p$ with the sample mean (See Eq. (2)). For sufficiently large $t$, our estimates $\hat{y}_t$ and $\hat{p}_t$ will be accurate with high probability (see Lemma D.2). In practice, the platform, potentially with access to historical data, can freely use any learning mechanism that yield unbiased estimators for user preferences and arrival rates. As long as the estimates get sufficiently accurate over time with high probability, FORM would ensure low revenue/fairness regrets, all while keeping the rest of its design unchanged.

### 3.4 Theoretical Analysis

We theoretically analyze the performance of FORM, under a mild local Lipschitzness assumption.

**Assumption 3.1** *Given instance $\theta \in \Theta$, there exists constants $B, \zeta > 0$ such that for any $\tilde{\theta} \in \Theta$ where $\|\tilde{\theta} - \theta\|_\infty \le \zeta$, $\max\{\|U(\theta) - U(\tilde{\theta})\|_\infty, \|f^I(\theta) - f^I(\tilde{\theta})\|_\infty\} \le B\|\theta - \tilde{\theta}\|_\infty$.*

Assumption 3.1 is well justified in practice. In terms of users' outcome matrix, the assumption readily holds for all prevalent users' choice models and valuation-based models (see examples in Section A.1) as long as $\theta$ is bounded away from the boundaries of $\Theta$. For item-fair solutions $f^I(\theta)$, a wide range of standard outcome functions (visibility, revenue, etc.) and fairness notions (maxmin, K-S, etc.) readily induce locally Lipschitz item-fair solutions; see Section A.3 for some examples.

Theorem 3.1 is the main result of this section, which states that FORM achieves both sublinear revenue regret and fairness regret, as desired. The proof of Theorem 3.1 is deferred to Section D.

**Theorem 3.1 (Performance of FORM)** *Given any problem instance $\theta \in \Theta$ and assume that Assumption 3.1 holds, for $T$ sufficiently large, we have that*

- *the revenue regret of FORM is at most $\mathbb{E}[\mathcal{R}(T)] \le \mathcal{O}(MN^{\frac{1}{3}}T^{-\frac{1}{3}})$;*
- *the fairness regret of FORM is at most $\mathbb{E}[\mathcal{R}_F(T)] \le \mathcal{O}(MN^{\frac{1}{3}}T^{-\frac{1}{3}})$.*

### 3.5 Computational complexity and scalability

In terms of the computational complexity of FORM, the dominant runtime cost in each iteration of FORM arises from solving Problem (FAIR-RELAX($\hat{\theta}_t, \eta_t$)). Note that for a wide variety of commonly used item-fairness notions (e.g., maxmin, K-S, demographic parity; see Table 1) and item outcome functions (e.g., visibility, revenue), solving Problem (FAIR-RELAX($\hat{\theta}_t, \eta_t$)) involves solving two linear programs with $MN$ variables, which is solvable in polynomial runtime $O^*((MN)^{2+1/18})$ [40]. The remaining operations in each iteration of FORM takes $O(MN)$ time. Consequently, each iteration of FORM has a worst-case complexity of $O^*((MN)^{2+1/18})$. In practice, however, much better performance can often be achieved by advanced LP solvers such as Gurobi and CPLEX.

For real-world deployments, FORM can be further adapted with scalability in consideration. First, in practice, there is no need to solve Problem (FAIR-RELAX($\hat{\theta}_t, \eta_t$)) at every user arrival. Instead, platforms can resolve the problem after a given number of user arrivals or periodically at fixed time intervals, while updating user data in real-time. This allows majority of the iterations to run in $O(MN)$ time and removes the computational overhead.[4] Second, we do not always encounter a large-scale optimization problem when applying our framework. Real-world recommendation systems often narrow down items through lightweight pre-filtering stages based on criteria like keywords or price range (e.g., [49]), allowing us to enforce fairness within smaller, context-specific subsets. Our fairness framework is also particularly impactful at this final stage, where items with similar attributes compete for visibility and a revenue-maximizing strategy could lead to extremely unfair outcomes.

---

[4]See our additional experiments on MovieLens data in Section F.3, where we resolve the constrained optimization problem at most once every 100 user arrivals, while our algorithm remains effective.

### 3.6 Extensions to Additional Setups

Our framework, Problem (FAIR), and the proposed algorithm can be extended to accommodate other variations of our setup. Below, we briefly introduce these extensions; see Section E for more details.

**Periodic arrivals.** While our current model focuses on stochastic arrivals with fixed user arrival probabilities $p$, real-world recommendation systems often observe non-stationary user arrivals with hourly, daily or weekly periodicity. In Section E.1, we show that by additionally integrating a sliding window mechanism, FORM can seamlessly accommodate periodic arrivals, and attain the same $\mathcal{O}(MN^{1/3}T^{-1/3})$ guarantees for both revenue and fairness regrets.

**Recommending an assortment.** As remarked in Section 2, while our model adopts a single-item recommendation setting, the high-level ideas behind our framework/method naturally extend to recommending an assortment of size at most $K$. To extend our framework, Problem (FAIR), we will let $\{q_j(S) : S \in [N], |S| \leq K, j \in [M]\}$ be the decision variables, where $q_j(S)$ is the likelihood of proposing assortment $S$ to a type-$j$ user. We can then apply the same relaxation-then-exploration techniques to solve the fair recommendation problem in a dynamic online setting. See Section E.2 for details of our extension. In the case study that follows (Section 4), we also validate the efficacy of our framework/method in a real-world assortment recommendation problem.

## 4 Case Studies on Amazon Review Data

In our case study on Amazon review data, we act as an e-commerce platform displaying featured products to incoming users, aiming to maximize revenue while ensuring fairness for items and users. Our experiments numerically validate the efficacy of our framework/method. All algorithms were implemented in Python 3.7 and run on a MacBook with a 1.4 GHz Quad-Core Intel Core i5 processor.

**Data and setup.** We use an Amazon review dataset [71] from the "Clothing, Shoes and Jewelry" category. Product reviews provide relevance scores between each item and user, serving as a proxy for purchase likelihood. Users are classified into $M = 5$ types using matrix factorization and $k$-means clustering on user feature vectors. The arrival probability $p_j$ is set to the proportion of type-$j$ users. We select $N = 30$ items with the highest variance in relevance scores across user types, indicating a discrepancy between item and user interests and making this a challenging instance. Item revenues $r_i$ are uniformly drawn from $[0.5, 1.5]$. The purchase probability and users' utilities are defined based on the multinomial logit (MNL) model [65]. For a type-$j$ user presented with assortment $S$, the probability of purchasing item $i \in S$ is $y_{i,j} = \frac{e^{v_{i,j}}}{1 + \sum_{i' \in S} e^{v_{i',j}}}$, where $v_{i,j}$ is the relevance score. The user's perceived utility is $\log(1 + \sum_{i' \in S} e^{v_{i',j}})$. Each instance simulates $T = 2000$ user arrivals. Upon arrival, each type-$j$ user is shown an assortment $S$ of up to $K = 3$ items.

The platform's primary goal is to maximize its revenue while ensuring maxmin fairness for items w.r.t. item revenue, and fairness for users w.r.t. utilities from the MNL model. We apply the extension of FORM for recommending assortments, using relaxation-then-exploration techniques to produce fair recommendations while learning user data (see Section E.2). To establish generality of our framework/method, we have also performed additional experiments under alternative outcomes and fairness notions (see Section F.2) and an alternative movie recommendation setting using MovieLens data (see Section F.3). In all cases, our experiments yielded consistent results.

**Baselines.** We consider six baselines for comparisons. Since all baselines assume full knowledge of the instance and lack a learning phase, we let them use our unbiased estimator to update their estimated instance as they observe purchase decisions, and recommend based on these estimates. (i) *greedy*: offers $K$ items with the highest expected revenue, prioritizing platform's goal; (ii) *max-utility*: offers $K$ items with the highest user utilities, a user-centric approach; (iii) *min-revenue*: offers $K$ items generating the least revenue so far, promoting maxmin fairness for items w.r.t. revenue; (iv) *random*: offers $K$ items uniformly at random, promoting maxmin fairness for items w.r.t. visibility; (v) FairRec [55]: an algorithm that addresses two-sided fairness in a *static* setting. While it's not designed for online settings, we adapt it for online arrivals by duplicating users and using the single-shot recommendation solution. It ensures maxmin fairness for item visibility and envy-free fairness for users; (vi) TFROM [71]: addresses two-sided fairness in an online setting, focusing on uniform item visibility and similar user normalized discounted cumulative gain. However, neither FairRec nor TFROM considers platform's revenue; see Section F.1 for more details on these two baselines.

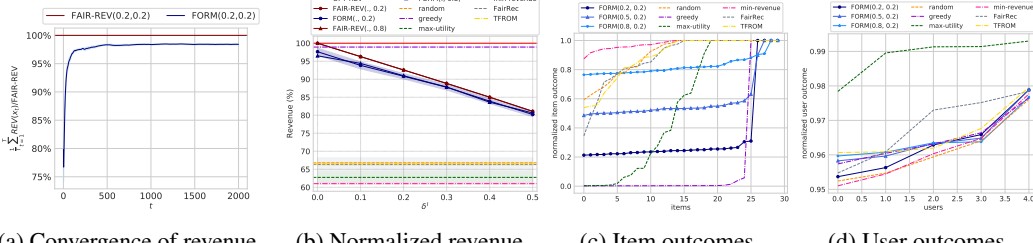

| (a) Convergence of revenue. | (b) Normalized revenue. | (c) Item outcomes. | (d) User outcomes. |

Figure 1: Experiment results for Amazon review data. FAIR-REV$(\delta^{\mathtt{I}}, \delta^{\mathtt{U}})$ is the platform's revenue from solving Problem (FAIR) in hindsight with fairness parameters $\delta^{\mathtt{I}}, \delta^{\mathtt{U}}$ and FORM$(\delta^{\mathtt{I}}, \delta^{\mathtt{U}})$ is FORM when adopting fairness parameters $\delta^{\mathtt{I}}, \delta^{\mathtt{U}}$. In Figures 1c and 1d, item (user) outcomes are shown in ascending order. All results are averaged over 10 simulations, with the line indicating the mean and shaded region showing mean $\pm \, \text{std}/\sqrt{10}$.

**Platform's revenue.** We first assess FORM's efficacy in achieving a low-regret solution. Figure 1a shows that the time-averaged revenue of FORM, i.e., $\frac{1}{T}\sum_{t=1}^{T} \text{REV}(\boldsymbol{x}_t)$, converges rapidly to the optimal revenue for Problem (FAIR), complementing Theorem 3.1. Figure 1b compares the time-averaged revenue (normalized by OPT-REV) of FORM with other baselines. As expected, *greedy* achieves revenue close to OPT-REV, while all other baselines result in significant revenue loss. FairRec and TFROM, in particular, reduce platform revenue by about 38% as their sole focus is on ensuring two-sided fairness. In contrast, FORM offers tunable parameters to balance platform and stakeholder interests. As shown in Figure 1b, adjusting $\delta^{\mathtt{I}}$ and $\delta^{\mathtt{U}}$ allows platforms to control revenue loss (e.g., choosing $\delta^{\mathtt{I}}, \delta^{\mathtt{U}} = 0.2$ keeps loss within 10%). See, also, Section C for how a platform can tune fairness parameters to control its "price of fairness" in practice.

**Item and user fairness.** Figure 1c shows average outcomes for each item $i$, normalized by the outcome under item-fair solution, $\max\{\frac{1}{T}O_i^{\mathtt{I}}(\boldsymbol{x}_t)/O_i^{\mathtt{I}}(\boldsymbol{f}^{\mathtt{I}}), 1\}$. As expected, since our item-fair solution $\boldsymbol{f}^{\mathtt{I}}$ adopts maxmin fairness w.r.t. item revenues, *min-revenue* achieves the highest level of item fairness, though at a high cost to the platform. Methods such as *random*, FairRec, TFROM also achieve high item fairness but with some discrepancies in maximum and minimum item outcomes. *greedy* and *max-utility* show extremely skewed allocations, with some items receiving minimal or no revenue. In comparison, our algorithm FORM strikes a good balance, ensuring all items nearly attain or surpass the specified fairness levels, whether the level is high ($\delta_{\mathtt{I}} = 0.8$) or moderate ($\delta_{\mathtt{I}} = 0.2$).

Figure 1d shows the average outcomes for each user type, normalized by their outcome under the user-fair solution, $\max\{\frac{1}{T}O_j^{\mathtt{U}}(\boldsymbol{x}_T)/O_j^{\mathtt{U}}(\boldsymbol{f}^{\mathtt{U}}), 1\}$. In the Amazon review data, user interests align well with the platform's objectives (as validated in Section C), leading to most baselines performing fairly well and achieving high levels of fairness for the users. FORM again ensures good user outcomes, all while maintaining high platform revenue and desired item fairness levels.

# 5  Conclusion and Future Directions

Our work introduced a novel fair recommendation framework that maintains platform revenue while addressing multi-stakeholder fairness, as well as a low-regret algorithm that effectively produces fair recommendations amidst data uncertainty. It is worth noting that the high-level ideas behind our versatile framework has the potential to be applied in settings beyond recommender systems, such as dynamic pricing and online advertising, ensuring fairness across different stakeholders and promoting stable market conditions in these applications.

There are several future directions worth investigating. (i) Our current framework calibrates recommendation policies within a shorter time period when user preferences and item attributes are relatively fixed. Future research can explore long-term effects of our method by developing adaptive fairness notions that account for evolving user and item attributes and quantifying the long-term multi-stakeholder fairness. (ii) It would be interesting to pursue real-world deployments of our framework/algorithm and evaluate their impact using an expanded set of metrics, such as user satisfaction, retention rates, and recommendation diversity.

## Acknowledgments and Disclosure of Funding

N.G. and Q.C. were partially supported by funding from the Office of Naval Research (ONR) (Award Number: N00014-23-1-2584) and the MIT-IBM Watson AI Lab.

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

Appendices for

# Interpolating Item and User Fairness for Multi-Sided Recommendations



---

## A  Example Choices of Outcomes, Fairness Notions and Fair Solutions

### A.1  Example Utility Functions

Below are some common example utility functions $\boldsymbol{U}(\boldsymbol{\theta})$ based on common discrete choice models (multinomial logit (MNL), probit) used in demand modeling [65] and valuation-based models used in online auction design [51]. For all of these examples, $U_{i,j}$ is strictly increasing w.r.t. the purchase probability $y_{i,j}$.

- **Multinomial logit (MNL) model.** Let the utility of item $i$ for a type $j$ user be $v_{i,j} + \epsilon_{i,j}$, where $v_{i,j} \geq 0$ and $\epsilon_{i,j}$ is the random part drawn i.i.d. from the standard Gumbel distribution. Let $\epsilon_{0,j}$ be the utility of no-purchase option, again drawn i.i.d. from the standard Gumbel distribution. The expected utility is $U_{i,j} = \mathbb{E}\left[\max\{v_{i,j} + \epsilon_{i,j}, \epsilon_{0,j}\}\right] = \log(1 + \exp(v_{i,j})) + \gamma$, where $\gamma$ is the Euler-Mascheroni constant. The purchase probability is given by $y_{i,j} = \mathbb{P}\left[v_{i,j} + \epsilon_{i,j} > \epsilon_{0,j}\right] = \frac{\exp(v_{i,j})}{1+\exp(v_{i,j})}$. Hence, the expected utility can be viewed as $U_{i,j} = \log(\frac{1}{1-y_{i,j}}) + \gamma$.

- **Probit model.** Let the utility of item $i$ for a type $j$ user again take the form of $v_{i,j} + \epsilon_{i,j}$, where the random part $\epsilon_{i,j}$ and the utility of the no-purchase option $\epsilon_{0,j}$ are drawn i.i.d. from a normal distribution $\mathcal{N}(0, \sigma)$. Here, the expected utility $U_{i,j} = \mathbb{E}\left[\max\{v_{i,j} + \epsilon_{i,j}, \epsilon_{0,j}\}\right] = v_{i,j}\Phi\left(\frac{v_{i,j}}{\sqrt{2}\sigma}\right) + \sqrt{2}\sigma\phi\left(\frac{v_{i,j}}{\sqrt{2}\sigma}\right)$, where $\Phi(.)$ and $\phi(.)$ are the CDF and PDF of a standard normal distribution. The purchase probability is $y_{i,j} = \Phi(\frac{v_{i,j}}{\sqrt{2}\sigma})$. We thus have $U_{i,j} = \sqrt{2}\sigma\Phi^{-1}(y_{i,j})y_{i,j} + \sqrt{2}\sigma\phi(\Phi^{-1}(y_{i,j}))$.

- **Valuation-based model.** Let $v_{i,j} \sim F_{i,j}$ be the value of item $i$ for a type $j$ user, where $F_{i,j}$ is the distribution of $v_{i,j}$. Then, $y_{i,j} = \mathbb{P}[(v_{i,j} - r_i) \geq 0]$ and $U_{i,j} = \mathbb{E}[(v_{i,j} - r_i)^+]$, where $r_i$ is the price (revenue) of item $i$. If the valuations follow exponential distribution $v_{i,j} \sim \mathrm{Exp}(\lambda)$ for some $\lambda > 0$, we have purchase probability $y_{i,j} = \exp(-\lambda r_i)$ and expected utility $U_{i,j} = \frac{1}{\lambda}\exp(-\lambda r_i) = y_{i,j}/\lambda$.

### A.2  Example Fairness Notions

Given a group of $S$ stakeholders, each indexed by $s \in [S]$, and any function that maps the platform's decision $\boldsymbol{x}$ to the stakeholders' outcome $\boldsymbol{O}(\boldsymbol{x}) \in \mathbb{R}_+^S$, we can solve the following optimization problem to ensure that each of the $S$ stakeholders are offered a fair outcome: $\boldsymbol{f} = \arg\max_{\boldsymbol{x}} W(\boldsymbol{O}(\boldsymbol{x}))$. where the social welfare function $W$ determines which fairness notion we adopt. In Table 1, we list some example fairness notions commonly adopted in practice, as well as their corresponding social welfare functions $W$. Among them, we have

- *Maxmin fairness* [58]: It maximizes the minimum outcome across all stakeholders, ensuring that the stakeholder with the least favorable outcome is as well-off as possible.
- *Kalai-Smorodinsky (K-S) fairness* [41]: It seeks an equitable outcome where each stakeholder receives a proportional share of their maximum possible outcome. This is represented by a proportional allocation up to a common factor $\beta$.
- *Hooker-Williams fairness [37]*: It aims to balance the efficiency/equity tradeoff using parameter $\Delta$, by prioritizing stakeholders whose outcomes are within $\Delta$ of the minimum outcome.
- *Nash bargaining solution*: It maximizes the product of stakeholders' outcomes, ensuring an equitable distribution that reflects their relative negotiating power and achieves proportional fairness.
- *Demographic parity*: It ensures that outcomes are equally distributed across different demographic groups, by minimizing the disparity in outcomes between a subset of stakeholders $\mathcal{S}'$ and the rest.

See [18] for a comprehensive overview for all of the above fairness notions.

Table 1: Fairness notions and their social welfare functions (SWF).

| Fairness notion | Social welfare function (SWF): $W(\boldsymbol{O}(\boldsymbol{x}))$ |
|---|---|
| Maxmin fairness [58] | $\min_s O_s(\boldsymbol{x})$ |
| Kalai-Smorodinsky (K-S) fairness [41] | $\sum_s O_s(\boldsymbol{x}) \cdot \mathbb{1}\left\{O_s(\boldsymbol{x}) = \beta \max_{\boldsymbol{x}} O_s(\boldsymbol{x}) \text{ for some } \beta \text{ for all } s\right\}$ |
| Hooker-Williams fairness [37] | $\sum_s \max\left\{O_s(\boldsymbol{x}) - \Delta, \min_s O_s(\boldsymbol{x})\right\}$, for some $\Delta \geq 0$ |
| Nash bargaining solution [53] | $\sum_s \log O_s(\boldsymbol{x})$ |
| demographic parity | $1 - \left\|\frac{1}{|\mathcal{S}'|}\sum_{s \in \mathcal{S}'} O_s(\boldsymbol{x}) - \frac{1}{S - |\mathcal{S}'|}\sum_{s \in [S]\setminus\mathcal{S}'} O_s(\boldsymbol{x})\right\|$, for some $\mathcal{S}' \subset [S]$ |

Choosing the right fairness notion is non-trivial and very much depends on the context. Some important considerations include:

- **Stakeholder needs and outcomes.** Understanding the desired outcomes for items and users is crucial. Our framework, Problem (FAIR), is designed to handle various outcome functions (e.g., revenue, marketshare, visibility), which can differ across platforms. For example, a video streaming platform might aim to ensure fair visibility for both independent content creators and popular studios, while an e-commerce platform might focus more on fair marketshare/revenue for small sellers versus large brands.
- **Implication of fairness notion.** Each fairness notion has a different implication, and platforms need to evaluate which best suits their goals and stakeholder needs. For example, maxmin fairness maximizes the outcome received by the most disadvantaged stakeholder, which can be ideal for video streaming platforms like Netflix or YouTube that wish to ensure independent and lesser-known content creators receive fair visibility alongside popular creators. K-S fairness ensures that each individual receives a fair share of his/her maximum attainable outcome. This can be suitable for platforms like LinkedIn or Indeed that wish to ensure fair opportunities for their job seekers relative to their qualifications and experience. Platforms may also need to experiment with different fairness notions to understand how their choices impact their "price of fairness" (see discussion in Section C).
- **Regulatory requirements.** As we discussed in Section 1, one important motivation for imposing fairness in online recommendations is the increasing regulatory action. Therefore, the choice of fairness notions can also depend on legislative or regulatory requirements. For example, since the Digital Markets Act [54] calls for a fair and open digital marketplace, maxmin fairness can be potentially suitable as it ensures even the most disadvantaged item receive a fair level of exposure.

### A.3 Example Item-Fair Solutions with Local Lipschitzness

In this section, we discuss a number of item-fair solutions that adopt different outcome functions and fairness notions, and show that they satisfy Assumption 3.1 in Section 3.4. In the following, we will assume that the item-fair solution satisfies the following condition:

**Condition A.1** *Under problem instance $\boldsymbol{\theta} \in \Theta$, the item outcome $\boldsymbol{L}$ and the social welfare function $W$ are chosen such that*

*(i) $L_{i,j}$ is Lipschitz in $\boldsymbol{\theta}$ for all $i \in [N], j \in [M]$.*

*(ii) The item-fair solution $\boldsymbol{f}^{\mathtt{I}}$ is unique.*

Note that statement (i) in Condition A.1 is satisfied by all of the common definitions of item outcome, such as (1) visibility: $L_{i,j} = p_j$; (2) marketshare: $L_{i,j} = p_j y_{i,j}$; (3) expected revenue: $L_{i,j} = r_i p_j y_{i,j}$. On the other hand, the uniqueness condition can be achieved via adding regularization or lexicographic optimization to the social welfare function. In light of Condition A.1, we enumerate several item-fair solutions adopting prevalent fairness notions (see Section A.2 for definitions) such that local Lipschitzness is satisfied, as outlined in Assumption 3.1.

**Example 1: Item-Fair Solution with Maxmin Fairness.** Suppose that an item-fair solution $\boldsymbol{f}^{\mathtt{I}}$ adopts maxmin fairness, $\boldsymbol{f}^{\mathtt{I}}$ would be the solution to the following linear program (LP):

$$\max_{\boldsymbol{x} \in \Delta_N^M, z \in \mathbb{R}} z \quad \text{s.t.} \quad \boldsymbol{L}_{i,:}^\top \boldsymbol{x}_{i,:} \geq z \quad \forall i \in [N].$$

Let $\boldsymbol{\theta} = (\boldsymbol{p}, \boldsymbol{y}, \boldsymbol{r}), \tilde{\boldsymbol{\theta}} = (\tilde{\boldsymbol{p}}, \tilde{\boldsymbol{y}}, \tilde{\boldsymbol{r}}) \in \Theta$ be two problem instances such that $\|\boldsymbol{\theta} - \tilde{\boldsymbol{\theta}}\|_\infty \leq \zeta$ for some $\zeta > 0$. We consider the two item-fair solutions enforcing maxmin fairness under the two problem instances. For simplicity of notation, we let $\boldsymbol{f}^{\mathrm{I}} = \boldsymbol{f}^{\mathrm{I}}(\boldsymbol{\theta})$ and $\tilde{\boldsymbol{f}}^{\mathrm{I}} = \boldsymbol{f}^{\mathrm{I}}(\tilde{\boldsymbol{\theta}})$; $\boldsymbol{L} = \boldsymbol{L}(\boldsymbol{\theta})$ and $\tilde{\boldsymbol{L}} = \boldsymbol{L}(\tilde{\boldsymbol{\theta}})$. The item-fair solutions are obtained via the following:

$$(\boldsymbol{f}^{\mathrm{I}}, z^\star) = \arg \max_{\boldsymbol{x} \in \Delta_N^M, z \in \mathbb{R}} z \quad \text{s.t.} \quad \boldsymbol{L}_{i,:}^\top \boldsymbol{x}_{i,:} \geq z \quad \forall i \in [N] \,. \tag{3}$$

and

$$(\tilde{\boldsymbol{f}}^{\mathrm{I}}, \tilde{z}^\star) = \arg \max_{\boldsymbol{x} \in \Delta_N^M, z \in \mathbb{R}} z \quad \text{s.t.} \quad \tilde{\boldsymbol{L}}_{i,:}^\top \boldsymbol{x}_{i,:} \geq z \quad \forall i \in [N] \,. \tag{4}$$

We note that

$$\|\boldsymbol{f}^{\mathrm{I}} - \tilde{\boldsymbol{f}}^{\mathrm{I}}\|_\infty \leq \|(\boldsymbol{f}^{\mathrm{I}}, z^\star) - (\tilde{\boldsymbol{f}}^{\mathrm{I}}, \tilde{z}^\star)\|_\infty \,. \tag{5}$$

Consider the region $\mathcal{D} = \{(\boldsymbol{x}, z) : \boldsymbol{L}_{i,:}^\top \boldsymbol{x}_{i,:} \geq z \text{ for all } i \in [N], z \geq z^\star\}$. Under Condition A.1, we know that $(\boldsymbol{f}^{\mathrm{I}}, z^\star)$ is only feasible point in $\mathcal{D}$. We can then invoke Lemma G.1 (with $\boldsymbol{x} = (\boldsymbol{f}^{\mathrm{I}}, z^\star)$ and $\boldsymbol{x}' = (\tilde{\boldsymbol{f}}^{\mathrm{I}}, \tilde{z}^\star)$ and the feasibility region $\mathcal{D}$) to bound the distance between $(\boldsymbol{f}^{\mathrm{I}}, z^\star)$ and $(\tilde{\boldsymbol{f}}^{\mathrm{I}}, \tilde{z}^\star)$ using a Hoffman constant $H > 0$ and a constraint violation term:

$$\|(\boldsymbol{f}^{\mathrm{I}}, \tilde{z}^\star) - (\tilde{\boldsymbol{f}}^{\mathrm{I}}, \tilde{z}^\star)\|_\infty \leq H \cdot \max\{\max_i \|(\tilde{z}^\star - \boldsymbol{L}_{i,:}^\top \tilde{\boldsymbol{f}}_{i,:}^{\mathrm{I}})^+\|_\infty, (z^\star - \tilde{z}^\star)^+\} \,. \tag{6}$$

where the Hoffman constant can be characterized by invoking Lemma G.2 for the matrix defining region $\mathcal{D}$.

Now, let us bound the two terms (i.e. $\max_i \|(\tilde{z}^\star - \boldsymbol{L}_{i,:}^\top \tilde{\boldsymbol{f}}_{i,:}^{\mathrm{I}})^+\|_\infty$ and $(z^\star - \tilde{z}^\star)^+$), on the right-hand side of (6) respectively. Let us first denote $E = \|\boldsymbol{L} - \tilde{\boldsymbol{L}}\|_\infty$.

To bound the first term, given that for all $i \in [N]$, $\tilde{\boldsymbol{f}}^{\mathrm{I}}$ satisfies $\tilde{\boldsymbol{L}}_{i,:}^\top \tilde{\boldsymbol{f}}_{i,:}^{\mathrm{I}} \geq \tilde{z}^\star$ , we must have

$$\tilde{z}^\star - \boldsymbol{L}_{i,:}^\top \tilde{\boldsymbol{f}}_{i,:}^{\mathrm{I}} \leq (\tilde{z}^\star - \tilde{\boldsymbol{L}}_{i,:}^\top \tilde{\boldsymbol{f}}_{i,:}^{\mathrm{I}}) + (\tilde{\boldsymbol{L}}_{i,:}^\top \tilde{\boldsymbol{f}}_{i,:}^{\mathrm{I}} - \boldsymbol{L}_{i,:}^\top \tilde{\boldsymbol{f}}_{i,:}^{\mathrm{I}}) \leq ME \,. \tag{7}$$

To bound the second term, let us additionally consider the following auxiliary problem:

$$(\hat{\boldsymbol{f}}^{\mathrm{I}}, \hat{z}^\star) = \arg \max_{\boldsymbol{x} \in \Delta_N^M, z \in \mathbb{R}} z \quad \text{s.t.} \quad \boldsymbol{L}_{i,:}^\top \boldsymbol{x}_{i,:} \geq z + ME \quad \forall i \in [N] \,. \tag{8}$$

Note that if $\boldsymbol{L}_{i,:}^\top \boldsymbol{x}_{i,:} \geq z + ME$, we must also have $\tilde{\boldsymbol{L}}_{i,:}^\top \boldsymbol{x}_{i,:} \geq \boldsymbol{L}_{i,:}^\top \boldsymbol{x}_{i,:} - ME \geq z$. Hence, the feasibility region of Problem (8) is included in the feasibility region of Problem (4), which implies that

$$\hat{z}^\star \leq \tilde{z}^\star \,. \tag{9}$$

On the other hand, note that Problem (8) is equivalent to

$$\max_{\boldsymbol{x} \in \Delta_N^M, z' \in \mathbb{R}} z' - ME \quad \text{s.t.} \quad \boldsymbol{L}_{i,:}^\top \boldsymbol{x}_{i,:} \geq z' \quad \forall i \in [N] \,. \tag{10}$$

if we perform change of variable $z' = z + ME$. This implies that

$$z^\star = \hat{z}^\star + ME \tag{11}$$

Equations (9) and (11) together imply that

$$z^\star - \tilde{z}^\star \leq ME \,. \tag{12}$$

Finally, combining Equations (5), (6), (7) and (12), we get

$$\|\boldsymbol{f}^{\mathrm{I}} - \tilde{\boldsymbol{f}}^{\mathrm{I}}\|_\infty \leq \|(\boldsymbol{f}^{\mathrm{I}}, \tilde{z}^\star) - (\tilde{\boldsymbol{f}}^{\mathrm{I}}, \tilde{z}^\star)\|_\infty \leq H \cdot \max\{\|(\tilde{z}^\star - \boldsymbol{L}_{i,:}^\top \tilde{\boldsymbol{f}}_{i,:}^{\mathrm{I}})^+\|_\infty, (z^\star - \tilde{z}^\star)^+\}$$
$$\leq H \cdot ME = H \cdot M \|\boldsymbol{L} - \tilde{\boldsymbol{L}}\|_\infty \,.$$

By statement (i) in Condition A.1, we thus establish the local Lipschitzness of item-fair solution $\boldsymbol{f}^{\mathrm{I}}$ that enforces maxmin fairness.

**Example 2: Item-Fair Solution with Hooker-Williams Fairness.** If the platform adopts an item-fair solution $\boldsymbol{f}^{\mathrm{I}}$ w.r.t. Hooker-Williams fairness with some given $\Delta > 0$ (see definition in Section A.2),

$\boldsymbol{f}^{\mathrm{I}}$ would be the solution to a mixed linear integer program, with the following LP relaxation (it is shown that the LP relaxation describes the convex hull of the feasibility set; see [37]):

$$\max_{\substack{\boldsymbol{x}\in\Delta_N^M,\\ z,v_i,w\in\mathbb{R}^+,\delta_i\in[0,1]}} z \quad \text{s.t.} \quad (N-1)\Delta + \sum_{i=1}^N v_i \geq z$$

$$\boldsymbol{L}_{i,:}^\top \boldsymbol{x}_{i,:} - \Delta \leq v_i \leq \boldsymbol{L}_{i,:}^\top \boldsymbol{x}_{i,:} - \Delta\delta_i \qquad \forall i \tag{13}$$

$$w \leq v_i \leq w + (\Gamma-\Delta)\delta_i \qquad \forall i$$

Here, $\Delta > 0$ is a constant that regulates the equity/efficiency tradeoff, and $\Gamma$ can be any constant such that $\Gamma \geq \bar{L}$ where $\bar{L} = \|\boldsymbol{L}\|_\infty$. We would fix $\Gamma = 2M\bar{L} + \Delta$ in the following.

Let $\boldsymbol{\theta}, \tilde{\boldsymbol{\theta}} \in \Theta$ be two problem instances such that $\|\boldsymbol{\theta} - \tilde{\boldsymbol{\theta}}\|_\infty \leq \zeta$ for some $\zeta > 0$. For simplicity of notation, we let $\boldsymbol{L} = \boldsymbol{L}(\boldsymbol{\theta})$ and $\tilde{\boldsymbol{L}} = \boldsymbol{L}(\tilde{\boldsymbol{\theta}})$. Here, using the same techniques as in maxmin fairness, we let $(\boldsymbol{f}^{\mathrm{I}}, z^\star, \boldsymbol{v}^\star, w^\star, \boldsymbol{\delta}^\star)$ denote the solution to Problem (13) under the problem instance $\boldsymbol{\theta}$, and $(\tilde{\boldsymbol{f}}^{\mathrm{I}}, \tilde{z}^\star, \tilde{\boldsymbol{v}}, \tilde{w}, \tilde{\boldsymbol{\delta}})$ denote the solution to Problem (13) under the problem instance $\tilde{\boldsymbol{\theta}}$. We can bound the difference in the two item-fair solutions similarly by

$$\|\boldsymbol{f}^{\mathrm{I}} - \tilde{\boldsymbol{f}}^{\mathrm{I}}\|_\infty \leq \|(\boldsymbol{f}^{\mathrm{I}}, z^\star, \boldsymbol{v}, w, \boldsymbol{\delta}) - (\tilde{\boldsymbol{f}}^{\mathrm{I}}, \tilde{z}^\star, \tilde{\boldsymbol{v}}, \tilde{w}, \tilde{\boldsymbol{\delta}})\|_\infty$$

$$\leq H \cdot \max\{\max_i \|(\tilde{v}_i - (\tilde{\boldsymbol{L}}_{i,:}^\top \tilde{\boldsymbol{x}}_{i,:} - \Delta\tilde{\delta}_i))^+\|_\infty, \max_i \|((\tilde{\boldsymbol{L}}_{i,:}^\top \tilde{\boldsymbol{x}}_{i,:} - \Delta) - \tilde{v}_i)^+\|_\infty, (z^\star - \tilde{z}^\star)^+\}, \tag{14}$$

where the second inequality follows from uniqueness of $\boldsymbol{f}^{\mathrm{I}}$ in Condition A.1 and applying Lemma G.1 to the feasibility region of Problem (13) with the additional constraint $z \geq z^\star$. The argument here is similar to what we did for item-fair solution with maxmin fairness. Here, $H$ is the Hoffman constant associated with Problem (13) under instance $\boldsymbol{\theta}$, which can be characterized using Lemma G.2.

It suffices to bound the three terms on the right hand side of (14). The first two terms can be bounded using similar techniques as done in the case of maxmin fairness. Using $E = \|\boldsymbol{L} - \tilde{\boldsymbol{L}}\|_\infty$ and the fact that $\tilde{\boldsymbol{L}}_{i,:}^\top \tilde{\boldsymbol{x}}_{i,:} - \Delta \leq \tilde{v}_i \leq \tilde{\boldsymbol{L}}_{i,:}^\top \tilde{\boldsymbol{x}}_{i,:} - \Delta\tilde{\delta}_i$ for all $i \in [N]$, we have

$$\tilde{v}_i - (\tilde{\boldsymbol{L}}_{i,:}^\top \tilde{\boldsymbol{x}}_{i,:} - \Delta\tilde{\delta}_i) \leq ME \quad \text{and} \quad (\tilde{\boldsymbol{L}}_{i,:}^\top \tilde{\boldsymbol{x}}_{i,:} - \Delta) - \tilde{v}_i \leq ME$$

for all $i \in [N]$. To bound the third term in (14), we again adopt a similar technique as above and consider an auxiliary problem

$$\max_{\substack{\boldsymbol{x}\in\Delta_N,\\ z,v_i,w\in\mathbb{R}^+,\delta_i\in[0,1]}} z \quad \text{s.t.} \quad (N-1)\Delta + \sum_{i=1}^N v_i \geq z$$

$$\boldsymbol{L}_{i,:}^\top \boldsymbol{x}_{i,:} - \Delta + ME \leq v_i \leq \boldsymbol{L}_{i,:}^\top \boldsymbol{x}_{i,:} - \Delta\delta_i - ME \qquad \forall i \tag{15}$$

$$w \leq v_i \leq w + (\Gamma-\Delta)\delta_i \qquad \forall i$$

Let $(\hat{\boldsymbol{f}}^{\mathrm{I}}, \hat{z}^\star, \hat{\boldsymbol{v}}, \hat{w}, \hat{\boldsymbol{\delta}})$ denote the solution to Problem (13). Consider the feasibility region of Problem (13) under the problem instance $\tilde{\boldsymbol{\theta}}$, which contains the feasibility region of Problem (15). This then gives

$$\hat{z}^\star \leq \tilde{z}^\star. \tag{16}$$

We note that if we solve Problem (13) under problem instance, we claim that we must have $\delta_i^\star \leq 1/2$. This is because if $\delta_i^\star > 1/2$, we would have $w + (\Gamma - \Delta)\delta_i^\star \geq 2M\bar{L}\delta_i^\star > \boldsymbol{L}_{i,:}^\top \boldsymbol{f}_{i,:}^{\mathrm{I}} - \Delta\delta_i$. That is, the upper bound in the third constraint is not tight, so the upper bound in the second constraint should be tight. However, setting $\delta_i^\star = 1/2$ would yield a better objective.

Having this in mind, we choose $\zeta > 0$ sufficiently small such that $ME < \frac{1}{4}\Delta$ (this is doable since $\ell_i(\boldsymbol{\theta})$ is Lipschitz in $\boldsymbol{\theta}$. Then, Problem (15) is feasible. We note that $(\boldsymbol{f}^{\mathrm{I}}, \hat{z}, \hat{\boldsymbol{v}}, \hat{w}, \boldsymbol{\delta}^\star)$ is a feasible solution to Problem (15), where $\hat{v}_i = v_i^\star - ME$ and $\hat{w} = w - ME, \hat{z} = z^\star - NME$. This is because under the optimal solution $(\boldsymbol{f}^{\mathrm{I}}, z^\star, \boldsymbol{v}^\star, w^\star, \boldsymbol{\delta}^\star)$ to Problem (13), both upper bounds for $v_i$ should be tight (otherwise, there exists a $\boldsymbol{\delta}$ that yields a better objective). We thus have

$$\hat{z}^\star \geq \hat{z} = z^\star - NME. \tag{17}$$

Equations (16) and (17) together give
$$z^\star - \tilde{z}^\star \leq NME\,,$$
which bounds the third term in the right-hand side of (14).

Having established the bounds in the right-hand side of (14), we have shown that
$$\|\boldsymbol{f}^{\mathrm{I}} - \tilde{\boldsymbol{f}}^{\mathrm{I}}\|_\infty \leq H \cdot NME = H \cdot NM\|\boldsymbol{L} - \tilde{\boldsymbol{L}}\|_\infty$$
By statement (i) in Condition A.1, we thus establish the local Lipschitzness of item-fair solution $\boldsymbol{f}^{\mathrm{I}}$ that enforces Hooker-Williams fairness.

**Example 3: Item-Fair Solution with Kalai-Smorodinsky (K-S) Fairness.** To obtain item-fair solution $\boldsymbol{f}^{\mathrm{I}}$ under K-S fairness, we solve the following problem given problem instance $\boldsymbol{\theta}$:
$$\max_{\boldsymbol{x} \in \Delta_N^M, \beta \in [0,1]} \beta \quad \text{s.t.} \quad \frac{\boldsymbol{L}_{i,:}^\top \boldsymbol{x}_{i,:}}{L_i^\star} \geq \beta \quad \forall i \in [N]\,. \tag{18}$$
where $L_i^\star = \sum_j L_{i,j}$ denotes the maximum outcome that item $i$ can receive. Note that this is achievable if the platform always show item $i$ regardless of the type of the arriving user.

By similar arguments as in our discussion for maxmin fairness, we would get
$$\|\boldsymbol{f}^{\mathrm{U}} - \tilde{\boldsymbol{f}}^{\mathrm{U}}\|_\infty \leq H \frac{E}{\min_i L_i^\star} = \frac{H}{\min_i L_i^\star} \cdot \|\boldsymbol{L} - \tilde{\boldsymbol{L}}\|_\infty\,.$$
where $H$ is the Hoffman constant that can be characterized by invoking Lemma G.2. By statement (i) in Condition A.1, we thus have the local Lipschitzness of item-fair solution $\boldsymbol{f}^{\mathrm{I}}$ that enforces K-S fairness.

# B  Proof of Proposition 2.1

As we discussed in Section 2.3, if a platform adopts a recommendation that solely benefits a single stakeholder group, this could result in extensive costs for the rest of the stakeholders, including the platform itself. The following Example B.1 establishes Proposition 2.1, where we consider a problem instance with one highly popular item and another notably profitable item. Under such a problem instance, the perceptions of a "fair solution" can vary widely among different stakeholders.

**Example B.1 (A single-sided solution can be extremely unfair to the other sides.)** *Consider a problem instance with $N$ items and $M$ types of users. For $i \in [M], j \in [N]$, let the probability of purchases $y_{1,j} = 1$ and $y_{i,j} = \epsilon_1$ where $\epsilon_1 \ll 1$ for all $i \neq 1$; probability of arrival $p_j = 1/M$; revenue $r_2 = 1/\epsilon_1^2$ and $r_i = 1$, for all $i \neq 2$. For simplicity, let the utility of the users $U_{i,j} = y_{i,j}$. Let $\epsilon = \max\{\epsilon_1, 1/N\}$. Under such a instance, we have the following.*

***Platform's revenue-maximizing solution.*** *A platform that seeks to maximize its revenue would always recommend item $2$ to any arriving user, which yields expected revenue of $1/\epsilon_1 \gg 1$. However, such a solution is extremely unfair for any items $i \neq 2$ that receive zero outcome. This is also unfair to all users, as they would receive utility $1$ from item $1$ but only receives utility $\epsilon_1$ from item $2$.*

***An item-fair solution.*** *Suppose items consider visibility as their outcome and adopts maxmin fairness, offering all items equal visibility $x_{i,j} = 1/N$ is an item-fair solution. However, this can be unfair for all users who prefers item $1$, whose utility under the item-fair solution is $1/N + (N-1)/N\epsilon_1 < 2\epsilon$. This is also unfair to the platform that prefers item $2$, as its current revenue becomes $1/N \cdot 1/\epsilon_1 + 1/N + (N-2)/N \cdot \epsilon_1 < 1/N \cdot 1/\epsilon_1 + 1 = 1/N \cdot 1/\epsilon_1 + \epsilon_1 \cdot 1/\epsilon_1 \leq 2\epsilon \cdot 1/\epsilon_1$.*

***A user-fair solution.*** *A user-fair solution would always display item $1$ to all types of users. Nonetheless, such a solution is extremely unfair to the rest of the items that receives zero outcome. It is also unfair to the platform, as always displaying item $1$ results in expected revenue of $1$, while if it displays item $2$ it would gain $1/\epsilon_1$.*

# C  Price of Fairness

In this section, we formally characterize the conflicting interests among different stakeholders using a concept called the *price of fairness*, and show that our formulation of Problem (FAIR) provides the platform with flexible handles to find the right middleground.

The concept of price of fairness was first introduced by [9], which we formally define as follows.

**Definition C.1 (Price of Fairness)** *Given a problem instance $\boldsymbol{\theta} \in \Theta$, an item-fair solution $\boldsymbol{f}^I$ and a user-fair solution $\boldsymbol{f}^U$. We let* OPT-REV $= \max_{\boldsymbol{x}}$ REV$(\boldsymbol{x})$ *be the maximum achievable expected revenue in the absence of fairness, and let* FAIR-REV *be the solution to Problem* (FAIR)*. The price of fairness (PoF) is defined as*

$$\text{PoF} = \frac{\text{OPT-REV} - \text{FAIR-REV}}{\text{OPT-REV}} .$$

The price of fairness, which depends on the problem instance $\boldsymbol{\theta}$ and $(\delta^I, \delta^U)$, quantifies the trade-off between item/user fairness and platform interests, where a lower value is favored by the platform. In Theorem C.1, we formally upper bound the price of fairness introduced by Problem (FAIR) when we interpolate item and user fairness with parameters $\delta^I, \delta^U$.

**Theorem C.1** *Given a problem instance $\boldsymbol{\theta} \in \Theta$, item-fair solution $\boldsymbol{f}^I$ and user-fair solution $\boldsymbol{f}^U$. Let $\boldsymbol{x}^{\text{OPT}} = \arg\max_{\boldsymbol{x}}$ REV$(\boldsymbol{x})$ be the revenue-maximizing solution in the absence of fairness. If we solve Problem* (FAIR) *with parameters $\delta^I$ and $\delta^U$ and assuming that the problem is feasible, the price of fairness is at most*

$$\text{PoF} \leq H \cdot \max \left\{ \max_{i \in [N]} (\delta^I \cdot O_i^I(\boldsymbol{f}^I) - O_i^I(\boldsymbol{x}^{\text{OPT}}))^+, \max_{j \in [M]} (\delta^U \cdot O_j^U(\boldsymbol{f}^U) - O_j^U(\boldsymbol{x}^{\text{OPT}}))^+ \right\},$$

*where constant $H$ is the Hoffman constant associated with Problem* (FAIR) *under instance $\boldsymbol{\theta}$ (see definition in Lemma G.2).*

There are two important takeaways from Theorem C.1: (1) The price of fairness arises from the misalignment of objectives, captured by the difference in items'/users' outcomes under the single-sided item/user-fair solution and the platform's optimal revenue solution, $\boldsymbol{x}^{\text{OPT}}$ (i.e., $(\delta^I \cdot O_i^I(\boldsymbol{f}^I) - O_i^I(\boldsymbol{x}^{\text{OPT}}))^+$ and $(\delta^U \cdot O_j^U(\boldsymbol{f}^U) - O_j^U(\boldsymbol{x}^{\text{OPT}}))^+$. A high price of fairness can result from high divergence of platform's goals from item/user interests (as illustrated by Proposition 2.1 and Example B.1). (2) Theorem C.1 also underscores the value of the parameters $\delta^I$ and $\delta^U$ in achieving a balance among stakeholder interests. Both takeaways are further supported empirically, as we next investigate the price of fairness associated with our case study on Amazon review data in Section C.1.

### C.0.1   Proof for Theorem C.1

Recall from Section 2.4 that $i_j^\star = \arg\max_{i \in [N]} r_i y_{i,j}$ is the item with the maximum expected revenue, if an arriving user is of type $j$. (Here, we assumed that $i_j^\star$ is unique without loss of generality.) The platform's revenue-maximizing solution is $x_{i,j}^{\text{OPT}} = 1$ if $i = i_j^\star$ and $x_{i,j}^{\text{OPT}} = 0$ otherwise.

Now, suppose that the platform solves Problem (FAIR) with some fairness parameters $\delta^I, \delta^U \in [0, 1]$ and assuming Problem (FAIR) is feasible. Let

$$\mathcal{F} = \{\boldsymbol{x} \in \Delta_N^M : O_i^I(\boldsymbol{x}) \geq \delta^I \cdot O_i^I(\boldsymbol{f}^I); \quad O_j^U(\boldsymbol{x}) \geq \delta^U \cdot O_j^U(\boldsymbol{f}^U) \quad \forall i \in [N], j \in [M]\}$$

denote the feasibility region of Problem (FAIR). We define

$$\boldsymbol{x}' = \arg\min_{\boldsymbol{x} \in \mathcal{F}} \|\boldsymbol{x} - \boldsymbol{x}^{\text{OPT}}\|_\infty . \tag{19}$$

That is, $\boldsymbol{x}'$ is a feasible solution to Problem (FAIR) that is closest to $\boldsymbol{x}^{\text{OPT}}$ w.r.t. $\|\cdot\|_\infty$. Then, by Lemma G.1, we have the following bound

$$\|\boldsymbol{x}' - \boldsymbol{x}^{\text{OPT}}\|_\infty \leq H \cdot \max \left\{ \max_{i \in [N]} (\delta^I \cdot \boldsymbol{L}_{i,:}^\top \boldsymbol{f}_{i,:}^I - \boldsymbol{L}_{i,:}^\top \boldsymbol{x}_{i,:}^{\text{OPT}})^+, \max_{j \in [M]} (\delta^U \cdot \boldsymbol{U}_{:,j}^\top \boldsymbol{f}_{:,j}^U - \boldsymbol{U}_{:,j}^\top \boldsymbol{x}_{:,j}^{\text{OPT}})^+, 0 \right\}, \tag{20}$$

where the first term $H$ is the Hoffman constant associated with the feasibility region $\mathcal{F}$, and the second term encapsulates how much $\boldsymbol{x}^{\text{OPT}}$ violates the item/user-fairness constraints.

Now, note that we also have

$$\|\boldsymbol{x}' - \boldsymbol{x}^{\text{OPT}}\|_\infty \overset{(a)}{=} \max_{j \in [M]} \max\{1 - x'_{i_j^\star, j}, \max_{i \neq i_j^\star} x'_{i,j}\} \overset{(b)}{=} \max_{j \in [M]} 1 - x'_{i_j^\star, j} \tag{21}$$

where (a) follows from the form of $\boldsymbol{x}^{\text{OPT}}$, and (b) follows from $1 - x'_{i^\star_j, j} = \sum_{i \neq i^\star_j, j} x'_{i,j} \geq x'_{i,j}$ for any $i \neq i^\star$. This then allows us to bound the PoF as follows

$$\text{PoF} = \frac{\text{OPT-REV} - \text{FAIR-REV}}{\text{OPT-REV}} \leq 1 - \frac{\text{REV}(\boldsymbol{x}')}{\text{OPT-REV}} \leq 1 - \frac{\sum_{j \in [M]} R_{i^\star_j} x'_{i^\star_j, j}}{\sum_{j \in [M]} R_{i^\star_j}} \leq \max_{j \in [M]} 1 - x'_{i^\star_j, j} = \|\boldsymbol{x}' - \boldsymbol{x}^{\text{OPT}}\|_\infty \,,$$

where the first inequality follows from $\boldsymbol{x}'$ being a feasible solution to Problem (FAIR), the second inequality follows from $\text{REV}(\boldsymbol{x}') \geq \sum_{j \in [M]} R_{i^\star_j} x'_{i^\star_j, j}$, and $\text{OPT-REV} = \sum_{j \in [M]} R_{i^\star_j}$, and the final equality follows from (21).

Finally, combining Eq. (20) and Eq. (22) finishes the proof. ∎

## C.1 Price of Fairness for Our Case Study on Amazon Review Data

In this section, we investigate the price of fairness from our case study on the Amazon review data (Section 4) and shed light on how our fair recommendation framework, Problem (FAIR), can help the platform achieve the right middleground among multiple stakeholders.

In our case study, we act as an e-commerce site (such as Amazon) recommending a collection of at most $K = 3$ products to incoming users. There are a total of $N = 30$ items to be shown to $M = 5$ types of users, where the relevance score between items and users as well as users' arrival rates are both obtained from an Amazon review dataset [71]; see Section 4 for a complete description of the dataset. Assuming that the platform has full knowledge of the problem instance, we solve Problem (FAIR-ASSORT) (here, we consider the assortment extension of Problem (FAIR); see Section E.2 for details) under different values of fairness parameters $\delta^{\text{I}}, \delta^{\text{U}}$ to investigate how much loss the platform needs to endure in order to achieve various levels of fairness for its items/users.

The left-hand side of Figure 2 shows the price of fairness (PoF) endured by the platform given different $(\delta^{\text{I}}, \delta^{\text{U}})$. Note that in our recommendation problem based on Amazon review data, item fairness is much more difficult to achieve than user fairness, since (i) the number of items is much larger, making the fair outcome of many items differentiate a lot from the outcome attained under platform's revenue-maximizing solution; and (ii) the objective of users (MNL utility) aligns fairly well with the platform's objective (revenue). As a result, the item-fair constraints are the binding constraints in majority of the cases. This can be seen in the left-hand side plot, where we see that PoF increases more significantly when we increase $\delta^{\text{I}}$. This concurs with the first takeaway from Theorem C.1, which suggests that the misalignment of objectives directly impacts the PoF.

In the right-hand side of Figure 2, we plot the evolution of PoF when we fix $\delta^{\text{U}} = 0$ (upper-right plot) and $\delta^{\text{I}} = 0$ (lower-right plot) respectively. Given that the item fairness constraints are primarily the binding constraints, as discussed above, we observe a linear increase in PoF for the platform as the fairness parameter $\delta^{\text{I}}$ for items increases. This observation again corroborates the findings presented in Theorem C.1. For the users, in the case of Amazon review data, the misalignment of objectives $(\delta^{\text{U}} \cdot O^{\text{U}}_j(\boldsymbol{f}^{\text{U}}) - O^{\text{U}}_j(\boldsymbol{x}^{\text{OPT}}))^+$ only becomes noteworthy as $\delta^{\text{U}}$ gets close to 1. This indicates that under the Amazon review data, the platform can potentially obtain a high level of user-fairness at very little cost, which also matches our findings in our case study in Section 4.

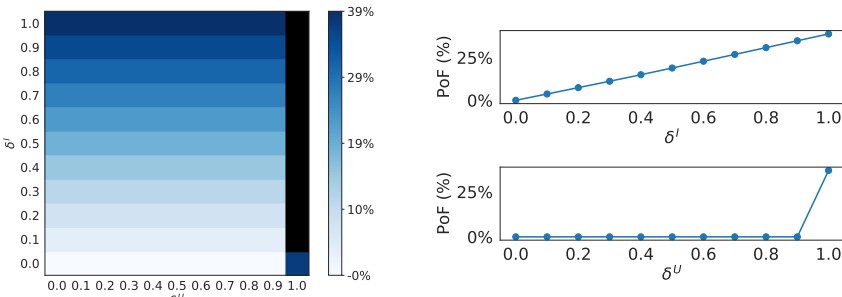

Figure 2: Price of Fairness (PoF) in our case study on the Amazon review data. Left: PoF when solving Problem (FAIR-ASSORT) under different fairness parameters $(\delta^{\text{I}}, \delta^{\text{U}})$. The grid is colored black if the problem is infeasible. Upper-right: PoF when $\delta^{\text{U}} = 0$. Lower-right: PoF when $\delta^{\text{I}} = 0$.

**Insights from the case study.** Our empirical analysis provides practical guidelines for selecting appropriate fairness parameters $\delta^{\text{I}}, \delta^{\text{U}}$ in real-world settings. Specifically, platforms should focus on (i) identifying which fairness constraints are binding and (ii) evaluating the degree of objective misalignment, typically captured by the slope between the PoF and the fairness parameters.

One potential approach is to first identify the binding constraints by assessing which stakeholders experience the most unfair outcomes under the current recommendation policy. Then, using the piecewise linear relationship established in Theorem C.1, the platform can estimate the tradeoff between the binding constraints and PoF. Based on the desired fairness levels and acceptable PoF, the platform can now narrow down the range of fairness parameters to experiment with. Once a small subset of promising fairness parameters is identified, a platform can conduct online A/B tests by splitting its traffic to experiment with these parameters in parallel, thus selecting the optimal fairness parameters efficiently without extensive trial-and-error.

# D  Proof for Theorem 3.1

## D.1  Definitions and Lemmas

We first state a few definitions and lemmas that are useful for the proof of Theorem 3.1.

**Good event and cutoff time.** We define the *good event* at round $t$, denoted by $\mathcal{E}_t$, as the event under which our estimates for $\boldsymbol{y}$ and $\boldsymbol{p}$ are accurate w.r.t. $t$.

**Definition D.1 (Good event)** *At round $t$, a good event $\mathcal{E}_t$ is*

$$\mathcal{E}_t = \{\|\hat{\boldsymbol{p}}_t - \boldsymbol{p}\|_1 \leq \Gamma_{p,t}\} \cap \{\|\hat{\boldsymbol{y}}_t - \boldsymbol{y}\|_\infty \leq \Gamma_{y,t}\}, \tag{22}$$

*where*

$$\Gamma_{y,t} = 2\log(T)/\sqrt{m(t)\epsilon_t} \text{ and } m(t) = \max\left\{1, t/\log(T) - \sqrt{t\log(T)/2}\right\} \tag{23}$$

$$\Gamma_{p,t} = 5\sqrt{\log(3T)/t}.$$

Observe that $\Gamma_{p,t}$ strictly decreases to 0 as $t \to \infty$. This is also the case for $\Gamma_{y,t}$. To see that, note that $m(t)$, used to define $\Gamma_{y,t}$, strictly increases in $t$ and goes to $\infty$ as $t \to \infty$. Hence, we can additionally define the *cutoff time* $\mathcal{T}$, which is time after which the good event becomes "sufficiently good":

$$\mathcal{T} = \min\left\{t \in [T] : \Gamma_{y,t} < 1 - \|\boldsymbol{y}\|_\infty, \ \max\{\Gamma_{y,t}, \Gamma_{p,t}\} \leq \zeta, \text{and } m(t) > 1\right\} = \mathcal{O}(1). \tag{24}$$

where $\zeta$ is defined in Assumption 3.1. Here we also used $y_{i,j} \in (0,1)$ for all $i \in [N], j \in [M]$, which is assumed in Section 2.1, so we must have $\|\boldsymbol{y}\|_\infty < 1$. Here, note that $\Gamma_{y,t} = \mathcal{O}(t^{-\frac{1}{3}})$, $\Gamma_{p,t} = \mathcal{O}(t^{-\frac{1}{2}})$ so $\mathcal{T} = \mathcal{O}(\max\{\zeta, \|\boldsymbol{y}\|_\infty\}^3)$ and is $\mathcal{O}(1)$ w.r.t. $T$.

The following lemma states that under the good event, for $t > 0$ that is sufficiently large, the optimal solution $\boldsymbol{x}^\star$ to Problem (FAIR) is feasible to the relaxed problem under the estimated instance, Problem (FAIR-RELAX($\hat{\boldsymbol{\theta}}_t, \eta_t$)), while our solution $\hat{\boldsymbol{x}}_t$ is also feasible to the relaxed problem under the ground-truth instance, Problem (FAIR-RELAX($\boldsymbol{\theta}, \eta_t$)). The proof is presented in Section D.3.

**Lemma D.1 (Feasibility under the good event)** *Given problem instance $\boldsymbol{\theta} \in \Theta$. Suppose that Assumption 3.1 holds, where $B$ is the Lipschitz constant. Let constant $\bar{U}$ be the maximum utility in the small neighborhood around $\boldsymbol{\theta}$; that is, $\|\boldsymbol{U}(\tilde{\boldsymbol{\theta}})\|_\infty \leq \bar{U}$ for all $\tilde{\boldsymbol{\theta}} \in \Theta$ such that $\|\boldsymbol{\theta} - \tilde{\boldsymbol{\theta}}\| \leq \zeta$. Under the good event $\mathcal{E}_t$, when $t > \mathcal{T}$ (defined in (24)) and $\log(T) > (2 + \|\boldsymbol{r}\|_\infty)B$, we have the following:*

*(i) $\|\hat{\boldsymbol{y}}_t\|_\infty < 1$, so $\boldsymbol{x}_{t,i,j} = (1 - N\epsilon_t)\hat{\boldsymbol{x}}_{t,i,j} + \epsilon_t$ according to* FORM.
*(ii) $\boldsymbol{x}^*$ is feasible to Problem (FAIR-RELAX($\hat{\boldsymbol{\theta}}_t, \eta_t$)).*
*(iii) $\hat{\boldsymbol{x}}_t$ is feasible to Problem (FAIR-RELAX($\boldsymbol{\theta}, 2\eta_t$))*

*where $\hat{\boldsymbol{\theta}}_t$ and $\eta_t$ are defined in Eq. (1) and Eq. (2).*

The second Lemma (Lemma D.2) shows that the good event happens with high probability. Its proof is deferred to Section D.4.

**Lemma D.2 (Concentration bounds for estimates)** *Assume $\log(T) > 1/\min_{j \in [M]} p_j$. Then for any $t > \mathcal{T}$ defined in Eq. (24), we have $\mathbb{P}(\mathcal{E}_{t+1}^c) \leq (MN + 2)/T$.*

## D.2 Complete Statement and Proof of Theorem 3.1

**Theorem D.3 (Complete statement of Theorem 3.1)** *Given problem instance $\boldsymbol{\theta} \in \Theta$ and assume that Assumption 3.1 holds. Then, for $\log(T) > \max\left\{(2 + \|\boldsymbol{r}\|_\infty)B, \ \frac{1}{\min_{j \in [M]} p_j}\right\}$, we have*

- *the revenue regret of* FORM *is at most* $\mathbb{E}[\mathcal{R}(T)] \leq \mathcal{O}(MN^{\frac{1}{3}}T^{-\frac{1}{3}})$
- *the fairness regret of* FORM *is at most* $\mathbb{E}[\mathcal{R}_F(T)] \leq \mathcal{O}(MN^{\frac{1}{3}}T^{-\frac{1}{3}})$.

*Proof of Theorem D.3.* We bound the revenue regret and the fairness regret respectively.

### (1) Bounding revenue regret.

For a given $t > \mathcal{T}$, assume that the good event $\mathcal{E}_t$ defined in Definition D.1 holds. Let $\boldsymbol{R} = (R_{i,j})_{\substack{i \in [N] \\ j \in [M]}}$, where $R_{i,j} = r_i p_j y_{i,j}$ denotes the expected revenue that the platform receives from offering item $i$ to user of type-$j$. That is, $\text{REV}(\boldsymbol{x}, \boldsymbol{\theta}) = \sum_{\substack{i \in [N] \\ j \in [M]}} R_{i,j} x_{i,j}$. Similarly, for simplicity of notation, let $\hat{\boldsymbol{R}}_t = (\hat{R}_{t,i,j})_{\substack{i \in [N] \\ j \in [M]}}$, where $\hat{R}_{t,i,j} = r_i \hat{p}_{t,j} \hat{y}_{t,i,j}$.

Let us first rewrite the revenue regret at round $t$ as the following:

$$
\text{REV}(\boldsymbol{x}^\star, \boldsymbol{\theta}) - \text{REV}(\boldsymbol{x}_t, \boldsymbol{\theta}) = \sum_{\substack{i \in [N] \\ j \in [M]}} R_{i,j} x_{i,j}^\star - \sum_{\substack{i \in [N] \\ j \in [M]}} R_{i,j} x_{t,i,j}
$$

$$
= \sum_{\substack{i \in [N] \\ j \in [M]}} \hat{R}_{t,i,j}(x_{i,j}^\star - x_{t,i,j}) + \sum_{\substack{i \in [N] \\ j \in [M]}} x_{i,j}^\star(R_{i,j} - \hat{R}_{t,i,j}) - \sum_{\substack{i \in [N] \\ j \in [M]}} x_{t,i,j}(R_{i,j} - \hat{R}_{t,i,j})
$$

$$
= \sum_{\substack{i \in [N] \\ j \in [M]}} \hat{R}_{t,i,j}(x_{i,j}^\star - x_{t,i,j}) + \sum_{\substack{i \in [N] \\ j \in [M]}} (x_{i,j}^\star - x_{t,i,j})(R_{i,j} - \hat{R}_{t,i,j})
$$

We then have

$$
\text{REV}(\boldsymbol{x}^\star, \boldsymbol{\theta}) - \text{REV}(\boldsymbol{x}_t, \boldsymbol{\theta})
$$

$$
\overset{(a)}{=} (1 - N\epsilon_t) \sum_{\substack{i \in [N] \\ j \in [M]}} \hat{R}_{t,i,j}(x_{i,j}^\star - \hat{x}_{t,i,j}) + N\epsilon_t \sum_{\substack{i \in [N] \\ j \in [M]}} \hat{R}_{t,i,j} x_{i,j}^\star - \epsilon_t \sum_{\substack{i \in [N] \\ j \in [M]}} \hat{R}_{t,i,j} + \sum_{\substack{i \in [N] \\ j \in [M]}} (x_{i,j}^\star - x_{t,i,j})(R_{i,j} - \hat{R}_{t,i,j})
$$

$$
\overset{(b)}{\leq} N\epsilon_t \sum_{\substack{i \in [N] \\ j \in [M]}} \hat{R}_{t,i,j} x_{i,j}^\star - \epsilon_t \sum_{\substack{i \in [N] \\ j \in [M]}} \hat{R}_{t,i,j} + \sum_{\substack{i \in [N] \\ j \in [M]}} (x_{i,j}^\star - x_{t,i,j})(R_{i,j} - \hat{R}_{t,i,j})
$$

$$
\leq NM\epsilon_t \|\hat{\boldsymbol{R}}_t\|_\infty + 2M \cdot \|\boldsymbol{R} - \hat{\boldsymbol{R}}_t\|_\infty
$$

$$
\leq NM\epsilon_t \|\boldsymbol{R} - \hat{\boldsymbol{R}}_t\|_\infty + NM\epsilon_t \|\boldsymbol{R}\|_\infty + 2M \cdot \|\boldsymbol{R} - \hat{\boldsymbol{R}}_t\|_\infty
$$

$$
\overset{(c)}{\leq} 3M\|\boldsymbol{R} - \hat{\boldsymbol{R}}_t\|_\infty + NM\epsilon_t \|\boldsymbol{R}\|_\infty
$$

$$
(25)
$$

where (a) follows from statement (i) in Lemma D.1, i.e., $\boldsymbol{x}_{t,i,j} = (1 - N\epsilon_t)\hat{\boldsymbol{x}}_{t,i,j} + \epsilon_t$; (b) follows from the fact that $\epsilon_t \leq \frac{1}{N}$ so $1 - N\epsilon_t \geq 0$, and by statement (ii) Lemma D.1, we have that $\boldsymbol{x}^\star$ and $\hat{\boldsymbol{x}}_t$ are both feasible to Problem (FAIR-RELAX($\hat{\boldsymbol{\theta}}_t, \eta_t$)), and hence $\text{REV}(\boldsymbol{x}^\star, \hat{\boldsymbol{\theta}}_t) \leq \text{REV}(\hat{\boldsymbol{x}}_t, \hat{\boldsymbol{\theta}}_t)$; (c) follows from the fact that $\epsilon_t \leq \frac{1}{N}$.

Note that since

$$
\begin{aligned}
\hat{R}_{t,i,j} - R_{i,j} &= r_i \hat{y}_{t,i,j} \hat{p}_{t,j} - r_i y_{i,j} p_j \\
&= r_i (\hat{y}_{t,i,j} \hat{p}_{t,j} - y_{i,j} \hat{p}_{t,j} + y_{i,j} \hat{p}_{t,j} - y_{i,j} p_j) \\
&= r_i \Big( (\hat{y}_{t,i,j} - y_{i,j})\hat{p}_j + y_{i,j}(\hat{p}_{t,j} - p_j) \Big) \\
&\leq r_i \Big( \|\hat{\boldsymbol{y}}_t - \boldsymbol{y}\|_\infty + \|\boldsymbol{y}\|_\infty \cdot \|\hat{\boldsymbol{p}}_t - \boldsymbol{p}\|_1 \Big) \\
&\leq r_i \Big( \Gamma_{y,t} + \Gamma_{p,t} \Big).
\end{aligned}
\tag{26}
$$

We thus have

$$
\text{REV}(\boldsymbol{x}^\star, \boldsymbol{\theta}) - \text{REV}(\boldsymbol{x}_t, \boldsymbol{\theta}) \leq 3M\bar{r}(\Gamma_{y,t} + \Gamma_{p,t}) + NM\epsilon_t \|\boldsymbol{R}\|_\infty.
\tag{27}
$$

Hence, for any $t > \mathcal{T}$, let $\mathcal{F}_{t-1}$ be the sigma-algbra generated from all randomness up to round $t - 1$, and

$$\mathbb{E}\left[\text{REV}(\boldsymbol{x}^\star, \boldsymbol{\theta}) - \text{REV}(\boldsymbol{x}_t, \boldsymbol{\theta})\right]$$

$$= \mathbb{E}\left[\mathbb{E}\left[\text{REV}(\boldsymbol{x}^\star, \boldsymbol{\theta}) - \text{REV}(\boldsymbol{x}_t, \boldsymbol{\theta}) \mid \mathcal{F}_{t-1}\right]\right]$$

$$\leq \mathbb{E}\left[\mathbb{E}\left[(\text{REV}(\boldsymbol{x}^\star, \boldsymbol{\theta}) - \text{REV}(\boldsymbol{x}_t, \boldsymbol{\theta}))\mathbb{I}\{\mathcal{E}_t\} \mid \mathcal{F}_{t-1}\right]\right] + \mathbb{E}\left[\text{REV}(\boldsymbol{x}^\star, \boldsymbol{\theta})\mathbb{I}\{\mathcal{E}_t^c\}\right] \tag{28}$$

$$\overset{(a)}{\leq} 3M\bar{r}\left(\Gamma_{y,t} + \Gamma_{p,t}\right) + NM\epsilon_t\|\boldsymbol{R}\|_\infty + \|\boldsymbol{R}\|_\infty\mathbb{P}(\mathcal{E}_t^c)$$

$$\overset{(b)}{\leq} 3M\bar{r}\left(\Gamma_{y,t} + \Gamma_{p,t}\right) + NM\epsilon_t\|\boldsymbol{R}\|_\infty + \frac{(MN+2)\|\boldsymbol{R}\|_\infty}{T}$$

where (a) follows from (27); (b) follows from the high probability bound for event $\mathcal{E}_t$ in Lemma D.2, given that $t > \mathcal{T}$ holds.

Finally, by (28), we have

$$\frac{1}{T}\sum_{t\in[T]}\mathbb{E}\left[\text{REV}(\boldsymbol{x}^\star, \boldsymbol{\theta}) - \text{REV}(\boldsymbol{x}_t, \boldsymbol{\theta})\right] \leq \mathcal{O}\left(\frac{\mathcal{T}}{T} + \frac{M}{T}\sum_{t>\mathcal{T}}(\Gamma_{p,t} + \Gamma_{y,t}) + \frac{NM}{T}\sum_{t>\mathcal{T}}\epsilon_t\right) = \mathcal{O}(MN^{\frac{1}{3}}T^{-\frac{1}{3}}) \tag{29}$$

where in the final equality we recall $\epsilon_t = \min\left\{N^{-1}, N^{-\frac{2}{3}}t^{-\frac{1}{3}}\right\}$ and $\Gamma_{y,t} = 2\log(T)/\sqrt{m(t)\epsilon_t} < \frac{2\log(T)}{\sqrt{(t/\log(T) - \sqrt{t\log(T)/2})\epsilon_t}} = \mathcal{O}(N^{\frac{1}{3}}t^{-\frac{1}{3}})$ since $m(t) > 1$ for all $t > \mathcal{T}$ defined in Eq. (24).

**(2) Bounding constraint violation.**

**Item-Fair Constraints.** For any item $i \in [N]$, and any $t > \mathcal{T}$, under the good event $\mathcal{E}_t$ defined in Definition D.1, we have

$$\delta^{\text{I}} \cdot \boldsymbol{L}_{i,:}(\boldsymbol{\theta})^\top \boldsymbol{f}_{i,:}^{\text{I}}(\boldsymbol{\theta}) - \boldsymbol{L}_{i,:}(\boldsymbol{\theta})^\top \boldsymbol{x}_{t,i,:} \overset{(a)}{=} \delta^{\text{I}} \cdot \boldsymbol{L}_{i,:}(\boldsymbol{\theta})^\top \boldsymbol{f}_{i,:}^{\text{I}}(\boldsymbol{\theta}) - \boldsymbol{L}_{i,:}(\boldsymbol{\theta})^\top \left((1 - N\epsilon_t)\hat{\boldsymbol{x}}_{t,i,:} + \epsilon_t\boldsymbol{e}_M\right)$$

$$\leq \delta^{\text{I}} \cdot \boldsymbol{L}_{i,:}(\boldsymbol{\theta})^\top \boldsymbol{f}_{i,:}^{\text{I}}(\boldsymbol{\theta}) - \boldsymbol{L}_{i,:}(\boldsymbol{\theta})^\top \hat{\boldsymbol{x}}_{t,i,:} + NM\|\boldsymbol{L}_{i,:}(\boldsymbol{\theta})\|_\infty\epsilon_t$$

$$\overset{(b)}{\leq} 2M\log(T)\max\{\Gamma_{p,t}, \Gamma_{y,t}\} + NM\|\boldsymbol{L}_{i,:}(\boldsymbol{\theta})\|_\infty\epsilon_t \tag{30}$$

where (a) follows from Lemma D.1 part (i), which states that under the good event $\mathcal{E}_t$ for $t > \mathcal{T}$, we have $\boldsymbol{x}_{t,i,:} = (1 - N\epsilon_t)\hat{\boldsymbol{x}}_{t,i,:} + \epsilon_t\boldsymbol{e}_M$ according to Algorithm 1; (b) follows from Lemma D.1 part (iii), which states $\hat{\boldsymbol{x}}_t$ is feasible to Problem (FAIR-RELAX($\boldsymbol{\theta}, \eta_t$)), where $\eta_t = \log(T)\max\{\Gamma_{p,t}, \Gamma_{y,t}\}$ according to Eq. (1).

Hence, we have

$$\frac{1}{T}\sum_{t=1}^{T}\mathbb{E}\left[(\delta^{\text{I}} \cdot \boldsymbol{L}_{i,:}(\boldsymbol{\theta})^\top \boldsymbol{f}_{i,:}^{\text{I}}(\boldsymbol{\theta}) - \boldsymbol{L}_{i,:}(\boldsymbol{\theta})^\top \boldsymbol{x}_{t,i,:})^+\right]$$

$$\overset{(a)}{\leq} \frac{\mathcal{T}}{T}\max\{1, \bar{r}\} + \frac{1}{T}\sum_{t>\mathcal{T}}\mathbb{E}\left[(\delta^{\text{I}} \cdot \boldsymbol{L}_{i,:}(\boldsymbol{\theta})^\top \boldsymbol{f}_{i,:}^{\text{I}}(\boldsymbol{\theta}) - \boldsymbol{L}_{i,:}(\boldsymbol{\theta})^\top \boldsymbol{x}_{t,i,:})^+\right]$$

$$\overset{(b)}{\leq} \frac{\mathcal{T}}{T}\max\{1, \bar{r}\} + \frac{1}{T}\sum_{t>\mathcal{T}}\mathbb{E}\left[\left(\delta^{\text{I}} \cdot \boldsymbol{L}_{i,:}(\boldsymbol{\theta})^\top \boldsymbol{f}_{i,:}^{\text{I}}(\boldsymbol{\theta}) - \boldsymbol{L}_{i,:}(\boldsymbol{\theta})^\top \boldsymbol{x}_{t,i,:}\right)^+\mathbb{I}\{\mathcal{E}_t\}\right] + \mathbb{P}(\mathcal{E}_t^c)\max\{1, \bar{r}\}$$

$$\overset{(c)}{\leq} \frac{\mathcal{T}}{T}\max\{1, \bar{r}\} + \frac{1}{T}\sum_{t>\mathcal{T}}[2M\log(T)\max\{\Gamma_{p,t}, \Gamma_{y,t}\} + NM\|\boldsymbol{L}(\boldsymbol{\theta})\|_\infty\epsilon_t] + \mathbb{P}(\mathcal{E}_t^c)\max\{1, \bar{r}\}$$

$$\overset{(d)}{\leq} \frac{\mathcal{T}}{T}\max\{1, \bar{r}\} + \frac{1}{T}\sum_{t>\mathcal{T}}[2M\log(T)\max\{\Gamma_{p,t}, \Gamma_{y,t}\} + NM\|\boldsymbol{L}(\boldsymbol{\theta})\|_\infty\epsilon_t] + \frac{(NM+2)\max\{1, \bar{r}\}}{T}$$

$$= \mathcal{O}(MN^{\frac{1}{3}}T^{-\frac{1}{3}}) \tag{31}$$

where (a) and (b) follows from $\delta^{\mathtt{I}} \leq 1$ and $\boldsymbol{L}_{i,:}(\boldsymbol{\theta})^{\top}\boldsymbol{f}_{i,:}^{\mathtt{I}}(\boldsymbol{\theta}) \leq \max\{1, \bar{r}\}$ depending on the item's outcome; (c) follows from Eq. (30); (d) follows from Lemma D.2 for sufficiently large $T$. Therefore, the time-averaged item-fairness constraint violation for any item $i$ is at most $\mathcal{O}(MN^{\frac{1}{3}}T^{-\frac{1}{3}})$.

We can perform the same analysis for user-fairness constraints and obtain the same upper bound. The proof is thus omitted.

### D.3 Proof of Lemma D.1

We prove the three statements in Lemma D.1 respectively as follows.

**Proof for (i).** Consider the following under the good event $\mathcal{E}_t$, defined in Eq. (D.1):

$$\|\hat{\boldsymbol{y}}_t\|_\infty - \|\boldsymbol{y}\|_\infty \leq \|\hat{\boldsymbol{y}}_t - \boldsymbol{y}\|_\infty \leq \Gamma_{y,t} \overset{(a)}{<} 1 - \|\boldsymbol{y}\|_\infty, \tag{32}$$

where (a) follows from $t > \mathcal{T}$ and the definition of $\mathcal{T}$ in (24). This gives $\|\hat{\boldsymbol{y}}_t\|_\infty < 1$, and hence given the design of FORM, $\boldsymbol{x}_{t,i,j} = (1 - N\epsilon_t)\hat{\boldsymbol{x}}_{t,i,j} + \epsilon_t$.

**Proof for (ii).** Here, we would like to show that $\boldsymbol{x}^*$ is feasible for Problem (FAIR-RELAX($\hat{\boldsymbol{\theta}}_t, \eta_t$)), where $\hat{\boldsymbol{\theta}}_t$ and $\eta_t$ are defined in Eq. (1) and Eq. (2).

**Item-Fair Constraints.** Let $\bar{L} = \|\boldsymbol{L}\|_\infty$. For any $i \in [N]$ and $\boldsymbol{x} \in \Delta_N^M$, under Assumption 3.1, we have

$$\left|\boldsymbol{L}_{i,:}(\hat{\boldsymbol{\theta}}_t)^{\top}\boldsymbol{x}_{i,:} - \boldsymbol{L}_{i,:}(\boldsymbol{\theta})^{\top}\boldsymbol{x}_{i,:}\right| \leq \|\boldsymbol{L}_{i,:}(\hat{\boldsymbol{\theta}}_t) - \boldsymbol{L}_{i,:}(\boldsymbol{\theta})\|_\infty \|\boldsymbol{x}_{i,:}\|_1 \leq MB \cdot \|\hat{\boldsymbol{\theta}}_t - \boldsymbol{\theta}\|_\infty = MB \cdot \max\{\Gamma_{p,t}, \Gamma_{y,t}\}. \tag{33}$$

On the other hand, for any $i \in [N]$, we also have

$$\begin{aligned}
&\left|\boldsymbol{L}_{i,:}(\hat{\boldsymbol{\theta}}_t)^{\top}\boldsymbol{f}_{i,:}^{\mathtt{I}}(\hat{\boldsymbol{\theta}}_t) - \boldsymbol{L}_{i,:}(\boldsymbol{\theta})^{\top}\boldsymbol{f}_{i,:}^{\mathtt{I}}(\boldsymbol{\theta})\right| \\
&= \left|\boldsymbol{L}_{i,:}(\hat{\boldsymbol{\theta}}_t)^{\top}\boldsymbol{f}_{i,:}^{\mathtt{I}}(\hat{\boldsymbol{\theta}}_t) - \boldsymbol{L}_{i,:}(\hat{\boldsymbol{\theta}}_t)^{\top}\boldsymbol{f}_{i,:}^{\mathtt{I}}(\boldsymbol{\theta}) + \boldsymbol{L}_{i,:}(\hat{\boldsymbol{\theta}}_t)^{\top}\boldsymbol{f}_{i,:}^{\mathtt{I}}(\boldsymbol{\theta}) - \boldsymbol{L}_{i,:}(\boldsymbol{\theta})^{\top}\boldsymbol{f}_{i,:}^{\mathtt{I}}(\boldsymbol{\theta})\right| \\
&\leq \left|\boldsymbol{L}_{i,:}(\hat{\boldsymbol{\theta}}_t)^{\top}\left(\boldsymbol{f}_{i,:}^{\mathtt{I}}(\hat{\boldsymbol{\theta}}_t) - \boldsymbol{f}_{i,:}^{\mathtt{I}}(\boldsymbol{\theta})\right)\right| + \left|\left(\boldsymbol{L}_{i,:}(\hat{\boldsymbol{\theta}}_t) - \boldsymbol{L}_{i,:}(\boldsymbol{\theta})\right)^{\top}\boldsymbol{f}_{i,:}^{\mathtt{I}}(\boldsymbol{\theta})\right| \\
&\leq M\bar{L} \cdot \|\boldsymbol{f}_{i,:}^{\mathtt{I}}(\hat{\boldsymbol{\theta}}_t) - \boldsymbol{f}_{i,:}^{\mathtt{I}}(\boldsymbol{\theta})\|_\infty + M \cdot \|\boldsymbol{L}_{i,:}(\hat{\boldsymbol{\theta}}_t) - \boldsymbol{L}_{i,:}(\boldsymbol{\theta})\|_\infty \\
&\leq M\bar{L}B \cdot \|\hat{\boldsymbol{\theta}}_t - \boldsymbol{\theta}\|_\infty + MB \cdot \|\hat{\boldsymbol{\theta}}_t - \boldsymbol{\theta}\|_\infty \\
&\leq MB(\bar{L} + 1)\max\{\Gamma_{p,t}, \Gamma_{y,t}\}
\end{aligned} \tag{34}$$

where the last inequality from the fact that for all $t > \mathcal{T}$ (see definition in (24)), under the good event $\mathcal{E}_t$:

$$\|\hat{\boldsymbol{\theta}}_t - \boldsymbol{\theta}\|_\infty = \max\{\|\hat{\boldsymbol{y}}_t - \boldsymbol{y}\|_\infty, \|\hat{\boldsymbol{p}}_t - \boldsymbol{p}\|_\infty\} \leq \max\{\|\hat{\boldsymbol{y}}_t - \boldsymbol{y}\|_\infty, \|\hat{\boldsymbol{p}}_t - \boldsymbol{p}\|_1\} \leq \max\{\Gamma_{y,t}, \Gamma_{p,t}\}.$$

Since $\boldsymbol{x}^\star \in \Delta_N^M$ is the optimal solution to Problem (FAIR), we know that $\boldsymbol{L}_{i,:}(\boldsymbol{\theta})^{\top}\boldsymbol{x}_{i,:}^\star \geq \delta^{\mathtt{I}} \cdot \boldsymbol{L}_{i,:}(\boldsymbol{\theta})^{\top}\boldsymbol{f}_{i,:}^{\mathtt{I}}(\boldsymbol{\theta})$. We thus have

$$\begin{aligned}
\boldsymbol{L}_{i,:}(\hat{\boldsymbol{\theta}}_t)^{\top}\boldsymbol{x}_{i,:}^\star + MB \cdot \max\{\Gamma_{p,t}, \Gamma_{y,t}\} \geq \boldsymbol{L}_{i,:}(\boldsymbol{\theta})^{\top}\boldsymbol{x}_{i,:}^\star &\geq \delta^{\mathtt{I}} \cdot \boldsymbol{L}_{i,:}(\boldsymbol{\theta})^{\top}\boldsymbol{f}_{i,:}^{\mathtt{I}}(\boldsymbol{\theta}) \\
&\geq \delta^{\mathtt{I}} \cdot \boldsymbol{L}_{i,:}(\hat{\boldsymbol{\theta}}_t)^{\top}\boldsymbol{f}_{i,:}^{\mathtt{I}}(\hat{\boldsymbol{\theta}}_t) - \delta^{\mathtt{I}}MB(\bar{L} + 1) \cdot \max\{\Gamma_{p,t}, \Gamma_{y,t}\}
\end{aligned} \tag{35}$$

where the first inequality holds because of Eq. (33) and the third inequality holds because of Eq. (34). Hence we have

$$\begin{aligned}
\boldsymbol{L}_{i,:}(\hat{\boldsymbol{\theta}}_t)^{\top}\boldsymbol{x}_{i,:}^\star &\geq \delta^{\mathtt{I}} \cdot \boldsymbol{L}_{i,:}(\hat{\boldsymbol{\theta}}_t)^{\top}\boldsymbol{f}_{i,:}^{\mathtt{I}}(\hat{\boldsymbol{\theta}}_t) - \left(MB + \delta^{\mathtt{I}}MB(\bar{L} + 1)\right) \cdot \max\{\Gamma_{p,t}, \Gamma_{y,t}\} \\
&\overset{(a)}{\Longrightarrow} \boldsymbol{L}_{i,:}(\hat{\boldsymbol{\theta}}_t)^{\top}\boldsymbol{x}_{i,:}^\star \geq \delta^{\mathtt{I}} \cdot \boldsymbol{L}_{i,:}(\hat{\boldsymbol{\theta}}_t)^{\top}\boldsymbol{f}_{i,:}^{\mathtt{I}}(\hat{\boldsymbol{\theta}}_t) - M\log(T)\max\{\Gamma_{p,t}, \Gamma_{y,t}\},
\end{aligned} \tag{36}$$

where (a) follows from $\log(T) > (2 + \bar{r})B > B + \delta^{\mathtt{I}}B(\bar{L} + 1)$. Since the amount of relaxation is $\eta_t = M\log(T)\max\{\Gamma_{p,t}, \Gamma_{y,t}\}$, as defined in Eq. 1, we show that $x^\star$ satisfies the item-fair constraints for Problem (FAIR-RELAX($\hat{\boldsymbol{\theta}}_t, \eta_t$)).

**User-Fair Constraints.** For any $j \in [M]$ and $\boldsymbol{x} \in \Delta_N^M$, under Assumption 3.1, we have

$$\left| \boldsymbol{U}_{:,j}(\hat{\boldsymbol{\theta}}_t)^\top \boldsymbol{x}_{:,j} - \boldsymbol{U}_{:,j}(\boldsymbol{\theta})^\top \boldsymbol{x}_{:,j} \right| \le \|\boldsymbol{U}_{:,j}(\hat{\boldsymbol{\theta}}_t) - \boldsymbol{U}_{:,j}(\boldsymbol{\theta})\|_\infty \|\boldsymbol{x}_{:,j}\|_1 \le B\|\hat{\boldsymbol{\theta}}_t - \boldsymbol{\theta}\|_\infty = B \max\{\Gamma_{p,t}, \Gamma_{y,t}\}.$$
(37)

Also note that for any problem instance $\boldsymbol{\theta}' \in \Theta$, we have that

$$\boldsymbol{U}_{:,j}(\boldsymbol{\theta}')^\top \boldsymbol{f}_{:,j}^{\mathtt{U}}(\boldsymbol{\theta}') = \max_{i \in [N]} U_{i,j}(\boldsymbol{\theta}')$$

Hence, we have the following:

$$\left| \boldsymbol{U}_{:,j}(\hat{\boldsymbol{\theta}}_t)^\top \boldsymbol{f}_{:,j}^{\mathtt{U}}(\hat{\boldsymbol{\theta}}_t) - \boldsymbol{U}_{:,j}(\boldsymbol{\theta})^\top \boldsymbol{f}_{:,j}^{\mathtt{U}}(\boldsymbol{\theta}) \right| = \left| \max_{i \in [N]} U_{i,j}(\hat{\boldsymbol{\theta}}_t) - \max_{i \in [N]} U_{i,j}(\boldsymbol{\theta}) \right|$$
$$\le \|\boldsymbol{U}_{:,j}(\hat{\boldsymbol{\theta}}_t) - \boldsymbol{U}_{:,j}(\boldsymbol{\theta})\|_\infty \le B\|\hat{\boldsymbol{\theta}}_t - \boldsymbol{\theta}\|_\infty = B \max\{\Gamma_{p,t}, \Gamma_{y,t}\}.$$
(38)

where the second inequality follows from local Lipschitzness of $\boldsymbol{U}(\boldsymbol{\theta})$.

Now, since $\boldsymbol{x}^\star \in \Delta_N^M$ is the optimal solution to Problem (FAIR), we know that $\boldsymbol{U}_{:,j}(\boldsymbol{\theta})^\top \boldsymbol{x}_{:,j}^\star \ge \delta^{\mathtt{U}} \cdot \boldsymbol{U}_{:,j}(\boldsymbol{\theta})^\top \boldsymbol{f}_{:,j}^{\mathtt{U}}(\boldsymbol{\theta})$. Combining this with the Eq. (37) and Eq. (38), we have

$$\boldsymbol{U}_{:,j}(\hat{\boldsymbol{\theta}}_t)^\top \boldsymbol{x}_{:,j}^\star + B \max\{\Gamma_{p,t}, \Gamma_{y,t}\} \ge \boldsymbol{U}_{:,j}(\boldsymbol{\theta})^\top \boldsymbol{x}_{:,j}^\star \ge \delta^{\mathtt{U}} \cdot \boldsymbol{U}_{:,j}(\boldsymbol{\theta})^\top \boldsymbol{f}_{:,j}^{\mathtt{U}}(\boldsymbol{\theta})$$
$$\ge \delta^{\mathtt{U}} \cdot \boldsymbol{U}_{:,j}(\hat{\boldsymbol{\theta}}_t)^\top \boldsymbol{f}_{:,j}^{\mathtt{U}}(\hat{\boldsymbol{\theta}}_t) - \delta^{\mathtt{U}} B \cdot \max\{\Gamma_{p,t}, \Gamma_{y,t}\}.$$
(39)

Hence we have

$$\boldsymbol{U}_{:,j}(\hat{\boldsymbol{\theta}}_t)^\top \boldsymbol{x}_{:,j}^\star \ge \boldsymbol{U}_{:,j}(\hat{\boldsymbol{\theta}}_t)^\top \boldsymbol{f}_{:,j}^{\mathtt{U}}(\hat{\boldsymbol{\theta}}_t) - \left(1 + \delta^{\mathtt{U}}\right) B \max\{\Gamma_{p,t}, \Gamma_{y,t}\}$$
$$\implies \boldsymbol{U}_{:,j}(\hat{\boldsymbol{\theta}}_t)^\top \boldsymbol{x}_{:,j}^\star \ge \boldsymbol{U}_{:,j}(\hat{\boldsymbol{\theta}}_t)^\top \boldsymbol{f}_{:,j}^{\mathtt{U}}(\hat{\boldsymbol{\theta}}_t) - M \log(T) \max\{\Gamma_{p,t}, \Gamma_{y,t}\}.$$
(40)

**Proof of (iii).** We can show $\hat{\boldsymbol{x}}_t$ is feasible to Problem (FAIR-RELAX($\boldsymbol{\theta}, \eta_t$)) via a similar argument as our proof for (ii)

**Item-Fair Constraints.** First note that

$$\boldsymbol{L}_{i,:}(\boldsymbol{\theta})^\top \hat{\boldsymbol{x}}_{t,i,:} + MB \max\{\Gamma_{p,t}, \Gamma_{y,t}\}$$
$$\overset{(a)}{\ge} \boldsymbol{L}_{i,:}(\hat{\boldsymbol{\theta}}_t)^\top \hat{\boldsymbol{x}}_{t,i,:}$$
$$\overset{(b)}{\ge} \delta^{\mathtt{I}} \cdot \boldsymbol{L}_{i,:}(\hat{\boldsymbol{\theta}}_t)^\top \boldsymbol{f}_{i,:}^{\mathtt{I}}(\hat{\boldsymbol{\theta}}_t) - M \log(T) \max\{\Gamma_{p,t}, \Gamma_{y,t}\}$$
(41)
$$\overset{(c)}{\ge} \delta^{\mathtt{I}} \cdot \boldsymbol{L}_{i,:}(\boldsymbol{\theta})^\top \boldsymbol{f}_{i,:}^{\mathtt{I}}(\boldsymbol{\theta}) - M \log(T) \max\{\Gamma_{p,t}, \Gamma_{y,t}\} - \delta^{\mathtt{I}} MB(\bar{L}+1) \max\{\Gamma_{p,t}, \Gamma_{y,t}\}.$$

Here, (a) follows from an identical argument as shown in (33), while replacing $\boldsymbol{x}$ with $\hat{\boldsymbol{x}}_t$; (b) follows from feasibility of $\hat{\boldsymbol{x}}_t$ to Problem (FAIR-RELAX($\boldsymbol{\theta}, \eta_t$)); (c) follows from (34). Hence, since $\log(T) > (2 + \bar{r})B > B + \delta^{\mathtt{I}} B(\bar{L}+1)$, we conclude

$$\boldsymbol{L}_{i,:}(\boldsymbol{\theta})^\top \hat{\boldsymbol{x}}_{t,i,:} \ge \delta^{\mathtt{I}} \cdot \boldsymbol{L}_{i,:}(\boldsymbol{\theta})^\top \boldsymbol{f}_{i,:}^{\mathtt{I}}(\boldsymbol{\theta}) - 2M \log(T) \max\{\Gamma_{p,t}, \Gamma_{y,t}\}.$$
(42)

**User-Fair Constraints.** We again follow the same arguments as in our proof for (ii) and establish the following:

$$\boldsymbol{U}_{:,j}(\boldsymbol{\theta})^\top \hat{\boldsymbol{x}}_{t,:,j} + B \max\{\Gamma_{p,t}, \Gamma_{y,t}\}$$
$$\ge \boldsymbol{U}_{:,j}(\hat{\boldsymbol{\theta}}_t)^\top \hat{\boldsymbol{x}}_{t,:,j}$$
$$\ge \delta^{\mathtt{U}} \cdot \boldsymbol{U}_{:,j}(\hat{\boldsymbol{\theta}}_t)^\top \boldsymbol{f}_{:,j}^{\mathtt{U}}(\hat{\boldsymbol{\theta}}_t) - M \log(T) \max\{\Gamma_{p,t}, \Gamma_{y,t}\}$$
(43)
$$\ge \delta^{\mathtt{U}} \cdot \boldsymbol{U}_{:,j}(\boldsymbol{\theta})^\top \boldsymbol{f}_{:,j}^{\mathtt{U}}(\boldsymbol{\theta}) - M \log(T) \max\{\Gamma_{p,t}, \Gamma_{y,t}\} - \delta^{\mathtt{U}} B \max\{\Gamma_{p,t}, \Gamma_{y,t}\},$$

which then gives

$$\boldsymbol{U}_{:,j}(\boldsymbol{\theta})^\top \hat{\boldsymbol{x}}_{t,:,j} \ge \delta^{\mathtt{U}} \cdot \boldsymbol{U}_{:,j}(\boldsymbol{\theta})^\top \boldsymbol{f}_{:,j}^{\mathtt{U}}(\boldsymbol{\theta}) - 2M \log(T) \max\{\Gamma_{p,t}, \Gamma_{y,t}\}.$$
(44)

## D.4 Proof of Lemma D.2

For completeness, let us recall the definitions of $\Gamma_{y,t}$ and $m(t)$ in Eq. (23):

$$\Gamma_{y,t} = \frac{2\log(T)}{\sqrt{m(t)\epsilon_t}} \, , \, m(t) = \max\left\{1, \frac{t}{\log(T)} - \sqrt{t\log(T)/2}\right\}.$$

We define related variables $\widetilde{\Gamma}_{y,j,t}$ and $\widetilde{m}_j(t)$ as followed:

$$\widetilde{\Gamma}_{y,j,t} = \frac{2\log(T)}{\sqrt{\widetilde{m}_j(t)\epsilon_t}} \, , \, \widetilde{m}_j(t) = \max\{1, tp_j - \sqrt{t\log(T)/2}\}. \tag{45}$$

Since for $\log(T) > \frac{1}{\min_{j\in[M]} p_j}$ and $t > \mathcal{T}$ (see definition of the cutoff time $\mathcal{T}$ in (24)) such that $m(t) > 1$, we know $\widetilde{m}_j(t) \geq m(t) > 1$ and thus $\Gamma_{y,t} \geq \widetilde{\Gamma}_{y,j,t}$ for any $j \in [M]$. We thus have

$$\mathbb{P}\left(|\hat{y}_{i,j,t+1} - y_{i,j}| \geq \Gamma_{y,t}\right) \leq \mathbb{P}\left(|\hat{y}_{i,j,t+1} - y_{i,j}| \geq \widetilde{\Gamma}_{y,j,t}\right) \tag{46}$$

Hence,

$$\begin{aligned}
\mathbb{P}(\mathcal{E}_t^c) &\leq \mathbb{P}(\|\hat{\boldsymbol{y}}_t - \boldsymbol{y}\|_\infty \geq \Gamma_{y,t}) + \mathbb{P}(\|\hat{\boldsymbol{p}}_t - \boldsymbol{p}\|_1 \geq \Gamma_{p,t}) \\
&\leq \sum_{i\in[N]}\sum_{j\in[M]} \mathbb{P}\left(|\hat{y}_{i,j,t+1} - y_{i,j}| \geq \Gamma_{y,t}\right) + \mathbb{P}(\|\hat{\boldsymbol{p}}_t - \boldsymbol{p}\|_1 \geq \Gamma_{p,t}) \\
&\overset{(a)}{\leq} \sum_{i\in[N]}\sum_{j\in[M]} \mathbb{P}\left(|\hat{y}_{i,j,t+1} - y_{i,j}| \geq \widetilde{\Gamma}_{y,j,t}\right) + \mathbb{P}(\|\hat{\boldsymbol{p}}_t - \boldsymbol{p}\|_1 \geq \Gamma_{p,t}) \\
&\overset{(b)}{\leq} \sum_{i\in[N]}\sum_{j\in[M]} \mathbb{P}\left(|\hat{y}_{i,j,t+1} - y_{i,j}| \geq \widetilde{\Gamma}_{y,j,t}\right) + \frac{1}{T},
\end{aligned} \tag{47}$$

where (a) follows from (46); (b) follows directly from the concentration inequality to bound empirical distribution estimates; in particular, we use Lemma G.3 and take $\delta = \frac{1}{T}$. Hence, in the following, it suffices to bound $\mathbb{P}\left(|\hat{y}_{i,j,t+1} - y_{i,j}| \geq \widetilde{\Gamma}_{y,j,t}\right)$ for any $i \in [N], j \in [M]$.

Recall that $n_{j,t}$ is the total number of type $j$ arrivals up to time $t$. Then, consider the following

$$\begin{aligned}
\mathbb{P}\left(|\hat{y}_{i,j,t+1} - y_{i,j}| \geq \widetilde{\Gamma}_{y,j,t}\right) &= \mathbb{P}\left(|\hat{y}_{i,j,t+1} - y_{i,j}| \geq \widetilde{\Gamma}_{y,j,t}, n_{j,t} \geq \widetilde{m}_j(t)\right) \\
&\quad + \mathbb{P}\left(|\hat{y}_{i,j,t+1} - y_{i,j}| \geq \widetilde{\Gamma}_{y,j,t}, n_{j,t} < \widetilde{m}_j(t)\right) \\
&\leq \underbrace{\mathbb{P}\left(|\hat{y}_{i,j,t+1} - y_{i,j}| \geq \widetilde{\Gamma}_{y,j,t}, n_{j,t} \geq \widetilde{m}_j(t)\right)}_{A} + \underbrace{\mathbb{P}\left(n_{j,t} < \widetilde{m}_j(t)\right)}_{B}.
\end{aligned} \tag{48}$$

We would bound terms $A$ and $B$ respectively.

**Bounding $A$.** Recall that $\tau_{j,k} \in [T]$ is the round at which the $k$th consumer of type $j$ arrived.

$$\begin{aligned}
A &= \mathbb{P}\left(\left|\hat{y}_{t+1,i,j} - y_{i,j}\right| \geq \widetilde{\Gamma}_{y,j,t}, n_{j,t} \geq \widetilde{m}_j(t)\right) \\
&\leq \mathbb{P}\left(\left|\max_{\widetilde{m}_j(t)\leq n\leq t} \frac{1}{n}\sum_{k=1}^n \frac{\mathbb{I}\{I_{\tau_{j,k}} = i, z_{\tau_{j,k}} = 1\}}{x_{\tau_{j,k},i,j}} - y_{i,j}\right| \geq \widetilde{\Gamma}_{y,j,t}, n_{j,t} \geq \widetilde{m}_j(t)\right) \\
&\leq \mathbb{P}\left(\left|\max_{\widetilde{m}_j(t)\leq n\leq t} \frac{1}{n}\sum_{k=1}^n \frac{\mathbb{I}\{I_{\tau_{j,k}} = i, z_{\tau_{j,k}} = 1\}}{x_{\tau_{j,k},i,j}} - y_{i,j}\right| \geq \widetilde{\Gamma}_{y,j,t}\right) \\
&\leq \sum_{\widetilde{m}_j(t)\leq n\leq t} \mathbb{P}\left(\left|\frac{1}{n}\sum_{k=1}^n \frac{\mathbb{I}\{I_{\tau_{j,k}} = i, z_{\tau_{j,k}} = 1\}}{x_{\tau_{j,k},i,j}} - y_{i,j}\right| \geq \widetilde{\Gamma}_{y,j,t}\right) \\
&\leq \sum_{\widetilde{m}_j(t)\leq n\leq t} \mathbb{P}\left(\left|\frac{1}{n}\sum_{k=1}^n \frac{\mathbb{I}\{I_{\tau_{j,k}} = i, z_{\tau_{j,k}} = 1\}}{x_{\tau_{j,k},i,j}} - y_{i,j}\right| \geq \frac{2\log(T)}{\sqrt{n}\epsilon_t}\right),
\end{aligned} \tag{49}$$

where in the final inequality we used the definition of $\widetilde{\Gamma}_{y,j,t}$ in (45) and the fact that $n \geq \widetilde{m}_j(t)$. Note that for any $k \in \mathbb{N}$, we have $\mathbb{E}\left[\frac{\mathbb{I}\{I_{\tau_{j,k}}=i,z_{\tau_{j,k}}=1\}}{x_{\tau_{j,k},i,j}}\,\middle|\,\mathcal{F}_{\tau_{j,k}}\right] = y_{i,j}$, where $\mathcal{F}_{\tau_{j,k}}$ is the sigma-algebra generated from all randomness up to round $\tau_{j,k}$. Therefore, $\mathcal{M}_k = \frac{\mathbb{I}\{I_{\tau_{j,k}}=i,z_{\tau_{j,k}}=1\}}{x_{\tau_{j,k},i,j}} - y_{i,j}$ is a Martingale difference sequence. Further, we have

$$-1 \;\leq\; \mathcal{M}_k \;\leq\; \frac{1}{x_{\tau_{j,k},i,j}} \;\overset{(a)}{\leq}\; \frac{1}{\epsilon_{\tau_{j,k}}} \;\leq\; \frac{1}{\epsilon_t} \;\overset{(b)}{\Longrightarrow}\; |\mathcal{M}_k| \leq \frac{1}{\epsilon_t}, \tag{50}$$

where (a) follows from the fact that $x_{\tau_{j,k},i,j} = \frac{1}{N} \geq \epsilon_{\tau_{j,k}}$ if $\|\hat{y}_t\|_\infty \geq 1$ and $x_{\tau_{j,k},i,j} = (1 - N\epsilon_t)\hat{x}_{\tau_{j,k},i,j} + \epsilon_{\tau_{j,k}} \geq \epsilon_{\tau_{j,k}}$ if $\|\hat{y}_t\|_\infty < 1$ according to FORM (Algorithm 1); (b) follows from $\epsilon_t \leq \frac{1}{N}$ so $\frac{1}{\epsilon_t} \geq N \geq 1$, and hence $\mathcal{M}_k \geq -1 \geq -\frac{1}{\epsilon_t}$. Also,

$$\mathbb{E}\left[\mathcal{M}_k^2\,\middle|\,\mathcal{F}_{\tau_{j,k}}\right] = \mathbb{E}\left[\frac{\mathbb{I}\{I_{\tau_{j,k}}=i,z_{\tau_{j,k}}=1\}}{x_{\tau_{j,k},i,j}^2}\,\middle|\,\mathcal{F}_{\tau_{j,k}}\right] - y_{i,j}^2 \leq \frac{1}{x_{\tau_{j,k},i,j}} \leq \frac{1}{\epsilon_t}. \tag{51}$$

Hence, by Bernstein's inequality (see Lemma G.4), we have

$$\mathbb{P}\left(\Big|\sum_{k\in[n]} \mathcal{M}_k\Big| \geq \sqrt{(e-2)V_n \log(2/\delta)}\right) \leq \delta$$

$$\overset{(a)}{\Longrightarrow} \mathbb{P}\left(\Big|\frac{1}{n}\sum_{k=1}^{n} \frac{\mathbb{I}\{I_{\tau_{j,k}}=i,z_{\tau_{j,k}}=1\}}{x_{\tau_{j,k},i,j}} - y_{i,j}\Big| \geq \frac{2\log(T)}{\sqrt{n\epsilon_t}}\right) \leq \frac{2}{T^2}, \tag{52}$$

where (a) follows from invoking Lemma G.4 by taking $c_k = \frac{1}{\epsilon_t}$ and $V_k = \frac{2k\log(T)}{(e-2)\epsilon_t}$, and $\delta = \frac{1}{2T^2}$. Combining Eq. (49) and Eq. (52) yields

$$A \;\leq\; \sum_{\widetilde{m}_j(t)\leq n\leq t} \frac{2}{T^2} \;\leq\; \frac{2t}{T^2} \;\leq\; \frac{2}{T}. \tag{53}$$

**Bounding** $B$. Denote $\mathcal{M}_t = \mathbb{I}\{J_t = j\} - p_j$ and it is easy to see $\mathcal{M}_t$ is a Martingale difference sequence bounded between $[-1,1]$. Hence, by recognizing $n_{j,t} - t \cdot p_j = \sum_{\tau\in[t]} \mathcal{M}_\tau$ and applying the Azuma-Hoeffding inequality, for any $\epsilon > 0$, we have

$$\mathbb{P}\Big(n_{j,t} - t\cdot p_j < -\epsilon\Big) = \mathbb{P}\Big(\sum_{\tau\in[t]} \mathcal{M}_\tau < -\epsilon\Big) \leq \exp\left(-\frac{2\epsilon^2}{t}\right)$$

$$\Longrightarrow B = \mathbb{P}\Big(n_{j,t} < \widetilde{m}_j(t)\Big) = \mathbb{P}\Big(n_{j,t} - t\cdot p_j < -\sqrt{t\log(T)/2}\Big) \overset{(a)}{\leq} \frac{1}{T}, \tag{54}$$

where in (a) we take $\epsilon = \sqrt{t\log(T)/2}$.

## E  Extensions to Additional Setups

Our fair online recommendation framework, Problem (FAIR), and method FORM can be extended to accommodate additional variants of our setup. In Section E.1, we discuss how our framework/method can additionally handle periodic arrivals with the same performance guarantees. In Section E.2, we detail how our framework/method extends to handling the assortment recommendation setting.

### E.1  Fair Online Recommendation with Periodic Arrivals

In real-world scenarios, user arrivals are often non-stationary, displaying variations over time due to periodic user preferences and behavior. This is particularly evident in many recommendation systems that exhibit daily or weekly periodicity, with consistent patterns of user arrivals and interactions throughout the day or week. For instance, in an e-commerce setting, higher user activity and engagement may be observed during lunch breaks, evenings, or late-night browsing. Similarly, recommendation systems may experience increased user engagement during weekends compared

to weekdays. Recognizing the presence of these periodic patterns, we extend our concept of a fair solution and our algorithm FORM to accommodate the online setting with periodic arrivals.

Consider a scenario where users arrive in each round $t \in [T]$ according to a probability distribution $\boldsymbol{p}_t \in \Delta_M$. Additionally, suppose that there exists a period length $Q \in (0, T)$ and distributions $\widetilde{\boldsymbol{p}}^{(1)} \ldots \widetilde{\boldsymbol{p}}^{(Q)}$ such that $\boldsymbol{p}_{mQ+q} = \widetilde{\boldsymbol{p}}^{(q)}$ for any $m = 0, 1, 2 \ldots$ and $q = 0, 1 \ldots Q - 1$. By defining $\boldsymbol{p}$ as the time-averaged arrival distribution, represented by $\boldsymbol{p} = \frac{1}{T} \sum_{t \in [T]} \boldsymbol{p}_t$, we can generalize the fair recommendation problem, Problem (FAIR), to incorporate this periodic arrival setup.

Yet, similar to the stationary setting, the platform does not have prior knowledge of the sequence of user arrival distributions $\boldsymbol{p}_1 \ldots \boldsymbol{p}_T$, nor the time-averaged distribution $\boldsymbol{p}$. Although solving Problem (FAIR) w.r.t. the time-averaged distribution $\boldsymbol{p}$ seems to be more challenging compared to that under stationary arrivals, we can in fact still obtain accurate estimates by slightly modifying the design of FORM as follows: instead of estimating $\boldsymbol{p}$ using all historical data (as done in Eq. (2)), we only obtain estimates over a sliding window of length $\mathcal{W} > 0$. That is, we estimate the arrival rates of type-$j$ user at round $t + 1$ via the following:

$$\hat{\boldsymbol{p}}_{t+1,j} = \frac{1}{\min\{t, \mathcal{W}\}} \sum_{\tau = \max\{1, t-\mathcal{W}+1\}}^{t} \mathbb{I}\{J_t = j\}. \tag{55}$$

By doing so and keeping all other procedures unchanged, we show in Theorem E.1 that FORM would yield the same theoretical guarantees for its final output under the setting with periodic arrivals, when the size of the sliding window size $\mathcal{W}$ is chosen appropriately.

**Theorem E.1 (Performance of FORM under Periodic Arrivals)** *If we apply FORM (Algorithm 1) with an estimated arriving probability $\hat{\boldsymbol{p}}_{t+1,j}$ taking the form in Eq. (55) and $\Gamma_{p,t} = \frac{5\sqrt{\log(3T)}}{\sqrt{\mathcal{W}}}$ with a sliding window of size $\mathcal{W} = QT^{\frac{2}{3}}$. Then, for sufficiently large $T$, we have the following:*

- *the revenue regret is at most $\mathbb{E}[\mathcal{R}(T)] \leq \mathcal{O}(MN^{\frac{1}{3}}T^{-\frac{1}{3}})$*
- *the fairness regret is at most $\mathbb{E}[\mathcal{R}_F(T)] \leq \mathcal{O}(MN^{\frac{1}{3}}T^{-\frac{1}{3}})$*

The proof of Theorem E.1 is provided in Section E.1.1.

**Remark E.1 (Lack of knowledge of the period length)** *We remark that in the periodic setup, we assume knowledge of the period length $Q > 0$. This assumption aligns with practical scenarios where platforms are aware of intraday, daily, weekly, or seasonal user arrival cycles. However, if the period length $Q$ is unknown, one might consider employing a standard meta "expert" algorithm from the bandit literature on top of FORM. This approach involves using sliding window lengths as experts, representing different expert windows in the set $\mathcal{A} = aT^{\frac{2}{3}} : a = 1, 2 \ldots \bar{Q}$ for some $\bar{Q} > Q$. Each expert produces estimates of $\boldsymbol{p}$ and the fair solution based on their window length. We maintain weights over these estimates using an EXP3 algorithm. However, while this approach may eventually approximate the cumulative revenue of the best expert with the correct period length $Q$, the expected constraint violation w.r.t. the true parameters $\boldsymbol{\theta}$ may not be sublinear or vanishing over time. This is because the estimates from experts with incorrect period lengths can significantly violate constraints, resulting in overall large constraint violations. Therefore, optimizing the fair solution under a periodic arrival setup with an unknown period length remains an open question.*

### E.1.1 Proof of Theorem E.1

First, similar to Lemma D.2 that presents a concentration bound for empirical estimates of $\boldsymbol{p}$ using all historical data, if we construct an empirical estimate for $\boldsymbol{p}$ with Eq. (55) using sliding window length $\mathcal{W}$, again applying Lemma G.3, we have for any $t > \mathcal{W}$ s.t.

$$\mathbb{P}\Big(\|\boldsymbol{p} - \hat{\boldsymbol{p}}_t\|_1 > \frac{5\sqrt{\log(3T)}}{\sqrt{\mathcal{W}}}\Big) \leq \frac{1}{T} \tag{56}$$

Recall the definition of $\mathcal{T}$ in Eq. (24). We additionally define

$$\overline{\mathcal{T}} = \max\{\mathcal{W} + 1, \mathcal{T}\} = \mathcal{O}(\mathcal{W}T^{\frac{2}{3}}). \tag{57}$$

Following the same analysis as bounding the regret for Theorem 3.1 in Eq. (29), while replacing $\mathcal{T}$ with $\overline{\mathcal{T}}$ and considering $\Gamma_{p,t} = \frac{5\sqrt{\log(3T)}}{\sqrt{\mathcal{W}}}$, we have

$$\frac{1}{T}\sum_{t\in[T]}\mathbb{E}\Big[\text{REV}(\boldsymbol{x}^\star,\boldsymbol{\theta}) - \text{REV}(\boldsymbol{x}_t,\boldsymbol{\theta})\Big] \leq \mathcal{O}\Big(\frac{\overline{\mathcal{T}}}{T} + \frac{M}{T}\sum_{t>\overline{\mathcal{T}}}(\Gamma_{p,t}+\Gamma_{y,t}) + \frac{MN}{T}\sum_{t>\overline{\mathcal{T}}}\epsilon_t\Big)$$

$$= \mathcal{O}(MN^{\frac{1}{3}}T^{-\frac{1}{3}}).$$

A similar analysis as Eq. (31) applies to bounding the fairness regret $\mathcal{R}_F(T)$. ∎

### E.2 Fair Online Assortment Recommendation

Our model in Section 2 primarily considered a setting where a single item is offered to each user at each round, which applies to various real-world settings. In this section, we discuss an extension of our framework/method to a setting in which the platform might wish to recommend an assortment to its users, which applies to real-world scenarios such as job recommendation, where a number of most relevant jobs are recommended to a candidate, or e-commerce sites, where a small assortment of items might be featured on the front page whenever a user arrives.

**Extension of our framework.** To consider an assortment recommendation setting, we let $w_{i,j} > 0$ be the *weight* associated with each item $i$ for a type-$j$ user. If a type-$j$ user arrives onto the platform and gets offered assortment $S$, he/she will choose/purchase item $i$ from the assortment with probability $\frac{w_{i,j}}{1+w_j(S)}$, where $w_j(S) = \sum_{i\in S} w_{i,j}$, based on the MNL model [65]. This is different from the single-item recommendation setting, where we have well-defined purchase probability $y_{i,j}$ for each item-user pair. When we recommend assortments, the purchase probability not only depends on an item's weight, but also the assortment it gets presented in. Given that, we define the problem instance as $\boldsymbol{\theta} = (\boldsymbol{p}, \boldsymbol{w}, \boldsymbol{r})$.

Our fair recommendation framework, Problem (FAIR), can be readily extended by letting our decision variable be $\boldsymbol{q} = \{q_j(S) : S \subseteq [N], |S| \leq K, j \in [M]\}$, where $q_j(S)$ denote the probability of presenting assortment $S$ to a type-$j$ user. Then, we can formulate the fair recommendaton problem under the same idea:

$$\max_{\boldsymbol{q}} \ \text{REV}(\boldsymbol{q}) \quad \text{s.t.} \quad O_i^{\text{I}}(\boldsymbol{q}) \geq \delta^{\text{I}} \cdot O_i^{\text{I}}(\boldsymbol{f}^{\text{I}}) \ \forall \ i \in [N]$$

$$O_j^{\text{U}}(\boldsymbol{q}) \geq \delta^{\text{U}} \cdot O_j^{\text{U}}(\boldsymbol{f}^{\text{U}}) \ \forall \ j \in [M],$$

(FAIR-ASSORT)

where

$$\text{REV}(\boldsymbol{q}) = \sum_{j\in[M]} p_j \sum_{S\in[N],|S|\leq K} q_j(S)\frac{\sum_{i\in S} r_i w_{i,j}}{1+w_j(S)}$$

denote the expected revenue under recommendation $\boldsymbol{q}$. On the other hand, the forms of the item/user outcome functions $O_i^{\text{I}}(\boldsymbol{q},\boldsymbol{\theta}), O_j^{\text{U}}(\boldsymbol{q},\boldsymbol{\theta})$ as well as their respective fair solutions $\boldsymbol{f}^{\text{I}}(\boldsymbol{\theta}), \boldsymbol{f}^{\text{U}}(\boldsymbol{\theta})$ can again take any forms that depend on the problem instance $\boldsymbol{\theta}$, based on their own needs.

**Extension of our algorithm for the online setting.** Given the extended framework Problem (FAIR-ASSORT), we can similarly extend the algorithm FORM for the assortment recommendation setting, which is outlined in Algorithm 2.

Here, Algorithm 2 follows the same relaxation-then-exploration design as discussed in Section 3.3 and Algorithm 1. Whenever our estimate of the problem instance $\boldsymbol{\theta}$ is updated, the algorithm first solves a relaxed version of Problem (FAIR-ASSORT) using the updated estimate, and then adds randomized exploration to stimulate learning; see Steps 2(b) and 2(c) in Algorithm 2.[5]

The main difference between Algorithm 2 and Algorithm 1 lies in the learning mechanism. In order to learn the weights $\boldsymbol{w}$ in the assortment setting, we adopt the learning mechanism from MNL-Bandit [3]. The rounds in which type-$j$ arrives are divided into epochs. In each epoch $\ell_j$, the same assortment $S_j$ is offered until a no-purchase option is observed. We then apply Eq. (58) to estimate $\boldsymbol{w}_{:,j}$, which is an unbiased estimator according to Corollary A.1 in [3]. To estimate arrival probabilities, we

---

[5]For solving Problem (FAIR-ASSORT) in our numerical experiments in Sections 4 and F.2, we use the CBC solver accessed via the PuLP library in Python.

**Algorithm 2** `FORM` for Assortment Recommendation

---

**Input:** (i) $N$ items with revenues $\boldsymbol{r} \in \mathbb{R}^N$, item outcome $O_i^{\text{I}}(.,.,.), i \in [N]$ and item-fair solution $\boldsymbol{f}^{\text{U}}(.)$; (ii) $M$ types of users with user outcome $O_j^{\text{U}}(.,.,.), j \in [M]$ and user-fair solution $\boldsymbol{f}^{\text{U}}(.)$ (iii) fairness parameters $\delta^{\text{I}}, \delta^{\text{U}} \in [0,1]$. (iv) parameters $\epsilon_t$ and $\eta_t$.

1. **Initialization and Setting the Parameters.** For all $i \in [N], j \in [M]$, initialize $\hat{w}_{1,i,j} = 1/2$ and $\hat{p}_{1,j} = 1/M$. Randomly select $S_j \sim \mathcal{S}$, where $\mathcal{S} = \{S \subseteq [N], |S| \leq K\}$ denote the collection of all possible assortments. Let $\ell_j = 0, \mathcal{T}_{i,j} = \emptyset$.

2. While $t \leq T$.
   - Observe the type of the arriving user $J_t$ and offer assortment $S_{J_t}$. Update number of arrivals $n_{J_t,t} \leftarrow n_{J_t,t} + 1$.
   - Observe purchase decision $z_t \in \{0\} \cup S_{J_t}$ (where 0 means no-purchase)
     - If $z_t = 0$,
       - (a) Let $\mathcal{T}_{i,J_t} = \mathcal{T}_{i,J_t} \cup \{\ell_{J_t}\}$ for $i \in S_{J_t}$, where $\mathcal{T}_{i,J_t}$ are the epochs in which item $i$ gets shown to user type $J_t$. Then, increment $\ell_{J_t} \leftarrow \ell_{J_t} + 1$, where $\ell_{J_t}$ denotes the number of epochs for user type $J_t$.
       - (b) **Update estimates for item weights and arrival probabilities.** Let

       $$\hat{w}_{t,i,J_t} = \frac{1}{|\mathcal{T}_{i,J_t}|} \sum_{\tau \in \mathcal{T}_{i,J_t}} \sum_{t \in \mathcal{E}_{\tau,J_t}} \mathbb{I}\{z_t = i\} \quad \forall i \in [N] \quad \text{and} \quad \hat{p}_{t,j} = \frac{1}{t} n_{j,t} \quad \forall j \in [M]$$
       (58)

       - (c) **Solve a Relaxed Problem** (`FAIR-ASSORT`) **with the Estimated Instance.** Given estimated instance $\hat{\boldsymbol{\theta}} = (\hat{\boldsymbol{p}}_t, \hat{\boldsymbol{w}}_t, \boldsymbol{r})$ and the magnitude of relaxation $\eta_t$, let $\hat{\boldsymbol{q}}_t$ be the optimal solution to the following:

       $$\max_{\boldsymbol{q}} \ \text{REV}(\boldsymbol{q}, \hat{\boldsymbol{\theta}}_t) \quad \text{s.t.} \quad O_i^{\text{I}}(\boldsymbol{q}, \hat{\boldsymbol{\theta}}_t) \geq \delta^{\text{I}} \cdot O_i^{\text{I}}(\boldsymbol{f}^{\text{I}}(\hat{\boldsymbol{\theta}}_t), \hat{\boldsymbol{\theta}}_t) - \eta_t \ \forall \ i \in [N]$$
       $$O_j^{\text{U}}(\boldsymbol{q}, \hat{\boldsymbol{\theta}}_t) \geq \delta^{\text{U}} \cdot O_j^{\text{U}}(\boldsymbol{f}^{\text{U}}(\hat{\boldsymbol{\theta}}_t), \hat{\boldsymbol{\theta}}_t) - \eta_t \ \forall \ j \in [M],$$
       (59)

       - (d) **Recommend with Randomized Exploration.** Update the assortment to be recommended to type-$J_t$ user based on the recommendation probabilities $S_{J_t} \sim \boldsymbol{q}_{t,J_t}$, where

       $$\boldsymbol{q}_{t,J_t}(S) = (1 - |\mathcal{S}|\epsilon_t)\hat{\boldsymbol{q}}_{t,J_t}(S) + \epsilon_t \quad \forall S \in \mathcal{S}, j \in [M].$$
       (60)

     - Else, $\mathcal{E}_{\ell_{J_t},J_t} \leftarrow \mathcal{E}_{\ell_{J_t},J_t} \cup \{t\}$
   - $t \leftarrow t + 1$.

---

again use the sample mean for $\boldsymbol{p}$. As discussed in Section 3.3, any learning mechanism that provides accurate estimates for sufficiently large $T$ would be suitable here. The magnitudes of exploration and relaxation, $\epsilon_t$ and $\eta_t$, are adjustable inputs for platforms to fine-tune in practice.

While a detailed theoretical analysis of the expanded algorithm is reserved for future work, it is clear that our high-level ideas behind `FORM`, including incorporating relaxations for fair recommendations and adding exploration to stimulate learning, seamlessly transition to the assortment recommendation context. The efficacy of our extended framework/method in the assortment recommendation setting is further validated in our case study on Amazon review data (see Section 4).

## F   Supplements for Case Study on Amazon Review Data

### F.1   More Details of Our Baselines

In this section, we discuss the baselines introduced in existing works, specifically FairRec [55] and TFROM [71], which are state-of-the-art approaches targeting multi-sided fairness. We detail the main setups of both approaches and highlight the key differences between these methods and ours.

**FairRec** [55]. As we briefly remarked in Section 4, FairRec targets a single-shot recommendation problem where there are $N$ items and $M$ users, and makes a single recommendation decision that displays $K$ items to each of the $M$ users, assuming full knowledge of the exact relevance score between each pair of item and user. Let $\mathcal{A} = \{(A_j)_{j \in [M]} : A_j \subset [N], |A_j| \leq K\}$ denote the the algorithm's recommendation decision. The authors of [55] show that FairRec can achieve (i) maxmin share (MMS) of visibility (exposure) for most of the items and non-zero visibility for the rest; (ii) envy-free up to one item (EF1) fairness for every user, meaning that for every pair of users

$j, j' \in [M]$, there should exist an item $p \in A_{j'}$ such that the utility that user $j$ receives from $A_j$ is at least the utility that user $j'$ receives from $A_{j'} \setminus \{p\}$.

There are several key differences between our setting/method and FairRec. (i) FairRec focuses on an offline setting where the platform makes a single-shot recommendation with full knowledge of user preferences (i.e., relevance scores). In contrast, our work targets the more challenging online setting where user data is unknown and must be continuously learned. We maintain fairness in this online setting, dealing with users who arrive stochastically, and ensuring fairness over a longer time horizon. (ii) FairRec assumes specific fairness notions (i.e., maxmin fairness regarding visibility for items and envy-free fairness up to one item for users). Our framework, on the other hand, is much more flexible, accommodating a wide range of outcome functions and fairness notions, as discussed in Section 2.2. See, also, Section F.2 where we conduct additional experiments on the Amazon review data under alternative outcome functions and fairness notions, and demonstrate that FORM remains effective in making fair recommendations in an online environment.

**TFROM** [71]. Similar to FairRec, the authors of [71] consider a recommendation problem where relevance scores between items and users are known in advance. To impose fairness for items, TFROM enforces either uniform fairness or quality-weighted fairness regarding item visibility (see Definitions 1 and 2 in [71]). For user fairness, TFROM aims to achieve uniform normalized discounted cumulative gain (NDCG) among users. [71] does not provide theoretical performance results, but numerically shows that TFROM achieves low variance in item visibility and NDCG among users in online settings, thus ensuring two-sided fairness. TFROM is provided in two versions: one for the offline setting and one for the online setting, with the latter dynamically maintaining low variance in item exposure and user NDCG scores over time. We compare our algorithm with the online version of TFROM in our experiments in Section 4.

While TFROM addresses online user arrivals, it again differs significantly from our work in several ways: (i) Like FairRec, it assumes full knowledge of user preferences (i.e., relevance scores) and lacks a learning stage, ignoring potential biases from corrupted data; (ii) It focuses on pre-specified outcomes and fairness notions for items/users, lacking the flexibility of our framework; (iii) No theoretical guarantees are provided for TFROM's performance, whereas we offer full theoretical guarantees for our algorithm despite data uncertainty.

Finally, it is important to note that neither FairRec nor TFROM considers the platform's revenue, which is another key focus of our work. While the aforementioned works focus on two-sided fairness in recommender systems, our framework takes a step forward and adopts a multi-sided perspective, balancing the interests of the platform, items, users, and potentially other stakeholders. The efficacy of our framework is highlighted in Figure 1b, showing how our algorithm manages to maintain high platform revenue while ensuring fairness for other stakeholders.

### F.2 Additional Experiments under Alternative Outcomes and Fairness Notions

To demonstrate the generality of our results, we have replicated our experiments on Amazon review data from Section 4 under alternative fairness notions and outcomes for items. In particular, we have considered the following definitions of the item-fair solutions $f^{\mathrm{I}}$, while keeping all other setups and baselines the same as in Section 4:

- *Maxmin fairness* w.r.t. item *visibility*. Results are shown in Figure 3.
- *K-S fairness* w.r.t. item *revenue*. Results are shown in Figure 4.
- *K-S fairness* w.r.t. item *visibility*. Results are shown in Figure 5.

Overall, the results from Figures 3, 4, and 5 are consistent with those previously observed in Figure 1. In terms of the platform's revenue, we again observe the convergence of time-averaged revenue towards the optimal revenue in hindsight, confirming FORM's low regret. Regarding normalized revenue, item outcomes, and user outcomes, FORM consistently strikes an effective balance among multiple stakeholders' interests under the specified fairness parameters $\delta^{\mathrm{I}}$ and $\delta^{\mathrm{U}}$, regardless of the fairness notion or outcome that stakeholders care about. The performance of our baselines, however, is more affected by the fairness notion or outcomes when defining within-group fairness. For instance, while *min-revenue* achieves high levels of fairness for all items when the item-fair solution considers maxmin fairness w.r.t. item revenues, its performance deteriorates when items instead care about visibility or adopt K-S fairness. In comparison, FORM quickly adapts to various fairness notions and outcomes, while maintaining its good performance.

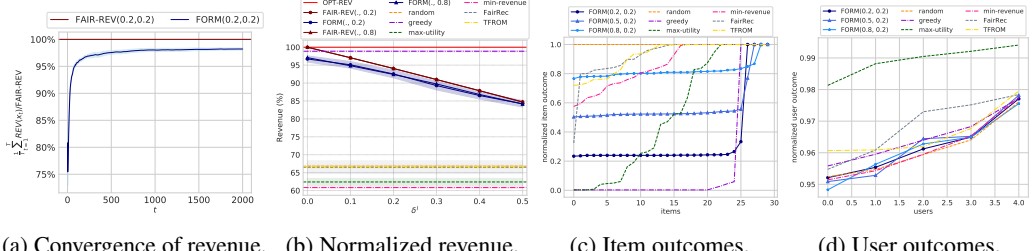

(a) Convergence of revenue.  (b) Normalized revenue.  (c) Item outcomes.  (d) User outcomes.

Figure 3: Additional experiment results for Amazon review data. Here, the item-fair solution adopts *maxmin fairness* w.r.t. item *visibility*.

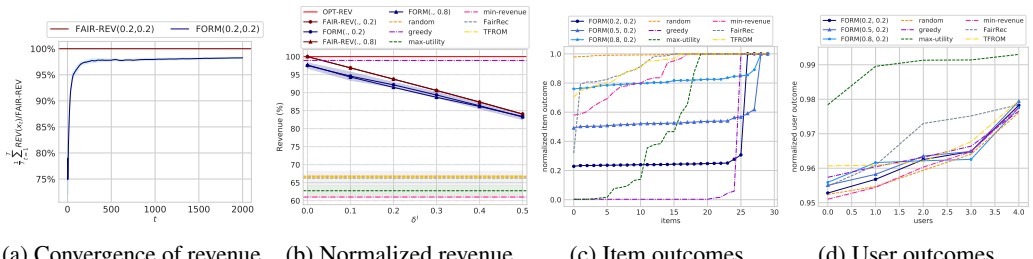

(a) Convergence of revenue.  (b) Normalized revenue.  (c) Item outcomes.  (d) User outcomes.

Figure 4: Additional experiment results for Amazon review data. Here, the item-fair solution adopts *K-S fairness* w.r.t. item *revenue*.

### F.3 Additional Experiments on MovieLens Data

To show generality of our framework/method, we have conducted additional experiments in an alternative setting (movie recommendation) using the MovieLens data, which complements our case study on Amazon review data in Section 4. Overall, the results are consistent with our Amazon case study, showing our method's effectiveness in balancing platform revenue and stakeholder fairness in various types of recommender systems.

Here, we act as a movie recommendation platform that shows a "trending action movie" to any arriving user. We considered the $N = 50$ action movies from MovieLens (ML-100K) data [35] with the highest number of ratings and clustered users into 6 types based on their preferences. As the movies are not associated with revenues, we let $r_i = 1$ for all $i \in [N]$. Here, the platform's main objective is to maximize its expected marketshare. The item-fair solution adopts maxmin fairness w.r.t. each movie's marketshare, with user utilities captured by the MNL model. In this set of experiments, we consider a total of $T = 2000$ rounds, where at each round there are 100 user arrivals. During each round $t$, we would solve the relaxed constrained optimization problem and update our recommendation probabilities only once, given scalability considerations as stated in Section 3.5; however, we will update our estimates for item weights and arrival probabilities based on Eq. (58) throughout the user arrivals.

Figure 6 shows the results obtained from our MovieLens experiments. It is evident the results are largely in line with what we observed in our case study on Amazon review data (Section 4), both in terms of the platform's revenue and the outcomes received by items/users. Our framework and method also remain effective when we only resolve the relaxed constrained optimization problem after a given number of user arrivals. It is noteworthy that in a movie recommendation setting with homogeneous revenues, the interests of the platform and the users completely align. This explains why the curves of *greedy* and *max-utility* completely overlaps with each other in our figures. However, *greedy* still suffers from 7-8% loss in marketshare (Figure 6b)), which is precisely because inadequate exploration of user data makes it overlook potentially more popular items and stick with a sub-optimal item. Overall, our algorithm FORM again adeptly balances the interests of both the platform and its stakeholders, while handling the tradeoff between learning and fair recommendation.

## G   Supplementary Lemma

In this section, we state some useful lemmas that would be invoked in this work.

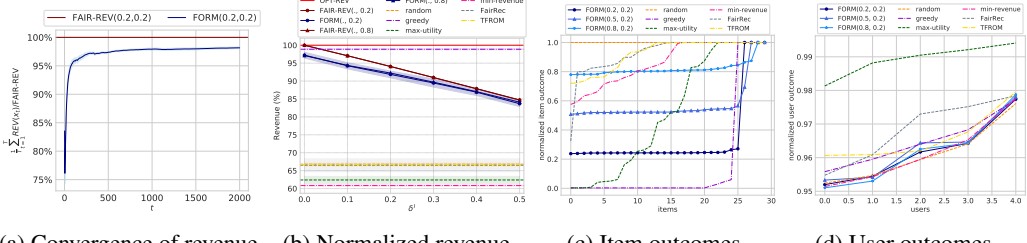

| (a) Convergence of revenue. | (b) Normalized revenue. | (c) Item outcomes. | (d) User outcomes. |

Figure 5: Additional experiment results for Amazon review data. Here, the item-fair solution adopts *K-S fairness* w.r.t. item *visibility*.

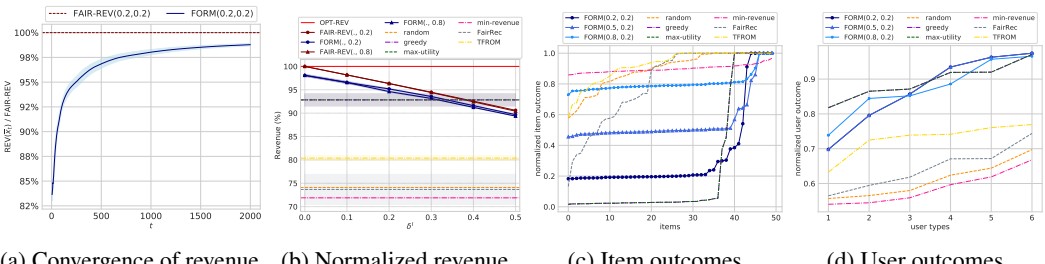

| (a) Convergence of revenue. | (b) Normalized revenue. | (c) Item outcomes. | (d) User outcomes. |

Figure 6: Additional experiment results on MovieLens data.

**Lemma G.1 (Approximations to solutions to systems of linear equations ([33], Theorem 2.1))**
*Let $A \in \mathbb{R}^{m \times n}$, and $H(A) = \max\{\|\lambda\|_1 : \lambda \text{ is an extreme point of } \sigma(A)\}$, where $\sigma(A) = \{\lambda \in \mathbb{R}^m : \lambda \geq 0, \|\lambda^\top A\|_1 \leq 1\}$. Then, for each $b \in \mathbb{R}^m$ such that $\{x \in \mathbb{R}^n : Ax \leq b\} \neq \emptyset$ and for each $x' \in \mathbb{R}^n$, we have:*

$$\min_{Ax \leq b} \|x - x'\|_\infty \leq H(A) \cdot \left\|(Ax' - b)^+\right\|_\infty,$$

*where $H(A) \geq 0$ is known as the Hoffman constant.*

**Lemma G.2 (Characterization of Hoffman constant ([56], Proposition 2))** *Let $A \in \mathbb{R}^{m \times n}$. The Hoffman constant $H(A)$ defined in Lemma G.1 can be characterized as follows:*

$$H(A) = \max_{\substack{J \subseteq \{1,\ldots,m\} \\ A_J \text{ full row rank}}} \max \left\{\|v\|_1 : v \in \mathbb{R}^J_+, \left\|A_J^\top v\right\|_1 \leq 1\right\} = \max_{\substack{J \subseteq \{1,\ldots,m\} \\ A_J \text{ full row rank}}} \frac{1}{\min_{v \in \mathbb{R}^J_+, \|v\|_1 = 1} \left\|A_J^\top v\right\|_1},$$

*where $A_J$ denotes the submatrix of $A$ that only includes the rows in $J \subseteq \{1, \ldots, m\}$.*

**Lemma G.3 (Empirical distribution concentration bound w.r.t. $\ell$-1 norm ([23], Lemma 3))**
*Let $p \in \Delta_N$ and $\hat{p} \sim Multinomial(n, p)$. Then, for any $\delta \in [0, 3\exp(-4N/5)]$, we have*

$$\mathbb{P}\left(\|p - \hat{p}\| > \frac{5\sqrt{\log(3/\delta)}}{\sqrt{n}}\right) \leq \delta.$$

**Lemma G.4 (Bernstein's inequality ([59], Theorem 8))** *Let $\mathcal{M}_1, \mathcal{M}_2 \ldots$ be a martingale difference sequence. Assume there exists deterministic sequences $c_1, c_2, \ldots$ and $V_1, V_2, \ldots$ such that $|\mathcal{M}_k| \leq c_k$ and $\mathbb{E}[\sum_{k' \in [k]} \mathcal{M}_{k'}^2] \leq V_k$ for all $k$. Then, if $\sqrt{\frac{\log(2/\delta)}{(e-2)V_n}} \leq \frac{1}{c_n}$, we have*

$$\mathbb{P}\left(\left|\sum_{k \in [n]} \mathcal{M}_k\right| \geq \sqrt{(e-2)V_n \log(2/\delta)}\right) \leq \delta. \tag{61}$$

