# OpenReview forum: "Interpolating Item and User Fairness in Multi-Sided Recommendations"
_NeurIPS.cc/2024/Conference — NeurIPS 2024 poster_

### Official Review · Reviewer_pMvw · 2024-06-25

**Soundness:** 3
**Presentation:** 4
**Contribution:** 3
**Rating:** 6
**Confidence:** 2

**Summary:**

The paper addresses the challenge of balancing multiple stakeholder interests in online recommendation systems, which include platforms, items (sellers), and users (customers). To tackle this challenge, the authors introduce a novel fair recommendation framework, FAIR, formulated as a constrained optimization problem to balance these competing interests. They further explore this problem in a dynamic online setting, introducing FORM, a low-regret algorithm that simultaneously learns user preferences and enforces fairness in real-time recommendations. Through theoretical analysis and a real-world case study, the paper demonstrates that FORM can maintain platform revenue while achieving fairness for both items and users, contributing to more sustainable and inclusive digital platforms.

**Strengths:**

The paper tackles the practically relevant and challenging problem of balancing platform revenue, user fairness, and item fairness


The paper proposes an online algorithm to learn parameters of the problem while ensuring some given fairness constraints, not merely assuming that the parameters are given as done in most existing papers


The paper provides sound and thorough regret analysis of the proposed algorithm


The empirical results on real-world data shows that the proposed method effectively and flexibly control user-item fairness while maximizing the platform revenue

**Weaknesses:**

This is not a weakness specific to this paper, but choosing the right fairness notion and the values of $\delta$ is non-trivial


Even though the paper performs offline experiments on public real-world datasets, it fails to perform any online production A/B test; so it might be risky to be overly optimistic about the empirical results of the paper until the proposed approach is verified in an online environment and provides some tangible business benefits

**Questions:**

How can we reasonably choose the right fairness notion and the values of $\delta$ in practice?


Do the authors have any plan to deploy and evaluate the proposed algorithms in some real production environment?

**Limitations:**

Yes

---

> ### Author Rebuttal · Authors · 2024-08-06
>
> Thank you for your thoughtful feedback! We'd like to address your questions below.
>
> **Regarding choosing the right fairness notion,**
> - Thank you for raising this insightful point! Indeed, choosing the right fairness notion is not trivial and very much depends on the context, such as the type of online platform and the stakeholder outcomes that matter. Here are some of the most important considerations:
>
>     - **Understanding stakeholders’ needs/outcomes.** One of the most crucial steps here is understanding the desired outcomes for items and users. Our framework, Problem (FAIR), is designed to handle various outcome functions, such as revenue, marketshare, and visibility, which can differ across platforms. For example, a video streaming platform might aim to ensure fair *visibility* for both independent content creators and popular studios, while an e-commerce platform might focus more on fair *marketshare/revenue* for small sellers versus large brands.
>
>     - **Assessing how each fairness notion aligns with the stakeholders' needs and the platform’s objective.**  Each fairness notion has a different implication, and platforms need to evaluate which best suits their goals and stakeholder needs.
>
>         For example, consider the following fairness notions from our Table 1:
>         - Maxmin fairness maximizes the outcome received by the most disadvantaged stakeholder. This can be ideal for video streaming platforms like Netflix or YouTube that wish to ensure independent and lesser-known content creators receive fair visibility alongside popular creators.
>         - K-S fairness ensures that each individual receives a fair share of his/her maximum attainable outcome. This can be suitable for platforms like LinkedIn or Indeed that wish to ensure fair opportunities for their job seekers relative to their qualifications and experience.
>
>         Platforms may also need to experiment to understand how different fairness notions impact their "price of fairness" (i.e., platform's revenue loss in enforcing fairness) before determining the right fairness notion. For instance, in our Amazon review case study, maxmin fairness appears to have a slightly higher price of fairness compared to K-S fairness (see Fig. 1b in Sec. 4 and Fig. 3b in Sec. F.2).
>
>     - **Regulatory considerations.** As we discussed in Sec. 1, one important motivation for imposing fairness in online recommendation decisions is the increasing regulatory action, such as the Digital Markets Act proposed by EU. Therefore, the choice of fairness notions can also depend on legislative or regulatory requirements. For example, since the Digital Markets Act calls for a fair and open digital marketplace, maxmin fairness can be a suitable option as it ensures even the most disadvantaged item/content creator receive a fair level of exposure.
>
> - Our fairness framework is designed for a wide range of fairness notions and outcome functions, allowing it to be tailored to each platform's specific needs. We appreciate the reviewer’s question and will include the above discussion in the camera-ready version of our paper.
>
> **Regarding how to choose the right fairness parameter $\delta_I, \delta_U$,**
> - Please see (2) in our global response for all reviewers.
>
> **Regarding online A/B test and deployment,**
> - Thank you for your question! For now, the paper has a modeling and methodological focus. We believe that the offline experiments in our case study provide a strong initial validation for the efficacy of our approaches. The Amazon review dataset was also used for evaluation in prior works on multi-sided fairness (e.g. TFROM [63]).
> - In our paper, we have also discussed how a platform can use its “price of fairness” along with online A/B testing to select the right fairness parameters, which can potentially inform the design of real-world A/B tests (see Sec C.1). Please also see (2) in our global response for all reviewers for a complete discussion on this topic.
>
>   At a high level, we propose that (i) a platform can leverage the concept of the "price of fairness” to identify a small range of fairness parameters to experiment with, ensuring that their fairness considerations do not negatively impact its key business objectives like revenue and marketshare. (ii) Upon determining a small subset of parameters for experimentation based on the desired fairness levels and acceptable price of fairness, a platform can then conduct A/B testing on segmented traffic to determine the best fairness parameters.
>
> - To facilitate potential real-world testing and deployments, we have presented this work to several online platforms and industry leaders and received positive feedback. While it is not known to us how these platforms intend to use our methodologies in their respective setups, we are open to collaborations with industry partners who have the necessary infrastructure to facilitate online A/B testing and deployments. We will mention this as an exciting future direction in our conclusion section.

---

> > ### Comment · Reviewer_pMvw · 2024-08-13
> >
> > Thank you to the authors for their useful clarification. I will maintain my positive assessment.

---

### Official Review · Reviewer_zhjK · 2024-07-02

**Soundness:** 3
**Presentation:** 3
**Contribution:** 3
**Rating:** 6
**Confidence:** 2

**Summary:**

This paper focuses on an important and interesting research direction: achieving multi-sided fairness for recommendation. Specifically, the authors aim to answer two research questions: (1) What constitutes a fair recommendation within a multi-sided platform? and (2) How would a platform implement a fair recommendation in a practical online setting? The contributions of this work include a novel multi-sided perspective that achieves within-group and across-group fairness among users/items, and also an online recommendation algorithm for multi-sided fairness. Empirical and theoretical results show the advantage of the authors' contributions over prior baselines, especially in online settings.

**Strengths:**

1. The motivation of this paper is clear, and it is well-written. The theoretical proof appears solid.
2. Prior works on fairness-aware recommendation generally focus on either a single side or an offline setting. This work combines both aspects and proposes a novel framework and algorithm, which is useful in real-world settings.
3. The authors compared their method with some well-studied offline and online fairness-aware recommendation methods, demonstrating that their proposed algorithm achieves a better tradeoff between fairness and revenue.

**Weaknesses:**

1. After doing a quick search on "multisided fairness + recommendation", it seems some related works [1, 2, 3] are missing. Even though the setting might be different, the authors should mention and discuss them somewhere in the paper since the topics seem related.
2. An efficiency comparison on the algorithm running time is missing.

Ref:

[1] Naghiaei et al. CPFair: Personalized Consumer and Producer Fairness Re-ranking for Recommender Systems, SIGIR, 2022.

[2] Wu et al. Joint Multisided Exposure Fairness for Recommendation. SIGIR, 2022.

[3] Wu et al. A multi-objective optimization framework for multi-stakeholder fairness-aware recommendation. ACM Transactions on Information Systems, 2022.

**Questions:**

1. Can the proposed method be applied to the scenario in which each user/item has multiple sensitive attributes? For instance, there may be multiple facets in describing a user, in terms of race, gender, occupation, age, income level, ... The number of facets here may be large sometimes, and it is hard to treat a specific combination of multiple facets as a single type. How does this algorithm deal with such "intersectional fairness" (e.g., to achieve fairness not only to "black"/"women"/"nurse"/"elder" separately, but also to the intersectional demographic groups)?

**Limitations:**

1. In the checklist, the authors claim that "We have clearly stated the assumptions needed for our theoretical results and the setups adopted for our numerical experiments" and "There is no negative societal impact of the work performed". However, these claims are too strong and I actually do not think the limitations and negative societal impact of this work are clearly or comprehensively discussed. For instance, one of the important challenges in recommendation is the "cold-start" problem, the author did not discuss whether their algorithm is capable of applying in this scenario, or it is also some limitation and future work. Also, one potential negative societal impact to me is the leakage of user/item sensitive information as the algorithm needs to use the "type" information. This is also not discussed or mentioned.

---

> ### Author Rebuttal · Authors · 2024-08-06
>
> Thank you for your thoughtful feedback! We'd like to address your questions below.
>
> **Regarding related works,** thank you for pointing us to these works! We'll add them and the following discussion to our related works section.
> - **A short summary of [1,2,3]**. [1] focuses on achieving producer and customer fairness in re-ranking decisions, where fairness is promoted w.r.t. item exposure and user nDCG. In contrast, our framework is not confined to a single outcome or fairness notion. [2] proposed and compared a family of joint multi-sided fairness metrics that tackle systematic biases in content exposure, while our main focus is on proposing a new fairness framework/algorithm; [3] proposed a multi-objective optimization framework which jointly optimized accuracy and fairness for consumers and producers. We proposed a different framework that optimizes the platform's revenue while incorporating fairness constraints for items/users.
> - **Key differences.** As acknowledged by the reviewer, our setting differs from [1,2,3] in several ways: (i) Our work proposes a constrained optimization framework that not only imposes fairness for multi-stakeholders (items/users), but also takes the platform’s revenue into consideration; (ii) Our framework is not confined to a single fairness/outcome notion; (iii) Most importantly, unlike works that solely focused on multi-sided fairness, we recognize the challenge of jointly handling fairness and learning (Sec. 3.2), and propose an algorithm with theoretical guarantees (Sec. 3.4). The tradeoff between learning and fair recommendation, to the best of our knowledge, has not been studied in prior works.
>
> **Regarding efficiency comparison on algorithm runtime,**
> - Please see (1) in our global response for a complete discussion on our algorithm’s complexity & scalability considerations.
> - Overall, the complexity for our algorithm will be $O((MN)^{2+1/18})$ at each round when we solve (FAIR-RELAX) and $O(N)$ at each round when we simply recommend and update user preference. As stated in our global response, Problem (FAIR-RELAX) need only be solved for $O(\log(T))$ rounds to maintain the same theoretical guarantee. In this way, our runtime will be greatly improved.
> -  In comparison to the two most relevant baselines—the per-round runtime of FairRec is $O(MN)$; the per-round runtime of TFROM is $O(NM^2log(M))$ in the offline setting and $O(N)$ in the online setting. While our runtime can be higher than these baselines during the rounds when (FAIR-RELAX) is solved, this is justified by the additional capabilities of our algorithm. Unlike the baselines, our method (i) maintains the platform’s revenue in addition to achieving item/user-fairness, and (ii) balances the processes of learning and making fair decisions, attaining strong theoretical guarantees. These enhancements lead to better performance (as validated by our case study in Sec. 4), all while maintaining a polynomial runtime under common fairness notions/outcomes.
>
> **Regarding users having multiple sensitive attributes,**
> - We’d like to first clarify that in our context, we expect user types to be primarily determined by user preferences rather than sensitive attributes such as gender, ethnicity, etc. Unlike scenarios like loan applications, where fairness involves ensuring equal opportunities across demographic groups, user-fairness in our personalized recommendation context focuses on ensuring users see items they prefer the most, rather than purely revenue-maximizing items (see our definition of user-fairness in Sec. 2.2).
>
> - With this in mind, when it comes to determining user types, we can cluster users based on their preferences, rather than directly grouping them based on genders or ethnicity. In real-world recommendation systems, platforms often have access to insensitive features such as the type of devices (Mac versus PC), zip code, as well as users’ past purchase history that can be used for clustering and determining user types. (See our response to reviewer kac1 for a complete discussion on how user type can be determined.) This is also how we determined user types in our case study (see Sec. 4). In this way, we can also incorporate multiple facets by clustering users based on comprehensive preference profiles rather than specific combinations of sensitive attributes.
>
> **Regarding the cold start problem,**
> - This can be readily addressed by design of our algorithm. As detailed in Sec. 3.1, FORM does not require any initial knowledge of user preferences or arrival rates. Instead, it employs an exploration mechanism to gather sufficient user data (see Sec. 3.3 for details on our “randomized exploration”). In practice, when new items or users join the platform mid-way through the time horizon, we can similarly enforce an initial exploration for them by having the algorithm recommend the new item with an additional small probability $\epsilon$. The amount of exploration can then diminish as we accumulate more data for these new items or users.
>
> **Regarding leakage of user/item information,**
> - Thank you for raising this important issue! As discussed previously, our user types are primarily determined by user preferences rather than sensitive attributes. To alleviate concerns on sensitive information, user clustering can be performed based on insensitive features such as types of devices, zip codes, past purchase histories, etc. As for items, we can categorize them based on publicly available attributes such as keywords, types, prices without the use of sensitive attributes, and enforce fairness w.r.t. items within the same category (see (1) of our global response for a discussion of applying our framework to the last stage of recommendation pipeline). If there is concern about using specific attributes or features, our algorithm can also be configured to exclude such information during the user clustering process. We'll make sure to include a discussion on this issue in the conclusion section.

---

> > ### Comment · Reviewer_zhjK · 2024-08-12
> >
> > Thanks the authors for the responses during the rebuttal. I keep my positive score.

---

### Official Review · Reviewer_kac1 · 2024-07-12

**Soundness:** 4
**Presentation:** 3
**Contribution:** 3
**Rating:** 7
**Confidence:** 3

**Summary:**

This paper presents a novel fairness recommendation framework called FAIR, and an online algorithm called FORM for solving multi-stakeholder fairness problems. The main contributions include 1. The FAIR framework is proposed to balance the platform revenue and the fairness of multi-stakeholders (items and users) through the form of constrained optimization problems. The framework can flexibly define fair solutions for items/users and adjust the trade-off between platform goals and stakeholder fairness.2. A FORM algorithm is designed for simultaneous learning and fair recommendation in online settings. The algorithm balances learning and fairness by relaxing the fairness constraints and introducing random exploration.3. Theoretical analyses demonstrate that the FORM algorithm achieves sub-linear regret in terms of both gain and fairness.4. A case study on Amazon review data verifies the effectiveness of the proposed method. The paper solves the fairness problem in multi-party recommender systems, and the proposed method can flexibly weigh the interests of multiple parties while maintaining the platform's revenue and demonstrates good performance in both theory and experiment. This is an important research direction, which is significant for the fairness of practical recommender systems.

**Strengths:**

1. The needs of multiple stakeholders (users, items) are considered simultaneously and Fairness parameters can be adjusted as needed.
2. Theoretical bounds on algorithmic performance are effectively proven.
3. It has a degree of practicality, with case studies demonstrating application to real data.

**Weaknesses:**

1. The computational complexity of the algorithm, especially its scalability in large-scale systems, is not discussed in detail in the paper.
2. The choice of δ_I and δ_U may require a lot of experiments to find suitable values.
3. The model assumes that the user type is known, which may not always hold in practical applications.
4. the paper mainly focuses on short-term fairness and does not delve into the impact of long-term fairness.
5. this paper mainly uses gain and fairness regret as evaluation metrics and more metrics may be needed to fully evaluate system performance.

**Questions:**

1. The paper assumes that user types are fixed, but in practice, user preferences may change over time.
2. the universality of the fairness definition: is the definition of fairness proposed in the thesis applicable to all types of recommender systems?
3. the case study uses only one category of data from Amazon, is it sufficient to demonstrate the universality of the approach?

**Limitations:**

please refer to "weakness"

---

> ### Author Rebuttal · Authors · 2024-08-06
>
> Thank you for your insightful feedback! We’d like to address your questions below.
>
> **W1. Regarding complexity and scalability,**
> - Please see (1) in our global response.
>
> **W2. Regarding choosing fairness parameters,**
> - Please see (2) in our global response for a detailed answer to this question.
> - In our global response, we discussed how a platform can select fairness parameters based on its “price of fairness” (PoF) (see Sec. C in our paper for more details). We also explained how a platform can identify a small range of promising parameters for A/B testing based on desired fairness levels and acceptable PoF, thus avoiding extensive experimentation.
>
> **W3. Regarding unknown user types,**
> - In real-world recommendation systems, platforms often have access to features that can be used to determine user types, even when sensitive features (e.g., gender, race) are restricted due to regulations. For example, Orbitz.com [https://tinyurl.com/yatek26w ] categorizes users based on the type of device (Mac versus PC), while other platforms may use zip codes for user categorization [“LARS: A location-aware recommender system"]. Historical data, such as purchase history, can also be used to cluster users, ensuring that users within each cluster have similar preferences, while users across different clusters exhibit distinct tastes. We'll include this discussion to our setup section.
>
> **W4. Regarding long-term fairness,**
> - This is a great point! Our method aims to calibrate the recommendation policy within a shorter time period (e.g., a month) when user preferences are relatively fixed. By imposing item/user-fairness over this shorter time period, we expect that in the long term, our approach can contribute in the following:
>     - **Long-term fairness.** By introducing more diversity in our recommendations, we can help mitigate the “winner-takes-all” phenomena often seen in online platforms [https://tinyurl.com/4efxc54k ].
>     - **Long-term growth.** The platform can attract more items (while retaining the existing ones) and more users (including those with niche tastes), due to its efforts to maintain fairness.
> - We leave formally quantifying such long-term effects as an exciting future direction and believe that our work lays a good foundation for such a study. We’ll discuss this in our conclusion section.
>
> **W5. Regarding alternative metrics,**
> - In our framework, we have considered various evaluation metrics:
>     - For items/users: Our framework accommodates different outcome metrics. E.g., items can consider visibility, marketshare, or revenue as their outcomes. In our case study, we verified that our fairness constraints are consistently met, regardless of the chosen metric (see Sec. F.2).
>     - For the platform, we not only evaluated the convergence of revenue regret (Fig 1a) and its revenue gain (Fig. 1b) but also examined its price of fairness under different fairness parameters (see Fig. 2).
> - We appreciate the reviewer’s suggestion and acknowledge that our evaluation can be further strengthened by incorporating additional metrics beyond the main objectives of our stakeholders, such as user satisfaction, user retention, and diversity of recommendations. These metrics can provide a more holistic view of our framework’s short/long-run impact on the system and are often best obtained through online experiments with real traffic. We’ll mention this in our future directions section.
>
> **Q1. Regarding changing user types/preference,**
> - As mentioned in our response on long-term fairness, our method attempts to calibrate the recommendation policy within the short period when user preferences are considered fixed. Over the long term, as user and item attributes evolve, adaptive fairness notions might need to be developed, which is something we mentioned in our future directions section (Sec. 5). We believe that our framework for multi-sided fairness under fixed user preference serves as a good starting point for such future studies.
> - One possible extension of our framework/method to accommodate changing user types/preferences can be inspired by existing adaptive partitioning methods (e.g.,the zooming method in “Contextual Bandits with Similarity Information”). This approach would involve using a meta-algorithm like the zooming method to update user types periodically at a slower pace, while our algorithm FORM can be applied as an inner algorithm that performs fair recommendation under the current user types/preferences. Exploring such ideas could be another interesting future direction.
>
> **Q2. Regarding applicability of fairness notions,**
> - Our proposed framework is meant to encapsulate a wide range of recommender systems, including e-commerce sites, social media, video streaming sites, etc. As remarked in Sec 2.4, we can even accommodate platforms with additional stakeholder groups (e.g., Doordash drivers, Airbnb hosts) by similarly incorporating additional fairness constraints. The generality of our framework allows these platforms to choose any fairness notions/outcomes that best suit their short-term and long-term goals (see Table 1 for example fairness notions; see Sec 2.3 for example outcome functions).
> - Determining the right fairness notion largely depends on the type of online platform and the desired outcomes for its stakeholders. See our response to reviewer pMvw for a more detailed discussion.
>
> **Q3. Regarding universality of our approach,**
> - Here, we worked with the clothing category of the Amazon review data because it was also used for evaluation in TFROM [63], which was one of the baselines we compared with. In addition to Amazon review data, we also performed experiments with the MovieLens data in a movie recommendation setting and obtained similar results. These results were previously omitted due to space constraints but are now included in our global response to demonstrate the broad applicability of our method. Please see the attached PDF in our global response.

---

> > ### Comment · Reviewer_kac1 · 2024-08-14
> >
> > Thanks to the authors for the detailed response. I keep my positive score.

---

### Official Review · Reviewer_Pu7z · 2024-07-13

**Soundness:** 3
**Presentation:** 3
**Contribution:** 3
**Rating:** 5
**Confidence:** 2

**Summary:**

The paper introduces a fair recommendation framework, FAIR, for balancing the interests of multiple stakeholders in recommendation systems, namely the platform, sellers, and users. The framework is formulated as a constrained optimization problem that addresses fairness concerns for both items and users in a dynamic online setting. The paper proposes a no-regret algorithm, FORM, which ensures real-time learning and fair recommendations. The effectiveness of the framework and the algorithm is demonstrated through theoretical analysis and a case studies on real-world data.

**Strengths:**

The paper addresses the often-overlooked complexity of balancing the interests of multiple stakeholders in optimizing recommendation systems. The proposed FAIR framework offers a solution by ensuring fairness for both items and users while maintaining platform revenue. The extension to a dynamic online setting where data uncertainty is present is reasonable. The FORM algorithm's ability to learn and adapt in real-time is a merit for practical applications in environments where user behavior and preferences are nonstationary. This work also provides a robust theoretical foundation for the proposed framework and algorithm, including proofs of sub-linear regret bounds.

**Weaknesses:**

While the paper presents promising results on a dataset with 30 items and 5 user types, it does not thoroughly address how the framework and algorithm would scale to much larger datasets with a large set of items and more diverse user profiles. My main concern is the proposed solution's scalability, a critical factor for real-world deployment but the proposed solution involves solving a constrained optimization problem at every time step, which could be complex and computationally intensive. This complexity might limit the feasibility of implementation for platforms with limited computational resources. I urge the author to include some discussion about the computational complexity of the proposed solution and also the limitations regarding the scalability.

**Questions:**

1. do you think the regret upper bound given in Theorem 3.1 is optimal? If it is not, any idea how can we improve upon this result?
2. If the platform wants to tradeoff between user fairness and item fairness, how does it need to set the parameters delta^{I} and delta^{U}? The current fairness regret simply cares about the maximum of R^I and R^U, what if we care about both, e.g., R^I + \lambda R^U? Does the proposed method still applicable to this situation?
3. I'm wondering if there could be any intrinsic tradeoff between the revenue guarantee and fairness guarantee.

**Limitations:**

Yes

---

> ### Author Rebuttal · Authors · 2024-08-06
>
> Thank you for your valuable feedback! We’d like to address each of your questions below.
>
> **W1. Regarding computational complexity and scalability,**
> - Please see (1) in our global response.
>
> **Q1. Regarding our sublinear regret upper bound**, we’d like to highlight the following:
> - **Challenges of designing no-regret algorithms in our setting.** While our algorithm essentially solves a constrained optimization problem at each round, it is important to note that one cannot directly apply other off-the-shelf algorithms for online constrained optimization with bandit feedback, including
>     - (i) Bandits with knapsack algorithms [37,15, 56]. In these works, their constraint is a budget constraint that can be directly evaluated, which is critical in helping attain a known optimal per-round regret of $O(T^{-1/2})$ in their setting.
>     - (ii) Gradient descent without a gradient (Flaxman et al. 2004, arXiv:cs/0408007), which assumes that the optimizer can analytically evaluate whether all constraints are satisfied and attains $O(T^{-1/4})$ per-round regret.
>
>   However, our setting is different from and more challenging than these off-the-shelf methods, as we can neither determine the amount of constraint violation, nor analytically evaluate the feasibility of our fairness constraints (see discussion in Sec. 3.2). Despite the extra challenge, our $O(T^{-1/3})$ regret upper bound is already superior to the $O(T^{-1/4})$ from the latter work.
>
> - **The optimal attainable regret remains an open question.** To the best of our knowledge, the optimal attainable regret for an online optimization problem with uncertain constraints like ours, where at each iteration we are only allowed a one-time bandit feedback (i.e., the purchase decision), remains an open problem. We appreciate the reviewer’s question and will mention this as an exciting future direction.
>
> **Q2.1 Regarding the tradeoff between item and user fairness,**
> - In our framework, there is no inherent tradeoff between item and user fairness. Since both are enforced as fairness constraints, increasing the level of item fairness (i.e., $\delta_I$) does not reduce the level of user fairness (i.e., $\delta_U$). The only thing to note is that if both $\delta_I, \delta_U$ approach 1, Problem (FAIR) might become infeasible, in which case the platform would need to decide whether to reduce the level of item fairness or user fairness. However, as observed in our case study, infeasibility occurs rarely—in our case study, Problem (FAIR) is only infeasible when $\delta_U $ → 1, representing a strictly user-fair solution (see Fig. 2 in Sec C.1, where we evaluate Problem (FAIR) under different parameters).
>
> **Q2.2 Regarding choosing parameters $\delta_I, \delta_U$,**
> - Please see (2) in our global response.
>
> **Q2.3 Regarding "What if we care about $R^I+\lambda R^U$",**
> - **Clarification on our metric.** We believe that there might be some misunderstanding regarding our framework's main objective and would like to first clarify the following. In contrast to a multi-objective optimization framework that **maximizes outcomes** for items and users, our framework Problem (FAIR) focuses on **minimizing regrets**. In Def. 3.1, our fairness regret is defined as $R_F(T) = \max (R^I_F(T),R^U_F(T))$ because our goal is to find a regret upper bound that can simultaneously apply to **all items/users**. Here, the fairness regret $R_F(T)$ measures the maximum gap between the fair outcome and the time-averaged outcome for any item/user. Therefore, the $O(T^{-1/3})$ regret upper bound we established for $R_F(T)$ is an upper bound for the fairness regret of any item $i$ or user $j$. As a byproduct, for any linear combination of item/user fairness regrets ${regret}_i+\lambda{regret}_j$, the same regret upper bound always holds. Our sublinear regret bound means that for any item/user, our algorithm guarantees that their long-run average outcome will reach the desired proportion of their fair outcome.
>
> - **Our framework’s advantage over multi-objective optimization problems.** We also remark that our Problem (FAIR) is a constrained optimization problem, not a multi-objective optimization problem that jointly maximizes outcomes for platform, items, users. This distinction is critical as it impacts how fairness is achieved—via constraints rather than via direct optimization of outcomes.
> Such a framework enjoys several advantages, as stated in Remark 2.1. At a high level, our framework uses interpretable parameters $\delta_I,\delta_U$ that directly relate to the level of fairness for items/users, rather than maximizing an aggregate function like $rev + \sum\lambda_iO_i+\sum\lambda_jO_j$ where the choice of Lagrangian multipliers $\lambda_i,\lambda_j$ are less straightforward.
> In our framework, if one wishes to impose fairness w.r.t. some linear combination of item/user outcomes, they can simply include a similar fairness constraint: $AO_i(x)+BO_j(x)\geq\delta\cdot(AO_i(f^I)+BO_j(f^U))$, where $f^I,f^U$ is the item-fair/user-fair solution and $x$ is the recommendation policy. All our methods/results continue to hold.
>
> **Q3. Regarding tradeoff between revenue and fairness,**
> - Our sublinear theoretical guarantees for **revenue regret** and **fairness regret** in Theorem 3.1 are always attainable under our framework/algorithm.
> - There indeed exists a tradeoff between platform’s revenue and fairness. As we highlighted in Remark 2.2 and discussed in detail in Sec. C, this tradeoff can be quantified by platform’s price of fairness (PoF), i.e., its loss in revenue due to maintaining fairness. See Fig. 2 for an illustration of PoF in our Amazon case study. As we showed in Theorem C.1, the PoF can be quantified by (i) the magnitude of fairness parameters $\delta_I, \delta_U$ and (ii) the amount of misalignment in platform’s and its stakeholders’ goals. As discussed in (2) of our global response, a platform can use PoF as an indicator for picking the right fairness parameters.

---

> > ### Comment · Reviewer_Pu7z · 2024-08-13
> > **Re: Rebuttal**
> >
> > I thank the authors for their detailed response, which addresses most of my concerns. I maintain my score and weakly lean towards acceptance.

---

### Author Rebuttal · Authors · 2024-08-06

We’d like to express our sincere gratitude to all reviewers for your insightful feedback! Below we’ve addressed several common questions from the reviewers. We’ll incorporate all discussions in our response into the revised version of the paper.

(1) **Regarding computational complexity and scalability,**
- **Complexity analysis.** For most commonly used item-fairness notions (e.g., maxmin fairness, K-S fairness, demographic parity; see Table 1) and item outcome functions (visibility, marketshare, revenue; see Sec 2.2), solving Problem (FAIR-RELAX) involves solving two linear programs with MxN variables, leading to a polynomial runtime of $\tilde{O}((MN)^{2+1/18})$. Consequently, each iteration of FORM has a worst-case complexity of $\tilde{O}((MN)^{2+1/18})$. However, practical implementations often achieve much better performance than this worst-case scenario using fast LP solvers (e.g., we use the CBC solver in PuLP for our case study).
- **Real-world scalability considerations.** Our algorithm can be further adapted with scalability in consideration.
    - **No need to solve Problem (FAIR-RELAXED) at every round.** In real-world deployment, there is no need to solve the constrained optimization problem at every user arrival. Theoretically, solving the LP in $\log(T)$ rounds can already establish the same theoretical guarantee, as shown in other works (e.g., “Optimal learning for structured bandits”). Practically, platforms can resolve the problem after every X user arrivals or at X-minute intervals, while updating user data in real-time, which can remove the computational overhead. In our MovieLens experiments (see attached PDF), we also only solved (FAIR-RELAXED) at every 1k user arrivals, and the algorithm remains effective.
    - **Applying our framework to the last stage of recommendation.** To apply our framework in real-world recommendation systems, it's not always necessary to solve a large-scale optimization problem. The recommendation process typically proceeds in stages, initially using a lightweight model to narrow down items based on keywords and filtering criteria. This allows us to focus on smaller subsets within the same category, defined by keywords, types, prices, etc., (e.g., "a silicone cake pan in the price range 10-20 usd") and maintain fairness among these smaller subsets, thus reducing the computational burden. Our fairness framework is also particularly impactful at this final stage, where items with similar qualities or features now compete for visibility/revenue, and a purely revenue-maximizing solution would thus be extremely unfair.

(2) **Regarding how to determine fairness parameters $\delta_I$ and $\delta_U$,**

There are two main factors in determining the right fairness parameters: (i) the extent of fairness needed for items/users, and (ii) the "price of fairness" the platform is willing to pay.
- **$\delta_I, \delta_U$ are interpretable, tunabled handles.** As discussed in Sec. 2.4, in our framework, $\delta_I, \delta_U$ are tunable handles that determine how much share of the fair outcome a platform would like to ensure for items/users respectively. For instance, an e-commerce platform that values high user retention might choose to impose higher user fairness to ensure user satisfaction.
- **“Price of fairness” measures the tradeoff between a platform’s revenue and fairness.** As highlighted in Remark 2.2 and discussed in detail in Sec. C, a platform’s “price of fairness” (PoF) measures a platform’s revenue loss in maintaining fairness for its stakeholders. Understanding its PoF under different parameters $(\delta_I, \delta_U)$ is crucial as a platform needs to also understand the cost of implementing fairness constraints. We show in Theorem C.1 that the upper bound of the PoF grows roughly linearly with (i) the amount of misalignment in the platform’s and its stakeholders’ objectives, (ii) the fairness parameters $\delta_I, \delta_U$.

Having these in mind, in Sec C.1 we’ve also illustrated how to determine the right fairness parameters using our case study on Amazon review data.
- **Insights from the case study.** In Fig. 2, we explored the PoF under various $\delta_I, \delta_U$ on Amazon review data. We found that $\delta_U$ has little impact on PoF as the user-fair constraints are not binding, while $\delta_I$ impacts the PoF in a roughly piecewise linear manner. This suggests that under Amazon review data, the platform can achieve high user-fairness at little cost, while it can gauge the amount of tradeoff between its revenue and item fairness (the slope between PoF and $\delta_I$) by adjusting $\delta_I$ incrementally and performing experimentation.
- **Guidelines to set $\delta_I, \delta_U$.** Similar methods can be applied to different online platforms in their respective contexts. A platform should first identify the binding constraints by determining which stakeholders experience the most unfairness under the current recommendation policy. Then, using the piecewise linear relationship, it can gauge the amount of tradeoff between binding fairness constraint and its PoF. Based on the desired fairness levels and acceptable PoF, the platform can narrow down the range of fairness parameters to experiment with. Once a small subset of promising fairness parameters is identified, a platform can conduct online A/B tests by splitting its traffic (e.g., 10-20%) to experiment with these parameters in parallel. This allows the platform to pick the most effective fairness parameters without extensive experimentation on all possible sets of parameters.

(3) **Additional Experiments on MovieLens Data.** In response to reviewer kac1’s question, in the attached PDF we’ve included experiments on the MovieLens data to validate the efficacy of our framework in an alternative setting (movie recommendation). The results are consistent with our Amazon case study, showing our method's effectiveness in balancing platform revenue and stakeholder fairness.

---

### Decision · Program_Chairs · 2024-09-25

**Decision:**

Accept (poster)

**Comment:**

This paper addresses the important problem of balancing multiple stakeholder interests in online recommendation systems. The authors propose a novel framework called FAIR that aims to optimize platform revenue while maintaining fairness for both items (sellers) and users (customers). They also introduce an algorithm, FORM, designed to learn user preferences and enforce fairness constraints simultaneously in an online setting. The reviewers unanimously agree on the significance and timeliness of the problem addressed. The paper's key strengths include a flexible framework that can accommodate various fairness notions and stakeholder objectives, a theoretical analysis providing sublinear regret bounds for the proposed algorithm, and empirical validation on real-world data demonstrating the effectiveness of the approach. The paper makes a solid contribution to the field of fair recommendation systems and has the potential for significant impact in both academic and practical contexts.
However, the reviewers have raised some concerns that the authors should address in the final version, for example, computational complexity and scalability (a more detailed discussion on the algorithm's scalability for large-scale systems and potential optimizations), parameter selection (offer better guidance on selecting appropriate fairness notions and parameters in practical settings). The assumption around static user preferences was deemed as restrictive by the authors, and the reviewers also asked questions about the long-term effects based on (for example) platform dynamics and stakeholder behaviors. The authors are encouraged to pursue real-world deployments as future work. In conclusion, despite the noted areas for improvement, this paper presents a significant contribution to the field of fair recommendation systems. The novel multi-stakeholder perspective, coupled with the theoretical guarantees and empirical validation, make a compelling case for acceptance. The authors are advised to address the reviewers' concerns in their final revision to further enhance the paper's impact.